# REVISIT KERNEL PRUNING WITH LOTTERY REGULATED GROUPED CONVOLUTIONS

**Shaochen (Henry) Zhong**[1], **Guanqun Zhang**[2], **Ningjia Huang**[1], and **Shuai Xu**[1]

[1]Department of Computer and Data Sciences, Case Western Reserve University
[1]{sxz517, nxh239, sxx214}@case.edu
[2]Center for Combinatorics, Nankai University
[2]zhanggq1994@mail.nankai.edu.cn

## ABSTRACT

Structured pruning methods which are capable of delivering a densely pruned network are among the most popular techniques in the realm of neural network pruning, where most methods prune the original network at a filter or layer level. Although such methods may provide immediate compression and acceleration benefits, we argue that the blanket removal of an entire filter or layer may result in undesired accuracy loss. In this paper, we revisit the idea of kernel pruning (to only prune one or several $k \times k$ kernels out of a 3D-filter), a heavily overlooked approach under the context of structured pruning. This is because kernel pruning will naturally introduce sparsity to filters within the same convolutional layer — thus, making the remaining network no longer dense. We address this problem by proposing a versatile grouped pruning framework where we first cluster filters from each convolutional layer into equal-sized groups, prune the grouped kernels we deem unimportant from each filter group, then permute the remaining filters to form a densely grouped convolutional architecture (which also enables the parallel computing capability) for fine-tuning. Specifically, we consult empirical findings from a series of literature regarding *Lottery Ticket Hypothesis* to determine the optimal clustering scheme per layer, and develop a simple yet cost-efficient greedy approximation algorithm to determine which group kernels to keep within each filter group. Extensive experiments also demonstrate our method often outperforms comparable SOTA methods with lesser data augmentation needed, smaller fine-tuning budget required, and sometimes even much simpler procedure executed (e.g., one-shot v. iterative). Please refer to our GitHub repository for code.

## 1 INTRODUCTION

The applications of convolutional neural networks (CNNs) have demonstrated proven success in various computer vision tasks (Voulodimos et al., 2018). However, with modern CNN architectures being increasingly deeper and wider, over-parameterization has become one of the major challenges of deploying such models to devices with limited computational resources and memory capacity (Frankle & Carbin, 2019). Therefore, the study of network pruning — the technique of removing redundant weights from the originally trained network without significantly sacrificing accuracy — has been an important subject both for practical concerns (Mozer & Smolensky, 1989) and better understanding of the properties and mechanisms of neural networks (Arora et al., 2018).

In the realm of CNN pruning, a spectrum of techniques have been studied where the two ends are populated by *structured* pruning and *unstructured* pruning methods (Mao et al., 2017). Methods from the former end often propose to remove redundant weights in groups while following some geometrical constraints — such as removing a certain filter or layer. The methods from the latter end, on the other hand, prune the network with a more fine-grained view where they evaluate every weight individually. Yet, there are many other methods lay in between of the two ends. Methods that are more "unstructured" are believed to be capable of yielding better accuracy retention with a commensurate amount of parameters pruned, due to having a higher degree of freedom on how

and where to introduce sparsity to the originally dense network. Empirical findings also support this claim (Liu et al., 2019; Mao et al., 2017).

Despite having advantages on accuracy retention, the resultant networks from methods closer to the unstructured end will be more sparse and less regulated on where to introduce sparsity. It may not provide actual compression and acceleration without relying on custom-indexing, sparse convolution libraries, or even dedicated hardware devices; thus, limits the deployability of such methods (Mao et al., 2017). Meanwhile, methods closer to the structured end, by preserving a more regulated resultant network, are more likely to be library/hardware-friendly. The most deployable structured pruning method may deliver a densely resultant network and therefore gain immediate compression and acceleration benefits. We denote this kind of pruning method as *densely structured*.

Naturally, many scholars want to develop new methods within the realm of densely structured pruning but with better accuracy retention. The majority of densely structured pruning methods focus on pruning the original network at a filter or layer level. We argue that the blanket removal of an entire filter or layer may harm the representation power of the network and result in undesired accuracy loss — as removing a filter would consequently remove all feature maps generated by such filter. Even worse, removing a layer would eliminate more feature maps and even face the danger of *layer-collapse*, a phenomenon of having an untrainable pruned network due to premature pruning of an entire layer (Tanaka et al., 2020).

In this paper, we revisit the idea of kernel pruning (to only prune one or several $k \times k$ kernels from a 3D-filter, instead of an entire one) as an alternative and less aggressive pruning approach with higher degree of freedom. We hypothesize that by not removing the entire filter, the representation power of the original network will be better preserved. Although the idea of kernel pruning is nothing too novel (as it is simply a special case of individual weights pruning with 100% of weights of a kernel pruned), it is mostly applied under the context of unstructured pruning or structured pruning methods which may not deliver a dense pruned network (Mao et al., 2017; Ma et al., 2020). This is because a direct implementation of kernel pruning with no constraint would introduce sparsity across the network and therefore make the pruned network no longer dense. We address this problem by proposing a versatile grouped pruning framework, where we:

1. Cluster similar filters from each convolutional layer into a (predefined) number of equal-sized filter groups.

2. For each filter group, identify a certain portion of grouped kernels to prune according to the required pruning ratio.

3. Permute the remaining filters to form a densely grouped convolutional architecture according to the number of groups used in step 1.

Like most other post-train grouped pruning methods, we face the challenge of determining which clustering schemes and which importance metrics to use in step 1 and 2 of the above procedure. Upon investigations and experiments, we discovered that a classical clustering scoring system (e.g., the *Silhouette score*) might not capture the better clustering scheme in regard to accuracy retention. Yet many filter importance metrics require sophisticated procedures, which are not computationally friendly or easy to execute when applied at a kernel level. We address the first challenge by consulting model-generated information — in this case, the empirical findings on *Lottery Ticket Hypothesis* (LTH) and related literature on weights shifting — to develop a scoring system that identifies the optimal clustering scheme among options per each convolutional layer (Frankle & Carbin, 2019; Renda et al., 2020). For the second challenge, we design a simple and cost-efficient greedy algorithm with multiple restarts to generate multiple candidate kernel selection queues and identify the one queue where the preserved kernels are most "distinctive" from each other yet "similar" to the pruned kernels. The main contributions and advantages of our method are:

- **Bring back an overlooked approach**: We brought attention to the heavily overlooked approach of kernel pruning under the context of densely structured pruning.

- **Simple but effective — a vanilla adaptation of our framework outperforms sophisticated variants of other comparable frameworks (e.g., filter pruning)**: Even by just applying well-understood classical mathematical tools, extensive experiments demonstrate our method outperforms comparable SOTA methods across different networks and datasets.

Additionally, our method often needs less data augmentation, a smaller fine-tuning budget, and it executes without requiring any custom retraining, special fine-tuning, or iterative prune-train cycles — which is rare for the approaches relying on LTH-related studies.

- **Better longevity**: We developed a framework that is compatible with further-developed/discovered clustering schemes and inductive biases, or more advanced variations upon them. This overcomes one of the major drawbacks of many filter pruning methods: as most of them propose different filter importance metrics that are largely incompatible with each other either in terms of their procedures or computational requirements.

- **Improved deployability**: The resultant network of our method is structured as a densely grouped convolution, which enables parallel computing capability and greatly increases the practical deployability of our methods: as we can now share the required computation and memory footprint across multiple end-user devices, where most of them have very limited said resources individually (e.g., IoT devices, mobile phones, and wearable technologies).

- **General impact on LTH and beyond network pruning**: Please refer to Section 5.

## 2 RELATED WORK

Many prior arts have explored the possibility of obtaining a smaller model with comparable performance by removing redundant weights (Zhu & Gupta, 2018; Han et al., 2016), filters (Molchanov et al., 2017; Yu et al., 2018; He et al., 2019; Wang et al., 2019a), layers (Wang et al., 2019b; Lin et al., 2019), image input (Howard et al., 2017; Han et al., 2020), or from all three dimensions (Wang et al., 2021). It is clear that filter pruning attracts the most attention among all structured pruning approaches.

Our method is inspired by **grouped convolution**, a widely adopted convolutional architecture which could be implemented efficiently on common devices (Iandola et al., 2017). Although kernel pruning used together with filter clustering is not a popular trend, we have seen such a combination in work like Yu et al. (2017). However, the proposed method by Yu requires iterative analysis of many different intermediate feature maps per layer, involves a complex knowledge distillation application during the fine-tuning stage, and lacks comparable experiment results to recent pruning literature. Where our method (and concurrent work like Zhang et al. (2022)) provides a much cleaner adaptation of the abovementioned combination that delivers beyond-SOTA performance with a straightforward one-shot pruning and standard fine-tuning procedure. In addition, our method consults empirical findings on the **lottery ticket hypothesis** and its derived literature regarding weights shifting (Frankle & Carbin, 2019; Renda et al., 2020; Zhou et al., 2019), and we propose a novel greedy kernel pruning algorithm that is again simple, efficient, yet effective — more on this in Section 3.

## 3 PROPOSED METHOD

### 3.1 PRELIMINARIES

Assume a convolutional neural network $\boldsymbol{W}$ has $L$ convolutional layers, we denote the $\boldsymbol{W}^\ell$ to be the $\ell$-th convolutional layer of $\boldsymbol{W}$ (for $\ell \in \{\mathbb{Z}^+ \mid [1, L]\}$). Therefore, we shall have a 4-D tensor $\boldsymbol{W}^\ell \in \mathbb{R}^{C_{\text{out}}^\ell \times C_{\text{in}}^\ell \times H^\ell \times W^\ell}$ where $C_{\text{out}}^\ell$ represents the number of filters in $\boldsymbol{W}^\ell$ (also known as the number of *output channels* in some literature), $C_{\text{in}}^\ell$ represents the number of kernels per filter (a.k.a. number of *input channels*), and $H^\ell \times W^\ell$ represents the size of each kernel.

The overall procedure of our method can be mainly divided into four stages: 1) Clustering filters into $n$ equal-sized groups, where the best clustering scheme for each convolutional layer is determined using the *tickets magnitude increase* score derived from prior arts on lottery ticket hypothesis and weight-shifting; 2) Evaluating several candidate grouped kernel pruning strategies generated by a greedy approximation algorithm with multiple restarts, where the strategy with preserved grouped kernels that are most distinctive from each other, yet most similar to the pruned grouped kernels gets selected; 3) Permuting the preserved filters to form a grouped convolutional architecture with $n$ groups; 4) Fine-tuning the pruned and grouped network to recover accuracy lost from pruning.

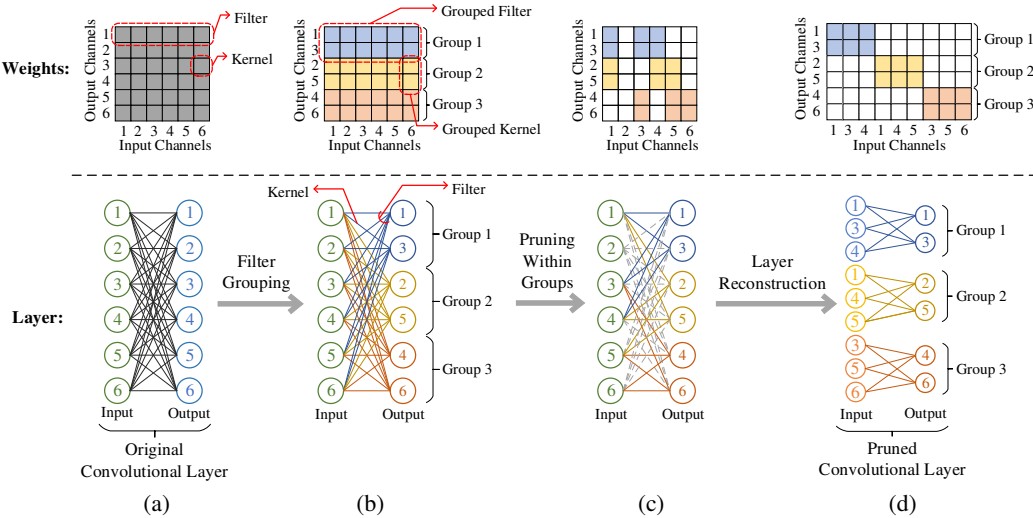

Figure 1: Steps of our method

## 3.2 CLUSTERING FILTERS INTO GROUPS

The first step of our method is to cluster filters from the same convolutional layer into $n$ equal-sized groups. Known that for a layer $\boldsymbol{W}^\ell$ we have $C_{\text{out}}^\ell$ filters, there shall be $C_{\text{out}}^\ell/n$ filters inside each equal-sized filter group. We denote $\boldsymbol{F}_i^\ell$ to be the $i$-th filter in $\boldsymbol{W}^\ell$ (namely, $\boldsymbol{F}_i^\ell = \boldsymbol{W}_{[i,:,:,:]}^\ell$) for $i \in \{\mathbb{Z}^+ \mid [1, C_{\text{out}}^\ell]\}$. Filter clustering is a maturely adopted technique in network pruning since it is a widely accepted assumption that when similar filters are clustered together, the representation power of some filters can be covered by the rest of the filters in the same group (which therefore enables the potential of pruning). Additionally, this technique drastically decomposes the scope of the problem, as we may now proceed to evaluate in a group-by-group fashion instead of evaluating all of the filters from $\boldsymbol{W}^\ell$ at the same time.

Many prior arts have developed methods on filter clustering with linearity assumptions (Guo et al., 2020), via retraining with a custom loss function (Wu et al., 2018), or through an iterative process (Yu et al., 2017). We argue that since each filter $\boldsymbol{F}_i^\ell$ is a tensor of $C_{\text{in}}^\ell \times H^\ell \times W^\ell$, considerations regarding non-linearity and high-dimensional relationships should be added. Therefore, we utilize the following three combinations of proven mathematical tools on dimensionality reduction and clustering in order to cluster filters from each layer into $n$ equal-sized groups.

### 3.2.1 AVAILABLE CLUSTERING SCHEMES

To reduce the dimension of filters in $\boldsymbol{W}^\ell$ and cluster them into $n$ equal-sized groups, we utilize the following three sets of methods: **K-PCA + $k$-Means**, **Spectral Clustering + $k$-Means**, and $t$-**SNE + DBSCAN** (with slight modifications done to obtain the desired cluster size and number). Since all five methods here have been maturely studied, we will omit the introduction of these methods; please refer to section A.5 for details. We denote each set of methods as a *clustering scheme*.

### 3.2.2 DECIDE OPTIMAL CLUSTERING SCHEME PER LAYER WITH TMI SCORE

With the three proposed and many other unimplemented clustering schemes, one natural question to ask is how are we suppose to decide the optimal clustering scheme? And at what scale should we make such a decision? The latter question was relatively easier to answer as we already set on clustering filters into groups in a layer-by-layer fashion, so it is reasonable enough to find out the most suitable clustering scheme for each layer. But to address the former question, we need to find a way to quantify the "quality" of proposed clustering results. Mathematicians have provided us with classical tools such as the *Silhouette score* to measure the quality of clusters. However, experiment results suggest such tools can hardly identify the better cluster method under the context of accuracy retention (A.1.1).

We hypothesize that this is because tools like the Silhouette score focus on properties regarding clusters themselves — such as distances between clusters, or cohesion within each cluster — while

accuracy retention relies on filter importance, or balance between filter groups under the context of the network. Therefore, we decide to introduce network-induced information to help us quantify the quality of different clustering results: where we consult empirical findings from a series of literature regarding *Lottery Ticket Hypothesis* and weights shifting.

**Background on Lottery Ticket Hypothesis, Weights Shifting, and Weights Rewinding**  For a dense convolutional neural network $\boldsymbol{W}$ initialized with weights of $\boldsymbol{W}_0$, the conventional post-train pruning procedure has three steps:

1. **Train** the proposed network for $t$ epochs. Where we denote the network has weights of $\boldsymbol{W}_k$ at the $k$-th epoch for $0 \leq k \leq t$.

2. **Prune** the network with weights $\boldsymbol{W}_t$ according to the defined pruning ratio and achieve subnetwork $\boldsymbol{W}_t^{'}$, where $\boldsymbol{W}_t^{'} \subset \boldsymbol{W}_t$.

3. **Fine-tune** the network with $\boldsymbol{W}_t^{'}$ and $f$ more epochs to recover accuracy.

The *Lottery Ticket Hypothesis* claims that instead of fine-tuning upon the subnetwork $\boldsymbol{W}_t^{'}$, there exists a subnetwork $\boldsymbol{W}_k^{'}$ (defined as $\boldsymbol{W}_t^{'}$ but replaced with its counterpart weights from $\boldsymbol{W}_k$) that, when trained in isolation for $f$ more epochs, may match or outperform the test accuracy of the fine-tuned $\boldsymbol{W}_t^{'}$ achieved from step 3 above. This $\boldsymbol{W}_k^{'}$ is therefore known as the *winning ticket* (Frankle & Carbin, 2019).

A vast amount of research has been done to demonstrate the existence of winning tickets across different networks and datasets, making the lottery ticket hypothesis one of the most tested inductive biases among neural networks (Renda et al., 2020). Scholars have additionally investigated the relationship between the winning ticket's weights and final weights in regard to accuracy retention. Zhou et al. (2019) did one of the most thorough experiments on exploring such relationship by deploying nine different zero *mask criteria* related to initial weights, final weights, and weights shifting during training. They concluded that $||\boldsymbol{w}_t| - |\boldsymbol{w}_0||$ (where $\boldsymbol{w}_t$, $\boldsymbol{w}_0$ respectively represent the same weight in $\boldsymbol{W}_t$ and $\boldsymbol{W}_0$), the *magnitude increase* of a weight at its initialization and after training, has demonstrated most significant positive correlation with accuracy retention.

Note that the Zhou et al. (2019) experiments were conducted under the assumption that the winning ticket exists at $\boldsymbol{W}_0$. This is because, for early or even concurrent arts on the lottery ticket hypothesis, scholars have their disagreements on whether to reinitialize the weights of $\boldsymbol{W}_t^{'}$ to their initial values (namely, $k = 0$ for $k$ in $\boldsymbol{W}_k^{'}$) (Frankle & Carbin, 2019), to near initialization ($0 < k \ll t$) (Renda et al., 2020), or just to reset randomly (Liu et al., 2019). However, with more comprehensive experiments conducted, the findings reveal there are multiple winning tickets that exist within a range of epochs starting from near initialization: such that for any $k$ where $0 < k_1 \leq k \leq k_2 < t$, $\boldsymbol{W}_k^{'}$ can be a winning ticket (with $k_1$ being close to 0). This technique is referred as *weights rewinding* and we denote the range of $[k_1, k_2]$ as the *tickets window* (Renda et al., 2020).

**Ticket Magnitude Increase scoring system**  Based on the fact that a winning ticket may occur at near initialization but not necessary exactly at initialization, we slightly modify the *magnitude increase* criteria from Zhou et al. (2019) by replacing the initialization weights as the ticket weights. Thus, we define the *ticket magnitude increase* score (hereinafter "TMI score") as $||\boldsymbol{w}_t| - |\boldsymbol{w}_k||$ for $0 < k \ll t$ (where $\boldsymbol{w}_t$, $\boldsymbol{w}_k$ respectively represent the same weight in $\boldsymbol{W}_t$ and $\boldsymbol{W}_k$). Since we aim to use such scoring system to govern the clustering quality of filter groups, we further expand this scoring system from individual weight to a filter-level. For a filter $\boldsymbol{F}_i^{\ell} \in \mathbb{R}^{C_{\text{in}}^{\ell} \times H^{\ell} \times W^{\ell}}$, we denote the weight at $(h, w)$ index from the $c$-th kernel of $\boldsymbol{F}_i^{\ell}$ as $\boldsymbol{W}_{[i,c,h,w]}^{\ell}$; so the TMI score for filter $\boldsymbol{F}_i^{\ell}$ in regard to a winning ticket at the $k$-th epoch should be defined as the following.

$$\text{TMI}(\boldsymbol{F}_i^{\ell}, k) = \sum_c^{C_{\text{in}}^{\ell}} \sum_h^{H^{\ell}} \sum_w^{W^{\ell}} (||\boldsymbol{W}_{t_{[i,c,h,w]}}^{\ell}| - |\boldsymbol{W}_{k_{[i,c,h,w]}}^{\ell}||). \tag{1}$$

Similarly, for a filter group $\boldsymbol{g}^{\ell}$ of $\{\boldsymbol{F}_i^{\ell}, \boldsymbol{F}_j^{\ell}, \dots\}$, its TMI score would be:

$$\text{TMI}(\boldsymbol{g}^{\ell}, k) = \sum_{g_i \in \boldsymbol{g}^{\ell}} \text{TMI}(\boldsymbol{F}^{\ell}_{g_i}, k). \tag{2}$$

**Qualifying cluster results with TMI scores**   With the *TMI score* for a filter group defined, the next question to address is what makes a good clustering result in regard to TMI scores? We already observed that tools like the Silhouette score could not capture the better clustering result in terms of accuracy retention, and we hypothesized that this is due to the lack of attention on **filter importance** and **balance between filter groups**. Since Zhou et al. (2019) has established the relationship between better accuracy retention and weights with larger magnitude increase, we expect such relationship will expand to a filter level where filters with large TMI scores may help on accuracy retention and therefore be deemed "more important." We confirm such assumption in A.4.1.

In terms of the balance between multiple filter groups — with each filter's TMI score being an indicator of its importance — we would prefer clustering results where all filter groups have a similar TMI score. As if one filter group's TMI score is significantly larger than another filter group, the former group will presumably contain more "important filters" than the latter group. When we proceed to prune the same ratio of grouped kernels out of both groups, the network will lose more "important kernels" and therefore damage the accuracy retention. In other words, we want to minimize the intervals between the TMI scores of all filter groups. So for a clustering result $\boldsymbol{G}^{\ell}$, assume $\boldsymbol{g}^{\ell}_i, \boldsymbol{g}^{\ell}_j, ..., \boldsymbol{g}^{\ell}_{n-1}, \boldsymbol{g}^{\ell}_n$ are the $n$ filter groups of $\boldsymbol{G}^{\ell}$ sorted in descending order according to their TMI scores. We denote the intervals of $\boldsymbol{G}^{\ell}$ with respect to winning ticket $k$ as:

$$\text{Interval}(\boldsymbol{G}^{\ell}, k) = [\text{TMI}(\boldsymbol{g}^{\ell}_i, k) - \text{TMI}(\boldsymbol{g}^{\ell}_j, k), \ ..., \ \text{TMI}(\boldsymbol{g}^{\ell}_{n-1}, k) - \text{TMI}(\boldsymbol{g}^{\ell}_n, k)], \tag{3}$$

where the clustering result $\boldsymbol{G}^{\ell} = \arg\min_{\boldsymbol{G}^{\ell}}(\text{Sum}[\text{Interval}(\boldsymbol{G}^{\ell}, k)])$ is preferred. To further identify the clustering result with the most balanced filter groups in terms of TMI scores, we also want to minimize the variance of intervals between the TMI scores of all filter groups. By combining the two heuristics with a balancing parameter $\alpha$, the overall scoring system for a clustering result $\boldsymbol{G}^{\ell}$ is defined as:

$$\underset{\text{filters clustering}}{\text{Score}}(\boldsymbol{G}^{\ell}, k) = R(\text{Sum}[\text{Interval}(\boldsymbol{G}^{\ell}, k)]/(n-1)) + \alpha \cdot R(\text{Var}[\text{Interval}(\boldsymbol{G}^{\ell}, k)]), \tag{4}$$

where the function $R$ denotes the ranking of the current $\boldsymbol{G}^{\ell}$ against other proposed clustering results (for ranking in an ascending order). For example, if clustering result $\boldsymbol{G}^{\ell}$ has the smallest $\text{Sum}[\text{Interval}(\boldsymbol{G}^{\ell}, k)]$ among all proposed clustering results, then $R(\text{Sum}[\text{Interval}(\boldsymbol{G}^{\ell}, k)]) = 1$. We choose to use rank instead of the raw value as we are only interested in if one clustering result is better than another on one particular criterion (interval mean or interval variance), but not by how much. Therefore, by projecting both criteria to rank indices, we avoid the imbalance between the raw values of the two criteria. The $\arg\min$ of this Equation 4 yields the best $\boldsymbol{G}^{\ell}$.

**Increase robustness with multiple ticket evaluations**   Now we defined how to identify the preferred clustering scheme from all proposed $\boldsymbol{G}^{\ell}$s in layer $\boldsymbol{W}^{\ell}$, the last challenge is to find the winning ticket $k$ for TMI score calculations. Unfortunately, finding the winning ticket requires an extremely computationally intensive process called *iterative magnitude pruning*, in which multiple training and pruning cycles are involved (Frankle et al., 2020). To make matters worse, since we know that there are often multiple winning tickets available from the training process (Renda et al., 2020), picking one ticket but not the other will potentially yield a different set of preferred clustering schemes for the network. This makes the $k$ both a hard-to-find task and a sensitive parameter to tune.

We address both challenges by relying on the same finding from Renda et al. (2020), which demonstrates that for any $k$ where $0 < k_1 \leq k \leq k_2 < t$, $\boldsymbol{W}^{'}_k$ can be a winning ticket. This tickets window $[k_1, k_2]$ is shown to be robust for the similar models across different datasets and pruning methods. Thus, we can decide which model we will prune on, identify its tickets window by consulting experiment results from Renda et al. (2020), and truncate some epochs in such window to conduct multiple evaluations. The optimal clustering scheme for $\boldsymbol{W}^{\ell}$ will be the one that most often yield

as the preferred clustering scheme per different $k$ settings[1]. By doing this, we first avoid searching for the winning ticket $k$, which greatly reduces the computational power needed to implement our method. Secondly, it increases the robustness of our method, since the optimal clustering scheme per layer is no longer sensitive to the choice of $k$. Last, granted the wide range of $[k_1, k_2]$ observed from most networks, practically we don't even need to search for the exact $(k_1, k_2)$ pair; an approximated guess with a reasonable overlap with the ideal $(k_1, k_2)$ would often suffice. The kind of tolerance, combined with the proven generalizability of winning tickets across different tasks, models, and datasets (Morcos et al., 2019), with more efficient methods of finding the winning tickets become available (Tanaka et al., 2020; You et al., 2020). Our proposed method can also be applied to experiments outside of Renda et al. (2020).

### 3.3 EVALUATE CANDIDATE PRESERVED GROUPED KERNEL QUEUES

With filters of $\boldsymbol{W}^\ell$ grouped into filter groups $\{\boldsymbol{g}_1^\ell, \boldsymbol{g}_2^\ell, ..., \boldsymbol{g}_n^\ell\}$, we will carry out the pruning procedure within each filter group. Let $\boldsymbol{K}_i^{\boldsymbol{g}^\ell}$ be the $i$-th grouped kernel in filter group $\boldsymbol{g}^\ell$ (for $1 \leq i \leq \boldsymbol{C}_{\text{in}}^\ell$) such that $\boldsymbol{K}_i^{\boldsymbol{g}^\ell}$ is the collection of the $i$-th kernels from every filter in $\boldsymbol{g}^\ell$. The pruning of a filter group $\boldsymbol{g}^\ell$ can be viewed as the task of identifying a set group kernels in $\boldsymbol{g}^\ell$ that needs to be preserved. To start off such identification, we first hypothesize a set of grouped kernels that are most "distinctive" from each other may provide the best help on preserving the representation power of the original filter group. We define a distance matrix between grouped kernels from $\boldsymbol{g}^\ell$ as $D(\boldsymbol{g}^\ell))$ where $D(\boldsymbol{g}^\ell)_{[u,v]} = |\boldsymbol{K}_u^{\boldsymbol{g}^\ell} - \boldsymbol{K}_v^{\boldsymbol{g}^\ell}|$. This $D(\boldsymbol{g}^\ell))$ shall be a symmetric matrix with its diagonal full of 0s.

With this matrix, we may now translate this problem as a *maximum edge-weight connected subgraph* problem: Given a filter group $\boldsymbol{g}^\ell$ represented as an undirected complete graph $G(V, E)$ with edge weights $w(u, v) = D(\boldsymbol{g}^\ell)_{[u,v]}$ between nodes $u$ and $v$, find a subset $V^* \subseteq V$ with $|V^*| = s$ where the total edge sum of $G[V^*] := (V^*, E \cap \binom{V^*}{2})$ is maximal (with $s = (1 - \text{pruning rate}) \cdot C_{\text{in}}^\ell$).

The brute force solution of this problem requires iterating through every possible subset of grouped kernels with size $s$ and calculate their edge sum. With $\binom{|V|}{s}$ possible subsets available and each subset having $\binom{s}{2}$ edges, this procedure has a time complexity of $\mathcal{O}(s^2 \cdot |V|^s)$. This can be quite a compute-extensive task given each graph $G(V, E)$ deduced from $\boldsymbol{g}^\ell$ has a $|V|$ equals to $C_{\text{in}}^\ell$; which can be as large as 512 in models like ResNet-101. To address this, we hereby propose a simple greedy algorithm with multiple restarts that may approximate the $V^*$ in question.

Our approximated solution was based on the hypothesis that, assume we already have a set of nodes $U^* \subset V^*$ representing grouped kernels which will be preserved. To find the next node $u_{\text{next}} \in V^*$ for $u_{\text{next}} \notin U^*$, such $u_{\text{next}}$ should have a maximal edge sum to all nodes in $U^*$. Namely, $u_{\text{next}} = \arg\max_{u_{\text{next}}}(\sum_{u \in U^*} w(u, u_{\text{next}}))$. With this, for a filter group $\boldsymbol{g}^\ell$ we may simply set the first grouped kernel $u_{\text{initial}} \in U^*$ to be every grouped kernel $\boldsymbol{K}_i^{\boldsymbol{g}^\ell} \in \boldsymbol{g}^\ell$, find the $u_{\text{next}}$ until $|V^*| = s$, and obtain $C_{\text{in}}^\ell$ pruning strategies. This trick is known as *multiple restarts* and is widely adopted in many algorithms such as $k$-Means clustering.

We formalize the procedure of our approximation algorithm in pseudocode at A.2.1. The core of our algorithm resides at line 10, where we use the computed result of previous row in $M$ to reduce time complexity of our algorithm. With $M \in \mathbb{R}^{s \times C_{\text{in}}^\ell}$ and filled $C_{\text{in}}^\ell$ times, our approximation algorithm therefore has a time complexity of $\mathcal{O}(s|V|^2)$ (as $|V| = C_{\text{in}}^\ell$). This provides a sensible improvement upon the $\mathcal{O}(s^2 \cdot |V|^s)$ brute force solution. To further and better identify the best grouped kernel pruning strategy among the $C_{\text{in}}^\ell$ candidate strategies in $PS$, we define the score of a strategy $V^*$ on $\boldsymbol{g}^\ell$ as the following:

$$\underset{\text{grouped kernel pruning}}{\text{Score}}(V^*, \boldsymbol{g}^\ell) = \sum_{s_u, s_v \in \binom{V^*}{2}} w(s_u, s_v) - \beta \Big( \sum_{p \in \boldsymbol{g}^\ell \setminus V^*} \big( \sum_{s_{i=1} \in V^*}^{s_\gamma} w(p, s_i) \big) \Big). \quad (5)$$

---

[1] For example, suppose $k_1 + 29 = k_2$, clustering scheme $A$ is chosen as the preferred clustering scheme by Equation 4 for 20 times on layer $\boldsymbol{W}^\ell$, yet clustering schemes $B$ and $C$ are chosen 5 times respectively. In this case, $A$ will be the optimal clustering scheme for $\boldsymbol{W}^\ell$.

The first term calculates the inner distance sum of $V^*$, where a larger value represents greater inner heterogeneity within the kept grouped kernels. The second term iterates through every pruned grouped kernel $p$, finds the $\gamma$ number of kept grouped kernels that are closest to each $p$ according to the distance matrix $D(\boldsymbol{g}^\ell)$, then sums over the distance between each $p$ to its corresponding $\gamma$ kept grouped kernels. A pruning strategy with a smaller second term has better homogeneity between kept and pruned grouped kernels. Since a strategy with greater inner heterogeneity and smaller outer homogeneity is desired, we further introduce a tunable parameter $-\beta$ to balance two terms and define the best the pruning strategy of $\boldsymbol{g}^\ell$ to be $V_{\text{best}}^* = \arg\max_{V^*}(\text{Score}(V^*, \boldsymbol{g}^\ell))$ for all $V^*$s available in $PS$. **We have also implemented a toy experiment to verify if our grouped kernel pruning method may obtain the optimal or a close-to-optimal solution, please refer to A.1.2 for details. Ablation studies conducted at A.4.5 also confirm the efficacy of our greedy algorithm.**

### 3.4 Layer reconstruction to grouped convolution architecture

The layer reconstruction of our method happens at two places: 1) Given a convolutional layer $\boldsymbol{W}^\ell$, we permute the output channels of such layer according to the filter clustering result. This makes filters within $\boldsymbol{W}^\ell$ form $n$ grouped filters as demonstrated in Figure 1 (a) to (b). 2) After a certain ratio of grouped kernels were pruned from their corresponding filter groups, we permute the input channels of the remaining grouped kernels to remove sparsity from each filter group (Figure 1(c) to (d)). As the result of the above two channel permutations, we will have $n$ dense grouped filters left in the pruned $\boldsymbol{W}^\ell$. Since every grouped filter has its own set of input-output pathways, the pruned $\boldsymbol{W}^\ell$ is essentially a grouped convolution architecture. This architecture enables parallel computing capabilities, because every grouped filter can be deployed to a different device so that all group filters may compute simultaneously. **Please refer to A.2.2 for the general procedure in pseudocode.**

## 4 Experiments and Results

### 4.1 Experiment Settings

**Network Architectures and Datasets**  We evaluate the efficacy of our method on popular networks with various depth and architectures: ResNet-20/32/56/110 with the `BasicBlock` implementation and ResNet-50/101 with the `BottleNeck` implementation (He et al., 2016). For datasets, we choose CIFAR-10 (Krizhevsky, 2009), Tiny-ImageNet (Wu et al., 2017), and ImageNet (ILSVRC-12) (Deng et al., 2009) as they vary in complexity and Wang et al. (2021) has carried out a fairly large-scale comparative experiments of many different structured pruning algorithms running under the same setting — which provides us with a rich background to compare against.

**Compared Methods and Performance Evaluation**  We test our method against various existing pruning methods showed in Table 2. All mentioned pruning methods are evaluated against the following criteria: $\Delta$Acc, $\downarrow$ Params, and $\downarrow$ FLOPs which are defined as the difference of top-1 accuracy, parameters reduction, and FLOPs reduction between the baseline and pruned model (where $x$ represents $x\%$ of parameters/FLOPs were removed/reduced comparing to the original network). In addition, we investigate the pruning procedure and pruning setting of the compared methods in the three following aspects: 1) Does the method require an iterative pruning procedure? 2) Does the method require a special fine-tuning setup that deviates from Section 3.2.2 or Figure 1(a) of Wang et al. (2020)? 3) What is the fine-tuning budget of the method-in-question? We believe these are crucial information for building a fair understanding of the three aforementioned numerical criteria. Note, a method with iterative pruning procedure will automatically require special fine-tuning, as it will fine-tune/retrain after each pruning operation, which is intrinsically different from the procedure introduced in Section 3.2.2. **Our training and pruning settings are fairly simple given our method follows a one-shot procedure, please refer to A.3.1 for details.**

### 4.2 Results and analysis

As demonstrated in Table 2, **our method yields superior accuracy retention than various modern SOTA methods across six tested network-dataset combinations with a commensurate amount of parameters pruned** with the exception of ResNet-56 on Tiny-ImageNet, where our performance is reasonably close ($-0.12\%$ behind) to the best offering of 3D by Wang et al. (2021). Specifically,

our method achieves such results with a simple one-shot pruning procedure and a standard fine-tuning setup. Which are usually considered two less advantageous design as many recent SOTA methods utilize iterative prune-train cycles (Wang et al., 2021; Frankle & Carbin, 2019). Yet some simple tricks like *dynamic pruning rate*, *soft-pruning* (keep the pruned components updatable until very end), and *weight reinitialization* before fine-tuning may often provide most algorithms another performance boost (Li et al., 2017; He et al., 2018; Renda et al., 2020; Liu et al., 2019). We opt not to implement such tricks to provide a cleaner delivery of our method. Last, although not universally observed, our method often requires a smaller fine-tuning budget than other offerings. **Here we include an abbreviated version of our experiment report at Table 1, please refer to Table 2 for the full one.** Please also refer to A.3.2 and A.3.3 for discussions on hyperparameters and speedup.

Table 1: "IT", "SF" respectively indicate if the method-in-question requires an iterative pruning or a special fine-tuning procedure. "FB" is the fine-tuning budget of each method (in terms of # of epochs). "BA" represents the pre-pruned network's accuracy. A cell with "-" implies either such information is inapplicable or we failed to confidently identify such information. Methods noted with $\odot$ are replicated by Wang et al. (2021), the rest are drawn from their original papers. The pruning rates (PR) of our method are adjusted to meet the ↓Params or ↓FLOPs of other methods.

| Method | IT | FB | SF | BA (%) | Δ Acc (%) | ↓ Params (%) | ↓ FLOPs (%) |
|---|---|---|---|---|---|---|---|
| **ResNet-56 on CIFAR-10: FLOPs: 1.27E8  Params: 8.53E5** | | | | | | | |
| DBP$^\odot$ (Wang et al., 2019b) | ✓ | 560 | ✓ | 93.69 | ↓ 0.42 | 40 | 52 |
| FPGM (He et al., 2019) | ✓ | 100 | ✓ | 93.59 | ↓ 0.33 | - | 52 |
| GAL (Lin et al., 2019) | ✗ | 100 | ✗ | 93.26 | ↑ 0.12 | 12 | 38 |
| PScratch (Wang et al., 2020) | ✓ | ≈600 | ✓ | 93.23 | ↓ 0.18 | - | 50 |
| SFP (He et al., 2018) | ✓ | 100 | ✓ | 93.59 | ↓ 1.33 | - | 53 |
| 3D (Wang et al., 2021) | ✓ | 560 | ✓ | 93.69 | ↑ 0.07 | 40 | 50 |
| TMI-GKP (ours, PR = 43.75%) | ✗ | 300 | ✗ | 93.78 | ↑ 0.22 | 43.49 | 43.23 |
| **ResNet-110 on CIFAR-10: FLOPs: 2.55E8  Params: 1.73E6** | | | | | | | |
| DHP (Li et al., 2020) | - | - | ✓ | 94.69 | ↓ 0.06 | 37 | 36 |
| FPGM (He et al., 2019) | ✓ | 100 | ✓ | 93.68 | ↑ 0.05 | - | 52 |
| GAL (Lin et al., 2019) | ✗ | - | ✗ | 93.50 | ↓ 0.76 | 45 | 49 |
| PScratch (Wang et al., 2020) | ✓ | ≈500 | ✓ | 93.49 | ↑ 0.20 | - | 40 |
| SFP (He et al., 2018) | ✓ | 100 | ✓ | 93.68 | ↓ 0.30 | - | 41 |
| TMI-GKP (ours, PR = 43.75%) | ✗ | 300 | ✗ | 94.26 | ↑ **0.64** | 43.52 | 43.31 |
| **ResNet-101 on Tiny-ImageNet: FLOPs: 1.01E10   Params: 4.29E7** | | | | | | | |
| DBP$^\odot$ (Wang et al., 2019b) | ✓ | 420 | ✓ | 64.83 | ↓ 3.48 | 76 | 77 |
| DHP$^\odot$ (Li et al., 2020) | - | 420 | ✓ | 64.83 | ↓ 0.01 | 50 | 75 |
| GAL$^\odot$ (Lin et al., 2019) | ✗ | 420 | ✗ | 64.83 | ↓ 0.50 | 45 | 76 |
| 3D (Wang et al., 2021) | ✓ | 420 | ✓ | 64.83 | ↑ 0.44 | 51 | 75 |
| TMI-GKP (ours, PR = 87.50%) | ✗ | 300 | ✗ | 65.51 | ↑ **1.38** | 43.53 | 43.25 |
| **ResNet-50 on ImageNet (ILSVRC-12):  FLOPs: 5.37E9  Params: 2.56E7** | | | | | | | |
| PScratch (Wang et al., 2020) | ✓ | ≈409 | ✓ | 77.20 | ↓ 0.50 | 29.8 | 26.8 |
| Taylor (Molchanov et al., 2019) | ✓ | 25 | ✓ | 76.18 | ↓ 0.70 | 30.0 | 28.1 |
| ThiNet (Luo et al., 2017) | ✓ | 1 per prune | ✓ | 72.88 | ↓ 0.84 | - | 36.7 |
| FPGM (He et al., 2019) | ✓ | 100 | ✓ | 75.96 | ↓ 0.92 | 33.2 | 33.7 |
| SFP (He et al., 2018) | ✓ | 100 | ✓ | 76.15 | ↓ 1.54 | - | 41.8 |
| TMI-GKP (ours, PR = 75%) | ✗ | 90 | ✗ | 76.15 | ↓ **0.62** | 33.21 | 33.74 |

### 4.3 Ablation study

We anatomize the effectiveness of different components of our proposed method. Specifically, we first demonstrate if the idea of our proposed TMI system and greedy approach are sound at a fundamental level; then we evaluate if they are better than some common alternative approaches with similar functionalities. With the procedure design evaluated, we also provide some real-word runtime experiments of our proposed pruning procedure. Last, we evaluate the relationship between the hyperparameters and the pruned network's accuracy retention. **Please refer to A.4 for details.**

## 5 Conclusion

We proposed a *densely structured* pruning framework capable of yielding beyond SOTA performance with a straightforward one-shot procedure. We believe the power of densely structured kernel pruning may go well beyond our implementation, as a) there will certainly be more — and hopefully better — inductive biases and unsupervised clustering schemes made available, and b) our adaptation of kernel pruning and grouped convolution is rather vanilla, where more sophisticated variants are available to explore; our framework may thrive on these further-discoveries.

In addition, our work serves as a proof of the lottery tickets-induced heuristics can be used to guide a structured pruning strategy. This is an often overlooked usage despite the popularity of LT-related research, possibly due to the high cost of winning tickets searching. In such case, we hope the *multiple (potential) tickets evaluations* trick we introduced in our method may help in levitating such concerns. This might have an impact beyond the field of network pruning, as scholars of other tasks might find the ticket-induced heuristics to be useful but are deterred by the high cost of tickets searching. Where our trick provides a form of approximation by just inspecting the weights shifting log saved during training.

## REPRODUCIBILITY STATEMENT

As we advocate our proposed framework is able to shine a new light on kernel pruning under the context of densely structured pruning, we have prepared a GitHub repository with checkpoints placed on every stage of our method. We hope this will facilitate the reproduction of our work and invite our fellow scholars to research and optimize on different stages of our pruning framework. To reproduce the exact training and pruning settings of our experiments in Table 2, please refer to A.3.

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

## A  APPENDIX

### A.1  SUPPLEMENTARY MATERIAL TO PROPOSED METHOD

#### A.1.1  *Silhouette score* AND ACCURACY RETENTION

We experiment ResNet-32 on CIFAR-10 (BA = $92.82\%$) with K-PCA + Equal-sized $k$-Means and Spectral + Equal-sized $k$-Means. The mean *Silhouette score* of the latter combination is $0.3349/0.0268 = 12.49$ times higher than the previous combination; indicating that if the Silhouette score may capture the better clustering result with respect to accuracy retention, the latter approach should be significantly better. However, the K-PCA approach has a pruned accuracy of $92.68\%$ where the Spectral approach has a pruned accuracy of $92.41\%$, a $+0.27\%$ in favor of the method which the Silhouette score strongly suggests against. This demonstrates that tools like the Silhouette score might not be a good judge of clustering quality in terms of accuracy retention.

#### A.1.2  EFFECTIVENESS OF OUR GREEDY APPROXIMATION ALGORITHM WITH TOY EXAMPLE

To test out whether our greedy approximation algorithm may yield a solution that is close enough to the optimal one, we conducted a trial of TMI-GKP with $\beta = 0$ in Equation 5 on ResNet-32. For the first ten convolutional layers (80 filter groups pruned), all approximated solutions are within $0.5\%$ in terms of the score difference (as defined in Equation 5) to the optimal solution — which implies the effectiveness of our approximation algorithm.

## A.2 Additional Method Procedure Information

### A.2.1 Greedy Grouped Kernel Pruning Procedure

---

**Algorithm 1** Generate $C_{\text{in}}^{\ell}$ grouped kernel pruning strategies

---

    **Input:** Filter group $\boldsymbol{g}^{\ell}$      ▷ one of the $n$ filter group in layer $\boldsymbol{W}^{\ell}$
    **Input:** $s$      ▷ number of grouped kernels to preserve, $s = (1 - \text{pruning rate}) \cdot C_{\text{in}}^{\ell}$
    **Initialize:** Empty list $PS$      ▷ for storing grouped kernels pruning strategies
    **Initialize:** $C_{\text{in}}^{\ell}, D(\boldsymbol{g}^{\ell})$      ▷ number of grouped kernel in $\boldsymbol{g}^{\ell}$ and distance matrix of $\boldsymbol{g}^{\ell}$
1: **for each** $\boldsymbol{K}_i \in \boldsymbol{g}^{\ell}$ **do**
2:     **Initialize:** Empty list $U^*$      ▷ storing preserved grouped kernels per the current strategy
3:     **Initialize:** Zero-filled matrix $M \in \mathbb{R}^{s \times C_{\text{in}}^{\ell}}$      ▷ the cost matrix
4:     $M_{[0,j]} \leftarrow w(\boldsymbol{K}_i, \boldsymbol{K}_j)$ for all $\boldsymbol{K}_j \in \boldsymbol{g}^{\ell} \backslash \boldsymbol{K}_i$
5:     $U^*$.append($\boldsymbol{K}_i$)      ▷ make $\boldsymbol{K}_i$ the first grouped kernel to keep
6:     **for** $\text{row} \leftarrow 1$ to $s$ **do**
7:         $\boldsymbol{K}_{\text{last}} \leftarrow U^*[-1]$
8:         **for each** grouped kernel $\boldsymbol{K} \in \boldsymbol{g}^{\ell}$ but $\boldsymbol{K} \notin U^*$ **do**
9:             $K_{\text{index}} \leftarrow \boldsymbol{g}^{\ell}$.getindex($\boldsymbol{K}$)
10:             $M_{[\text{row}, K_{\text{index}}]} \leftarrow M_{[\text{row}-1, K_{\text{index}}]} + w(\boldsymbol{K}_{\text{last}}, \boldsymbol{K})$
11:         $U^*$.append($\arg\max_{\boldsymbol{K}}(M_{[\text{row}]})$)
12:         **if** $|U^*| == s$ **then**
13:             $V^* \leftarrow U^*$, $PS$.append($V^*$) and **break**
14: **return** $PS$

---

### A.2.2 General Procedure

---

**Algorithm 2** General Procedure of TMI-GKP

---

    **Input:** $\boldsymbol{W}, (k_1, k_2)$      ▷ trained model, tickets window
    **Input:** $CS, PR, n$      ▷ clustering schemes, pruning ratio, number of groups
1: **for each** convolutional layer $\boldsymbol{W}^{\ell} \in \boldsymbol{W}$ **do**
2:     **Initialize:** Empty list $\boldsymbol{G}_{\text{candidates}}^{\ell}$      ▷ To store candidate clustering results
3:     **for** $k_i \in [k_1, k_2]$ **do**
4:         Cluster filters from $\boldsymbol{W}^{\ell}$ into $n$ equal-sized groups according to each scheme in $CS$
5:         Identify the preferred clustering result $\boldsymbol{G}_{k_i}^{\ell} \leftarrow \arg\min_{\boldsymbol{G}^{\ell}}(\underset{\text{filters clustering}}{\text{Score}}(\boldsymbol{G}^{\ell}, k_i))$[Equation 4]
6:         $\boldsymbol{G}_{\text{candidates}}^{\ell}$.append($\boldsymbol{G}_{k_i}^{\ell}$)
7:     Identify $\boldsymbol{G}_{\text{best}}^{\ell} \leftarrow$ the most occurred item in $\boldsymbol{G}_{\text{candidates}}^{\ell}$
8:     Apply $\boldsymbol{G}_{\text{best}}^{\ell}$ on $\boldsymbol{W}^{\ell}$
9:     **for each** filter group $\boldsymbol{g}_i^{\ell} \in \boldsymbol{G}_{\text{best}}^{\ell}$ **do**
10:         Get grouped kernel pruning strategies $V^* \leftarrow$ **Algorithm 1**($\boldsymbol{g}_i^{\ell}$) [A.2.1]
11:         Identify $V_{\text{best}}^* \leftarrow \arg\max_{V^*}(\underset{\text{grouped kernel pruning}}{\text{Score}}(V^*, \boldsymbol{g}_i^{\ell}))$ [Equation 5]
12:         Apply $V_{\text{best}}^*$ on $\boldsymbol{g}_i^{\ell}$
13:     $\boldsymbol{W}^{\ell} \leftarrow$ pruned $\boldsymbol{W}^{\ell}$ reconstructed in grouped convolution architecture
14: **return** $\boldsymbol{W}$

---

### A.2.3 Time Complexity Analysis of the TMI Clustering Procedure

Our pruning procedure is basically two-stage: filter clustering and grouped kernel pruning. Since we have already analyzed the time complexity of the second stage at the end of Section 3.3, we hereby focus on the time complexity of our filter clustering procedure.

Given a clustering result of convolutional layer $\boldsymbol{W}^{\ell}$, we first calculate the TMI score of each filter group defined by Equation 2. The TMI score of a clustering result in regards to an epoch $k$ has a

time complexity of $\mathcal{O}(\boldsymbol{W}^\ell)$ for $\boldsymbol{W}^\ell \in \mathbb{R}^{C_{\text{out}}^\ell \times C_{\text{in}}^\ell \times H^\ell \times W^\ell}$. This is because TMI scores are essentially achieved by finding out the magnitude increase of individual weights in $\boldsymbol{W}^\ell$ from epoch $k$ to epoch $t$ (the final epoch). TMI scores for filters and filters groups are simply sum of the TMI scores of a certain individual weights — where the summing procedure is at most $\mathcal{O}(\boldsymbol{W}^\ell)$ as we only got these many weights to sum.

The TMI score for this clustering result is then determined by Equation 4. Which is a $\mathcal{O}(n)$ procedure as we have $n$ groups (thus $n-1$ intervals). Granted $\mathcal{O}(n) \ll \mathcal{O}(\boldsymbol{W}^\ell)$, this term is negligible and the overall time complexity is still $\mathcal{O}(\boldsymbol{W}^\ell)$

Note we may have multiple clustering results generated by multiple clustering schemes, where we use their TMI scores and Equation 4 to determine the best clustering result for layer $\boldsymbol{W}^\ell$. Thus, we need to multiple the time complexity of a single clustering result to the number of clustering schemes available, we denote this number as $CS_{\text{num}}$. The overall complexity is now $CS_{\text{num}} \cdot \mathcal{O}(\boldsymbol{W}^\ell)$. It is hard to analyze the time complexity of each clustering scheme as it involves various dimension-reduction and clustering combinations. For the ease of expression, we uniformly denote all clustering schemes to have a time complexity of $\mathcal{O}(CS(\boldsymbol{W}^\ell))$ on layer $\boldsymbol{W}^\ell$ (for $CS(\boldsymbol{W}^\ell) > \boldsymbol{W}^\ell$ as a clustering scheme need to read all weights in $\boldsymbol{W}^\ell$ to produce a proper clustering result).

Last, as illustrated in Section 3.2.2 - **Increase robustness with multiple ticket evaluations**, we need to repeat the whole procedure for $k_{\text{num}}$ times for $k_{\text{num}}$ being the number of potential ticket epochs we evaluated. Thus, the final "rough" big-$\mathcal{O}$ complexity for our filter clustering procedure is:

$$CS_{\text{num}} \cdot \mathcal{O}(CS(\boldsymbol{W}^\ell)) + CS_{\text{num}} \cdot k_{\text{num}} \cdot \mathcal{O}(\boldsymbol{W}^\ell). \tag{6}$$

In our experiments, such $k_{\text{num}}$ is usually set to 35 (unless the network is too large) and $CS_{\text{num}}$ is 3 as we have three different clustering schemes available (see Section 3.2.1). By the "absorption" law of big-$\mathcal{O}$ analysis, the theoretical time complexity is only $\mathcal{O}(CS(\boldsymbol{W}^\ell))$ — which is identical to a standard single-shot filter clustering procedure. This implies the theoretical lightweight-ness of our proposed method.

For more information, we included a discussion on how to adjust the pruning procedure to meet a time/computation budget at A.2.4 with real-world runtime experiments available at A.4.7.

### A.2.4 ADJUST THE PRUNING PROCEDURE TO MEET A TIME/COMPUTATION BUDGET

Although the analysis at A.2.3 and experiments in A.4.7 demonstrate our TMI pruning procedure is reasonably fast, we admit that it can be slow if given a wide network to prune or were asked to evaluate many potential ticket epochs — as our algorithm will have to repetitively evaluate the network with different TMI scores over and over again. We hereby provide several points for adjustability of our method.

1. **Reduce the range of ticket window**: If a ticket window is defined to be $[k_1, k_2]$, consider using $[k_1', k_2']$ where $k_1 < k_1'$ and $k_2' < k_2$. So less potential ticket epochs were evaluated.

2. **Add a ticket step**: For a ticket window $[k_1, k_2]$, consider adding a $k_{\text{step}}$ so instead of evaluating $k_1, k_{1+1}, k_{1+1+1}, ...$ we now evaluate $k_1, k_{1+k_{\text{step}}}, k_{1+k_{\text{step}}+k_{\text{step}}}, ....$ By doing this, less potential ticket epochs were evaluated while a wide range of potential ticket epochs from different stage of the network training are still considered.

3. **Relax the granularity of clustering evaluation**: The proposed TMI-GKP determines the optimal clustering scheme at a per layer manner. For CNN models with block-like structure (such as ResNet), one may opt to determine the clustering scheme for one layer of the block, then proceed to use such clustering scheme on the whole block.

4. **Adjust clustering schemes**: One may opt to reduce the number of clustering schemes available for the TMI score evaluation. Or one may opt to use clustering schemes which are less computational demanding. Granted the TMI system is likely to capture the "better" clustering scheme among the options, the method would still function, but likely not at its full potential.

### A.3 Supplementary material to Experiments and Results

#### A.3.1 Training and Pruning Settings

For all experiments done on CIFAR-10 and Tiny-ImageNet, we train the baseline models for 300 epochs with the learning rate starting at $0.1$ and dividing by 10 per every 100 epochs. The baseline model is trained using SGD with a weight-decay set to $5e-4$, momentum set to $0.9$, and a batch-size of 64. All data are augmented with random crop and randomly horizontal flip. For the experiments done on ImageNet, we train the ResNet-50 model for 90 epochs with the weight-decay set to $1e-4$ and learning rate dividing by 10 per every 30 epochs (while keeping all other settings the same as CIFAR-10 and Tiny-ImageNet experiments). Our pruning settings are largely identical to our training settings except for the learning rate, which is set to $0.01$ at the start.

#### A.3.2 Choice of Hyperparameters

There are three main tunable hyperparameters in the mainframe of TMI-GKP, $\alpha$ from Equation 4 and $\beta, \gamma$ from Equation 5. Note in practice $\gamma$ is not a fixed value but rather a value that has a fixed proportion to its the convolutional layer's input channels, i.e., $\gamma = \gamma_{\text{ratio}} \cdot C_{\text{in}}^{\ell}$ for layer $\boldsymbol{W}^{\ell}$. This is because different convolutional layer may have a different $C_{\text{in}}$, so it makes better sense to let $\gamma$ of a layer adjust along with its $C_{\text{in}}$.

For experiments shown in Table 2, we fix $\alpha$ to $0.5$ and $\gamma_{\text{ratio}}$ to $0.2$ for convenience. For experiments of ResNet-56 on CIFAR-10 and ResNet-101 on Tiny-ImageNet, we set $\beta$ to 2. For the experiment of ResNet-32 on CIFAR-10, we set $\beta$ to 1. For the rest of the experiments in Table 2, we set $\beta = \binom{V^*}{2}/(|\boldsymbol{g}^{\ell} \backslash V^*| \cdot \gamma)$ with respect to Equation 5. This basically implies we want the two terms in Equation 5 to evaluate an equal amount of grouped kernel pairs.

To further ensure/demonstrate the reproducibility of our work, in our codebase we will provide a Google Colab notebook that replicates all CIFAR-10 pruning experiments conducted in Table 2. For every completed experiment, our codebase may register a `setting.json`, a `cluster.json` (upon extraction of a pruned model), and an `experiment.log`. Where the `setting.json` includes all hyperparameters' settings, `cluster.json` includes the clustering scheme per each convolutional layer and its resultant permutation matrix for converting the baseline model from (a) to (b) in Figure 1, and `experiment.log` contains the experiment printouts. In addition, we have implemented a set of methods so a fellow researcher may pipeline the `cluster.json` file to the baseline model and try for a different set of pruning strategies. This will save them the work of re-clustering the filters and re-calculating/re-evaluating the TMI scores for every layer.

#### A.3.3 Omitting Speed Up Analysis

We purposefully omitted the speedup analysis between the original and the pruned network mainly due to lack of optimization of grouped convolution on current ML platforms. As an example, on `PyTorch`, a grouped convolution with `groups = 8` is much slower than the standard convolution despite the former one has much less parameters and FLOPs.

Please direct to our GitHub repository where we discuss in length of why `PyTorch` is slowing us done, what's the reason behind it, and empirically show that it is indeed `PyTorch` lacks of optimization that causing this problem. We also discuss why this will not be an issue for long (by presenting ML platforms' commitments of optimizing the speed of grouped convolution), why this is an achievable goal (as scholars have already accelerated grouped convolution on said platforms (Gibson et al., 2020; Qin et al., 2018)), and how this is not likely to affect the serious implementation of our method for its intended purposes — as a group convolution network can be deployed as standard convolution on multiple edge devices in a parallel fashion (Su et al., 2020).

#### A.3.4 Specific experiments requested by reviewers

Upon the requests of two reviewers, we have additionally conducted experiments with our proposed method on VGG-16 (Simonyan & Zisserman, 2014) and on ResNet-56 on CIFAR-10 against pruning method GAL by Lin et al. (2019) (but with a more aggressive pruning rate as $\downarrow$ Params and $\downarrow$ FLOPs are $\approx 60\%$ instead of $\approx 43\%$).

Please refer to our OpenReview entry (former experiment, latter experiment, context of the latter experiment) for details — as these requests should only make sense under their Q&A context, and we have not (or for some literature, we are not able to) conduct the pruning procedure investigations like we did in Table 2 as defined in Section 4.1.

## A.4 ABLATION STUDIES

In this section we will anatomize our method in the following aspects. The general idea is we first show the basic format(s) of our approach works — as if the results achieved by following our approach would be better than the results achieved by going against our approach — then, we show that our approach works better than some common alternative approaches with similar functionalities.

First, we confirm the effectiveness of our TMI scoring system by conducting the following ablation studies:

1. **TMI/MI scores and filter pruning**: where we investigate if filters with higher TMI score or larger magnitude increase will lead to better accuracy retention. This shows our TMI scoring system works at a filter level.

2. **TMI-driven clustering and accuracy retention**: where we compare the TMI-driven clustering results with shuffled clustering results in terms of accuracy retention. This shows our TMI scoring system works in terms of filter clustering.

3. **TMI-driven clustering v. Other filter clustering schemes**: where we investigate if the TMI-driven clustering schemes are better than some common alternative clustering schemes.

4. **TMI-GKP v. Grouped convolution-only**: where we investigate if the performance of our method is from the procedure we proposed, or it is simply due to the adaptation of grouped convolution.

Then, we investigate our greedy grouped kernel pruning algorithm with the following ablation studies:

1. **Greedy-induced pruning and accuracy retention**: where we compare the grouped kernel pruning results induced by our greedy approach to two approaches that take the "inverse" and "complement" of our greedy approach. This shows the basic principle of our greedy approach works.

2. **Greedy-induced pruning v. Other grouped kernel pruning policies**: where we compare the greedy approach against some alternative kernel pruning policies applied on grouped kernel pruning. This shows our greedy approach works better than some common alternative kernel pruning policies.

Last, we provide some experiments to evaluate the real-world runtime of the pruning procedure of TMI-GKP and ablation studies on hyperparameters.

### A.4.1 TMI/MI SCORES AND FILTER PRUNING

We first address the question of whether the TMI score (or the original "MI" magnitude increase score from Zhou et al. (2019)) has a relationship with accuracy retention when relaxed to a filter level. Although the question is straightforward, it is up to different interpretations as the TMI score was proposed as a tool to determine which clustering result is the optimal one — which requires multiple tickets evaluation. Thus, it is hard to determine the TMI scores of a filter, as it is sensitive to the choice of $k$ in Equation 1.

To address such problem, we run Equation 1 on the same set of $k$s as our TMI-GKP algorithm, then we rank (sort) all filters according to their TMI scores under each $k$. Namely, when $k = 35$, we may have filter $A$ to be rank 1, filter $B$ to be rank 2... and for $k = 36$ we may have $B$ to be rank 1 and filter $C$ to be rank 2. We then sum all ranks of each filter across different $k$s, where the filter with smallest sum (highest sum of ranks) is considered the one that is most preferred by the TMI system. We denote this sum the *TMI Filter Ranking Score*.

We opt to use rank-per-each-$k$, but not simply adding TMI scores of the same filter on different $k$s together and rank all filters, because the former approach is a) less sensitive to potential extreme value introduced by a certain $k$ and b) similar to the proposed scoring mechanism defined in Equation 4, which is used in TMI-GKP.

We zero-mask different filters according to the following five criteria. All experiments are conducted on ResNet-32 (baseline accuracy: 92.82%) with pruning rate set to 50% (half of the filters per each layer are zero masked).

1. **TMI preferred**: We kept filters that have the lowest *TMI Filter Ranking Scores*, and zero-masked the rest.

2. **TMI complement**: We kept filters that have the highest *TMI Filter Ranking Scores*, and zero-masked the rest. Since the pruning rate is set to 50%, this is essentially taking the complement of the above scheme.

3. **MI preferred**: We kept filters that have the highest magnitude increase since initialization, and zero-masked the rest.

4. **MI complement**: The complement of above scheme.

5. **Random**: Half of the filters per each layer were randomly zero-masked.

| Scheme | TMI preferred | TMI complement | MI preferred | MI complement | Random |
|---|---|---|---|---|---|
| **Acc. (%)** | **64.48** | 61.93 | 63.25 | 62.06 | 63.92 |

The above experiments clearly demonstrate that both better *TMI filter ranking scores* and larger MI scores are correlated with better accuracy retention. **In addition, the** $1.23\%$ **lead of *TMI preferred* to *MI preferred* scheme also implies that our TMI system — with multiple tickets evaluation — is superior than the original magnitude increase system introduced in Zhou et al. (2019) at the filter level.**

### A.4.2 TMI-DRIVEN CLUSTERING AND ACCURACY RETENTION

Knowing that TMI preferred filters might lead to better accuracy retention. We are interested in learning if the clustering results produced by the TMI system may also lead to better accuracy retention. Thus, we implement the following criterion:

- **TMI shuffled**: Take a clustering result determined by the TMI scoring system ($argmin$ of Equation 5) and shuffle its filters across different groups. By "shuffle," we mean that if a set of filters were originally in the same group, after the shuffle, no two of the above-mentioned filters will be in the same group anymore.

| Model | Baseline (%) | TMI-GKP (%) | TMI shuffled (%) |
|---|---|---|---|
| **ResNet-20** | 92.35 | ↓ **0.34** | ↓ 0.67 |
| **ResNet-32** | 92.82 | ↑ **0.22** | ↓ 0.09 |
| **ResNet-56** | 93.78 | ↑ **0.22** | ↓ 0.05 |
| **ResNet-101** | 94.26 | ↑ **0.64** | ↑ 0.25 |

The experiment results confirmed that our TMI system may provide positive contribution even under the filter clustering context. Although the improvement is not as significant as the one provide by the greedy approach (see A.4.6), the improvement is consistent. Yet without it, our method may not exceed SOTA performance at all.

We further include a discussion in A.4.6 on why empirical evidence suggests the TMI-driven clustering may not guarantee on finding the best clustering in terms of accuracy retention, but rather a robust policy with more comprehensive considerations done to deliver a "better" and very usable solution.

### A.4.3    TMI-DRIVEN CLUSTERING V. OTHER FILTER CLUSTERING SCHEMES

We set the greedy grouped kernel pruning procedure to fixed and feed our method with different schemes on filter clustering: three clustering schemes individually and a uniform random assignment of the three clustering schemes. The results with ResNet-32 on CIFAR-10 suggest the TMI-driven clustering may deliver the best accuracy retention.

| Method | BA (%) | $\Delta$ Acc (%) | $\downarrow$ Params (%) | $\downarrow$ FLOPs(%) |
|---|---|---|---|---|
| **ResNet-32 on CIFAR-10:** | | **FLOPs: 6.95E7** | **Params: 4.29E7** | |
| TMI Clustering w/ Greedy | 92.82 | $\uparrow$ **0.22** | 43.43 | 43.09 |
| K-PCA + $k$-Means w/ Greedy | 92.82 | $\uparrow$ 0.01 | 43.43 | 43.09 |
| Spectral + $k$-Means w/ Greedy | 92.82 | $\downarrow$ 0.16 | 43.43 | 43.09 |
| $t$-SNE + DBSCAN w/ Greedy | 92.82 | $\downarrow$ 0.14 | 43.43 | 43.09 |
| Random Clustering Schemes w/ Greedy | 92.82 | $\uparrow$ 0.03 | 43.43 | 43.09 |

However, by conducting experiments for the A.4.6, we notice that for the combination of ResNet-110 on CIFAR-10, forcing the clustering scheme as *Spectral Clustering* may achieve slightly superior accuracy retention to TMI-GKP ($+0.06\%$). But by running the same clustering scheme on other experiment combinations like ResNet-20 and ResNet-32 on CIFAR-10, spectral clustering induces much lower results ($-0.33\%$ and $-0.16\%$ in comparison to TMI-GKP). This implies the TMI-driven clustering may not guarantee on finding the best clustering in terms of accuracy retention, but rather a robust policy with more comprehensive considerations done to deliver a "better" solution than many other structured pruning methods when combined with our greedy-induced pruning procedure.

We observe similar phenomena on the *K-PCA + $k$-Means* clustering scheme as it only shows slightly lower performance ($-0.21\%$ in comparison to TMI-GKP) on ResNet-32 on CIFAR-10. But the same clustering scheme performs consistently worse than TMI-GKP on ResNet-20/56/100 with CIFAR-10: coming as respectively $-0.38\%$, $-0.29\%$, and $-0.24\%$ to our TMI-GKP method; which implies TMI-GKP is likely more robust on providing a "better" solution across different networks.

### A.4.4    TMI-GKP V. GROUPED CONVOLUTION-ONLY

Since the architecture of grouped convolution itself may induce a reduction of parameters, we are also interested in learning how much it contributes to accuracy retention. We modified the original network as a vanilla grouped convolution, trained it with the same training settings as in A.3.1, and compared it against TMI-GKP. With a comparable amount of parameters and FLOPs reduction, the results achieved with grouped convolution-only are well below our TMI-GKP.

| Method | BA (%) | Pruned Acc (%) | $\downarrow$ Params (%) | $\downarrow$ FLOPs(%) |
|---|---|---|---|---|
| **ResNet-20 on CIFAR-10:** | | **FLOPs: 4.09E7** | **Params: 2.70E5** | |
| Grouped Conv ($n = 2$) | 90.48 | - | 49.0 | 49.5 |
| TMI-GKP | 92.35 | **92.01** | 43.4 | 42.9 |
| **ResNet-32 on CIFAR-10:** | | **FLOPs: 6.95E7** | **Params: 4.64E5** | |
| Grouped Conv ($n = 2$) | 91.75 | - | 49.2 | 49.6 |
| TMI-GKP | 92.82 | **93.04** | 43.4 | 43.1 |
| **ResNet-56 on CIFAR-10:** | | **FLOPs: 1.27E8** | **Params: 8.53E5** | |
| Grouped Conv ($n = 2$) | 92.34 | - | 49.4 | 49.7 |
| TMI-GKP | 93.78 | **94.00** | 43.5 | 43.2 |
| **ResNet-110 on CIFAR-10:** | | **FLOPs: 2.55E8** | **Params: 1.73E6** | |
| Grouped Conv ($n = 2$) | 92.90 | - | 49.5 | 49.7 |
| TMI-GKP | 94.26 | **94.90** | 43.5 | 43.3 |

### A.4.5    GREEDY-INDUCED PRUNING AND ACCURACY RETENTION

Our greedy grouped kernel pruning approach is developed on the assumption of *(grouped) kernels that are most distinctive from each other are better*. We now put this assumption to test with the following two schemes:

1. **Greedy complement**: where we keep the group kernels that was originally pruned in TMI-GKP.

2. **Greedy reverse**: where we flip the $\arg\max$ on `line 11` of Algorithm A.2.1 to $\arg\min$ with $\beta = 0$ in Equation 5. This means the algorithm is now searching for the next grouped kernel that is "most similar" to the selected grouped kernels.

Here are the experiment results with pruning rate set to $43.75\%$:

| Model | Baseline (%) | TMI-GKP (%) | Greedy complement (%) | Greedy reverse (%) |
|---|---|---|---|---|
| **ResNet-20** | 92.35 | $\downarrow$ **0.34** | $\downarrow$ 1.50 | $\downarrow$ 1.38 |
| **ResNet-32** | 92.82 | $\uparrow$ **0.22** | $\downarrow$ 1.38 | $\downarrow$ 0.56 |
| **ResNet-56** | 93.78 | $\uparrow$ **0.22** | $\downarrow$ 0.88 | $\downarrow$ 0.54 |
| **ResNet-110** | 94.26 | $\uparrow$ **0.64** | $\downarrow$ 0.76 | $\downarrow$ 0.67 |

The experiment results once again demonstrate the significance of our greedy approach. In addition, we observe *Greedy reverse* to have better accuracy retention to *Greedy complement*. We believe this is because the pruning strategy produced by *Greedy reverse* may have overlaps with the one produced by TMI-GKP; yet the pruning strategy produced *Greedy complement* is mutually exclusive with the one produced by TMI-GKP. This indirectly suggests the grouped kernels selected by our greedy approach are certainly "the better" ones, as only a partial overlap with the TMI-GKP's pruning strategy may lead to noticeably better accuracy retention.

Upon request, we further investigated **Greedy reverse with** $\beta = 1$ on ResNet-20/32/56/110 on CIFAR-10, where such scheme is identical to **Greedy reverse** with the exception of setting $\beta$ to 1. This means the algorithm is still searching for the next grouped kernel that is "most similar" to the selected grouped kernels. But among the $C_{\text{in}}^{\ell}$ candidate pruning strategies, it will pick the one with best outer homogeneity (pruned and kept filters are most similar). The results are $\downarrow 1.52\%$, $\downarrow 0.42\%$, $\downarrow 0.45\%$, and $\downarrow 0.61\%$ respectively to the baseline of four aforementioned networks.

This is in line with our anticipation. As per the design of Algorithm 2 and Equation 5, the set of grouped kernels preserved by greedy reverse with $\beta = 1$ should be very similar to the set group kernels preserved by greedy reverse with $\beta = 0$. This is because both of them will preserve a set of grouped kernels that are very similar to each other; and as 56.25% of grouped kernels per layer are preserved, the two policies might very likely end up on the same (or similar) set of grouped kernels. Empirical evidence supports this hypothesis, as the performance differences between the two policies are marginal.

### A.4.6 GREEDY-INDUCED PRUNING V. OTHER GROUPED KERNEL PRUNING POLICIES

We set the filter clustering scheme to fix as *Spectral Clustering* and feed our method with different commonly applied grouped kernel pruning policies ($L^2$ and center pruning). Experiments of ResNet-32/110 on CIFAR-10 have confirmed that our greedy-induced pruning scheme may yield better accuracy retention in a controlled setting.

| Method | BA (%) | $\Delta$ Acc (%) | $\downarrow$ Params (%) | $\downarrow$ FLOPs(%) |
|---|---|---|---|---|
| **ResNet-32 on CIFAR-10:** | **FLOPs: 6.95E7** | **Params: 4.29E7** | | |
| Greedy-induced Pruning | 92.82 | $\uparrow$ **0.06** | 43.4 | 43.1 |
| $L^2$ Pruning | 92.82 | $\downarrow$ 0.30 | 43.4 | 43.1 |
| Center Pruning | 92.82 | $\downarrow$ 0.19 | 43.4 | 43.1 |
| **ResNet-110 on CIFAR-10:** | **FLOPs: 2.55E8** | **Params: 1.73E6** | | |
| Greedy-induced Pruning | 94.26 | $\uparrow$ **0.70** | 43.3 | 43.5 |
| $L^2$ Pruning | 93.76 | $\downarrow$ 0.08 | 43.3 | 43.5 |
| Center Pruning | 93.76 | $\downarrow$ 0.05 | 43.3 | 43.5 |

### A.4.7 RUNTIME ANALYSIS OF TMI-GKP

The following experiments are conducted on a 2.00GHz 4 core Intel Xeon CPU and Tesla V100. Evaluated on 35 potential ticket epochs.

In our code implementation, we first cluster a layer, then prune it, then move on to the next layer. So the runtime is a mixed product of both the TMI filter clustering procedure and the greedy grouped kernel pruning procedure. Please refer to the table below for the runtime of the pruning procedure of our TMI-GKP method. All experiments are against the CIFAR-10 dataset.

| Model | ResNet-20 | ResNet-32 | ResNet-56 | ResNet-101 |
|---|---|---|---|---|
| **Clustering and Pruning Runtime** | 2,977 sec (49.62 min) | 4,741 sec (78.18 min) | 10,376 sec (172.93 min) | 19,818 sec (330.3 min) |

We also separately analyze the runtime of our greedy grouped kernel pruning procedure. This is done by assigning a pre-determined permutation matrix to the network (same effect as clustering, as it permutes a convolutional layer from Figure 1(a) to Figure 1(b)), so the actual greedy-only approach will be even slightly faster). Also note the runtime of this greedy procedure is theoretically related to the choice of $\gamma$ in Equation 5, but the greedy procedure itself is so fast to the point the value of $\gamma$ does not matter anymore.

| Model | ResNet-20 | ResNet-32 | ResNet-56 | ResNet-101 |
|---|---|---|---|---|
| **Greedy Pruning Runtime** | 47 sec | 16 sec | 250 sec (4.17 min) | 2,205 sec (36.75 min) |

We consider this sort of runtime is totally tolerable as an overhead. When compared to methods involve iterative prune-train cycles (Wang et al., 2021), custom loss function (Wu et al., 2018), or feature maps analysis (Yu et al., 2017), our method is significantly more efficient and applicable to a broader set of pre-trained networks.

### A.4.8 EFFECTS OF DIFFERENT HYPERPARAMETERS CHOICES

As mentioned in A.3.2, the two hyperparameters we tuned are $\beta$ and $\gamma$ (derived from $\gamma_{\text{ratio}}$) in Equation 5. We hereby provide a set of experiments to show how different choices of such two parameters will affect the accuracy retention of the pruned network. All experiments were done on ResNet-32 on CIFAR-10 with pruning rate set to $43.75\%$ (baseline: $92.82\%$). Note the experiments below were done with setting one parameter fixed and adjusting the other, so the two tables should be inspect in a collective manner.

We first fixed $\gamma_{\text{ratio}}$ to $0.2$ — namely, for every pruned grouped kernel, $20\%$ of all kept grouped kernels which are most similar to such pruned grouped kernel were evaluated — and try with different $\beta$ settings. The term "auto" implies $\beta = \binom{V^*}{2}/(|\boldsymbol{g}^\ell \backslash V^*| \cdot \gamma)$ with respect to Equation 5, please refer to A.3.2 on how we inferred this value.

| Fixed / Tuned param | $\beta = 0$ | $\beta = 1$ | $\beta = 2$ | $\beta = 4$ | $\beta = 6$ | $\beta = 8$ | $\beta = $ auto |
|---|---|---|---|---|---|---|---|
| $\gamma_{\text{ratio}} = 0.2$ | 92.71 | **93.04** | 92.58 | 92.67 | 92.8 | 92.76 | 92.74 |

The experiment results suggest a relatively more "balanced" relationship between the two terms in Equation 5 may lead to better accuracy retention. This implies both the inner heterogeneity of kept grouped kernels and outer homogeneity between kept and pruned grouped kernels should be taken into consideration — which is exactly how Equation 5 was designed.

We then fixed $\beta$ to "auto" and try with different $\gamma_{\text{ratio}}$ settings:

| Fixed / Tuned param | $\gamma_{\text{ratio}} = 0.1$ | $\gamma_{\text{ratio}} = 0.2$ | $\gamma_{\text{ratio}} = 0.4$ | $\gamma_{\text{ratio}} = 0.6$ | $\gamma_{\text{ratio}} = 0.8$ | $\gamma_{\text{ratio}} = 1$ |
|---|---|---|---|---|---|---|
| $\beta = $ auto | **92.97** | 92.74 | 92.7 | 92.65 | 92.74 | 92.53 |

The experiment results suggest $\gamma_{\text{ratio}}$ should be a relatively small value as increasing its value may lead to worse accuracy retention of the pruned network. This is a rather intuitive result, as when $\gamma_{\text{ratio}} = 1$ every pruned grouped kernel is evaluated against all kept grouped kernels. In such case, the second term of Equation 5 can no longer reveal if there are some kept grouped kernels that

are similar to a pruned one, because all kept kernels are evaluated, yet all kept kernels are already distinctive from each others due to the selection procedure introduced in A.2.1.

Additionally, we have conducted/disclosed more experiments with specific settings required by one reviewer (e.g., with $\beta =$ and $\gamma_{\text{ratio}} = 0.1$ on ResNets; choice of $\beta$ being 2 verses auto). Please refer to our OpenReview entry for more details — as these specifically requested experiments should only make sense under the Q&A context, and they do not fit well to the structure of our ablation studies.

## A.5 DIMENSIONALITY REDUCTION AND CLUSTERING SCHEMES

### A.5.1 KERNEL PRINCIPAL COMPONENT ANALYSIS (K-PCA)

Kernel PCA is considered an improved version of PCA. The choice of kernel functions will significantly affect the experimental results. Global kernels (e.g. polynomial, Sigmoid) and local kernels (e.g. RBF, Laplacian) are the most commonly used kernel functions for K-PCA. They capture different characteristics of the data: the former one demonstrates better extrapolation abilities, and the latter one has better interpolation abilities (Jordaan, 2004). Since the filters are usually not linearly separable and often there are fewer filters than the number of features, we will mix the global and local kernel functions to benefit both extrapolation and interpolation properties. Specifically, we mixed a polynomial and an RBF kernel. We use a parameter $\lambda$ to control the balance of these two functions.

$$
\begin{aligned}
K_{\text{new}}(x, y) &= \lambda K_{\text{Poly}}(x, y + (1 - \lambda) K_{\text{RBF}}(x, y) \\
&= \lambda(ax^{\text{T}} \cdot y + c)^d + (1 - \lambda) \exp\left(-\frac{\| x - y \|^2}{2\sigma^2}\right),
\end{aligned}
\tag{7}
$$

where $\lambda \in [0,\ 1)$, parameters $a$ and $c$ in polynomial kernel have been fixed to 1 and 0. $d = 2$ is the degree of polynomial kernel and $\sigma$ in RBF kernel is the reciprocal of the width of the radial basis function. In practice, such $\lambda$ is set fixed to $0.5$.

### A.5.2 SPECTRAL CLUSTERING

Spectral Clustering is yet another maturely studied and widely adopted clustering method. Unlike in K-PCA, for this method we simply utilize the implementation of Von Luxburg (2007). Specifically in TMI-GKP, we utilize the variation of *mutual-KNN* as the similarity graph and cosine distance as the adjacency matrix based on empirical observations. We rely on Yikun Zhang's version of implementation.

### A.5.3 EQUAL-SIZED $k$-MEANS

To ensure we may have $n$ equal-sized groups out of each filter group, we utilize the default *Same-size k-Means Variation* offered by Schubert & Zimek (2019). Specifically, we rely on Nathan Danielsen's version of implementation.

### A.5.4 $t$-SNE AND DBSCAN

We explore the option of $t$-SNE and DBSCAN because we want to add a density-based clustering scheme from the two distance-based offerings. For implementation details: we set the `perplexity` of $t$-SNE to be the size of the filter group (namely, $C_{\text{in}}^\ell/n$ in layer $\boldsymbol{W}^\ell$) and iterate through different combinations of `n_components`, $\epsilon$, and `min_samples` until the DBSCAN algorithm may yield a clustering result of $\geq n + 1$ groups (for `n_components` $\in [2, n)$). After that, we execute the same data-point reassignment procedure as listed in Schubert & Zimek (2019) until $n$ equal-sized groups are achieved.

### A.6 ACKNOWLEDGEMENT

We would like to acknowledge and give our warmest thanks to John Mays for his timely help on analytical writing.

Table 2: Full comparisons of different pruning methods with ResNet-20/32/56/110 on CIFAR-10, ResNet-56/101 on Tiny-ImageNet, and ResNet-50 on ImageNet (ILSVRC-12). "IT", "SF" respectively indicate if the method-in-question requires an iterative pruning or a special fine-tuning procedure. "FB" is the fine-tuning budget of each method (in terms of # of epochs). "BA" represents the pre-pruned network's accuracy. A cell with "-" implies either such information is inapplicable or we failed to confidently identify such information. Methods noted with $\odot$ are replicated by Wang et al. (2021), the rest are drawn from their original papers. The pruning rates (PR) of our method are adjusted to meet the ↓Params or ↓FLOPs of other methods.

| Method | IT | FB | SF | BA (%) | Δ Acc (%) | ↓ Params (%) | ↓ FLOPs(%) |
|---|---|---|---|---|---|---|---|
| **ResNet-20 on CIFAR-10:** | | **FLOPs: 4.09E7** | | | **Params: 2.70E5** | | |
| DHP (Li et al., 2020) | - | - | ✓ | 92.54 | ↓1.00 | 44 | 48 |
| FPGM (He et al., 2019) | ✓ | 100 | ✓ | 92.20 | ↓1.11 | - | 42 |
| PScratch (Wang et al., 2020) | ✓ | ≈600 | ✓ | 91.75 | ↓1.20 | - | 50 |
| Rethink (Liu et al., 2019) | ✓ | 40 | ✓ | 92.41 | ↓1.34 | - | 40 |
| SFP (He et al., 2018) | ✓ | 100 | ✓ | 92.20 | ↓1.37 | - | 42 |
| TMI-GKP (ours, PR = 43.75%) | ✗ | 300 | ✗ | 92.35 | ↓**0.34** | 43.35 | 42.87 |
| **ResNet-32 on CIFAR-10:** | | **FLOPs: 6.95E7** | | | **Params: 4.64E5** | | |
| DBP$^\odot$ (Wang et al., 2019b) | ✓ | 560 | ✓ | 93.18 | ↓0.53 | 28 | 48 |
| FPGM (He et al., 2019) | ✓ | 100 | ✓ | 92.63 | ↓0.32 | - | 42 |
| GAL$^\odot$ (Lin et al., 2019) | ✗ | 560 | ✗ | 93.18 | ↓1.46 | 39 | 50 |
| PScratch$^\odot$ (Wang et al., 2020) | ✓ | 560 | ✓ | 93.18 | ↓1.00 | - | 50 |
| SFP (He et al., 2018) | ✓ | 100 | ✓ | 92.63 | ↓0.55 | - | 42 |
| 3D (Wang et al., 2021) | ✓ | 560 | ✓ | 93.18 | ↑0.09 | 38 | 49 |
| TMI-GKP (ours, PR = 43.75%) | ✗ | 300 | ✗ | 92.82 | ↑**0.22** | 43.43 | 43.09 |
| **ResNet-56 on CIFAR-10:** | | **FLOPs: 1.27E8** | | | **Params: 8.53E5** | | |
| CP (He et al., 2017) | - | 20 | ✗ | 92.80 | ↓1.90 | - | 50 |
| DBP$^\odot$ (Wang et al., 2019b) | ✓ | 560 | ✓ | 93.69 | ↓0.42 | 40 | 52 |
| DHP$^\odot$ (Li et al., 2020) | - | 560 | ✓ | 93.65 | ↓0.07 | 42 | 49 |
| FPGM (He et al., 2019) | ✓ | 100 | ✓ | 93.59 | ↓0.33 | - | 52 |
| GAL (Lin et al., 2019) | ✗ | - | ✗ | 93.26 | ↑0.12 | 12 | 38 |
| HRank (Lin et al., 2020) | - | 30 per layer | - | 93.26 | ↓0.09 | 42 | 50 |
| PScratch (Wang et al., 2020) | ✓ | ≈600 | ✓ | 93.23 | ↓0.18 | - | 50 |
| Rethink (Liu et al., 2019) | ✓ | 40 | ✓ | 93.80 | ↓0.73 | - | 50 |
| SFP (He et al., 2018) | ✓ | 100 | ✓ | 93.59 | ↓1.33 | - | 53 |
| 3D (Wang et al., 2021) | ✓ | 560 | ✓ | 93.69 | ↑0.07 | 40 | 50 |
| TMI-GKP (ours, PR = 43.75%) | ✗ | 300 | ✗ | 93.78 | ↑**0.22** | 43.49 | 43.23 |
| **ResNet-110 on CIFAR-10:** | | **FLOPs: 2.55E8** | | | **Params: 1.73E6** | | |
| CP (He et al., 2017) | - | 20 | ✗ | 92.80 | ↓1.90 | - | 50 |
| DHP (Li et al., 2020) | - | - | ✓ | 94.69 | ↓0.06 | 37 | 36 |
| FPGM (He et al., 2019) | ✓ | 100 | ✓ | 93.68 | ↑0.05 | - | 52 |
| GAL (Lin et al., 2019) | ✗ | - | ✗ | 93.50 | ↓0.76 | 45 | 49 |
| PFEC (Li et al., 2017) | ✗ | - | ✗ | 93.53 | ↓0.61 | - | 39 |
| PScratch (Wang et al., 2020) | ✓ | ≈500 | ✓ | 93.49 | ↑0.20 | - | 40 |
| Rethink (Liu et al., 2019) | ✓ | 40 | ✓ | 93.77 | ↑0.15 | - | 40 |
| SFP (He et al., 2018) | ✓ | 100 | ✓ | 93.68 | ↓0.30 | - | 41 |
| TMI-GKP (ours, PR = 43.75%) | ✗ | 300 | ✗ | 94.26 | ↑**0.64** | 43.52 | 43.31 |
| **ResNet-56 on Tiny-ImageNet:** | | **FLOPs: 5.06E8** | | | **Params: 8.65E5** | | |
| DBP$^\odot$ (Wang et al., 2019b) | ✓ | 420 | ✓ | 56.55 | ↓0.98 | 25 | 53 |
| DHP$^\odot$ (Li et al., 2020) | - | 420 | ✓ | 56.55 | ↓0.73 | 46 | 55 |
| GAL$^\odot$ (Lin et al., 2019) | ✗ | 420 | ✗ | 56.55 | ↓0.68 | 32 | 52 |
| 3D (Wang et al., 2021) | ✓ | 420 | ✓ | 56.55 | ↓**0.51** | 34 | 59 |
| TMI-GKP (ours, PR = 37.25%) | ✗ | 300 | ✗ | 55.59 | ↓0.63 | 36.74 | 37.05 |
| **ResNet-101 on Tiny-ImageNet:** | | **FLOPs: 1.01E10** | | | **Params: 4.29E7** | | |
| DBP$^\odot$ (Wang et al., 2019b) | ✓ | 420 | ✓ | 64.83 | ↓3.48 | 76 | 77 |
| DHP$^\odot$ (Li et al., 2020) | - | 420 | ✓ | 64.83 | ↓0.01 | 50 | 75 |
| GAL$^\odot$ (Lin et al., 2019) | ✗ | 420 | ✗ | 64.83 | ↓0.50 | 45 | 76 |
| 3D (Wang et al., 2021) | ✓ | 420 | ✓ | 64.83 | ↑0.44 | 51 | 75 |
| TMI-GKP (ours, PR = 87.50%) | ✗ | 300 | ✗ | 65.51 | ↑**1.38** | 43.53 | 43.25 |
| **ResNet-50 on ImageNet (ILSVRC-12):** | | | | **FLOPs: 5.37E9** | **Params: 2.56E7** | | |
| FPGM (He et al., 2019) | ✓ | 100 | ✓ | 75.96 | ↓**0.19** | 22.1 | 22.5 |
| PScratch (Wang et al., 2020) | ✓ | ≈409 | ✓ | 77.20 | ↓0.50 | 29.8 | 26.8 |
| TMI-GKP (ours, PR = 50%) | ✗ | 90 | ✗ | 76.15 | ↓**0.19** | 22.1 | 22.5 |
| Taylor (Molchanov et al., 2019) | ✓ | 25 | ✓ | 76.18 | ↓0.70 | 30.0 | 28.1 |
| ThiNet (Luo et al., 2017) | ✓ | 1 per prune | ✓ | 72.88 | ↓0.84 | - | 36.7 |
| FPGM (He et al., 2019) | ✓ | 100 | ✓ | 75.96 | ↓0.92 | 33.2 | 33.7 |
| SFP (He et al., 2018) | ✓ | 100 | ✓ | 76.15 | ↓1.54 | - | 41.8 |
| TMI-GKP (ours, PR = 75%) | ✗ | 90 | ✗ | 76.15 | ↓**0.62** | 33.21 | 33.74 |

