# OpenReview forum: "Revisit Kernel Pruning with Lottery Regulated Grouped Convolutions"
_ICLR.cc/2022/Conference — ICLR 2022 Poster_

### Official Review · Reviewer_4Wb3 · 2021-10-19

**Correctness:** 3
**Technical Novelty And Significance:** 3
**Empirical Novelty And Significance:** 3
**Recommendation:** 5
**Confidence:** 4

**Main Review:**

pros.
1. The proposed method seems simple yet effective, though it has not been fully evaluated on the large dataset.
2. Figure 1 clearly illustrates the implementation of TMI-GKP. We may easily deploy it on general-purpose devices like CPU/GPU, which benefits from fine-grained and coarse network pruning schemes simultaneously.
3. The technique part of this paper is easy to follow. The combination of the Lottery Ticket Hypothesis and structured pruning seems interesting. This paper may shed some light on this new direction.

cons.
1. The main concern is the experiment part of this manuscript. Since post-training grouped pruning has been frequently discussed in the recent literature (in static mode or dynamic mode) [1,2,3,4,5], a direct comparison with those methods is more appropriate. Besides, ImageNet has been a common setting in recent works. Though I fully understand the authors are stuck with limited computing resources, a solid result on ImageNet can be more convincing than CIFAR-10.
2. I expect the real speedup reported in Table 1 to verify the superiority of learned group convolutions.
3. It would be better to include the real time cost of the "Ticket Magnitude Increase scoring system" and "Qualifying cluster results with TMI scores". It seems that the computing complexity of those steps are $\mathcal{O}(\prod_{i=0}^{K-1} \binom{C_{out}-i\cdot\frac{C_{out}}{K}}{\frac{C_{out}}{K}})$ where $C_{out}$ is the number of filters, $K$ is the number of groups, which is still very large for $C_{out}=512$.
4. I have checked section A.3.1. The proposed "TMI Clustering w/Greedy" with $+0.14$% on CIFAR-10 is somewhat incremental when compared with K-PCA+k-means.
5. It would be better to verify the effectiveness of Algorithm 1 via a toy experiment, such as setting the channel number to $16$. The brute force solution to this problem can be obtained and serves as a baseline.
6. Since the authors apply the Lottery Ticket Hypothesis to structured pruning, I expect an ablation study under the same setting as *winning ticket* or *weights rewinding* to prove the existence of the ticket. It is hard to evaluate the correlation between $TMI$ and the Lottery Ticket Hypothesis, given the existing experiments.
7. According to Table 1, GAL seems to outperform TMI with $12$% Params and $38$% FLOPs?
8. There seems to be no ablation study on $\beta$ (in Eq.(5)). Did I miss anything?

ref:
* [1] Dynamic Group Convolution for Accelerating Convolutional Neural Networks. ECCV2020
* [2] Fully Learnable Group Convolution for Acceleration of Deep Neural Networks. CVPR2019
* [3] Building Efficient Deep Neural Networks with Unitary Group Convolutions. CVPR2019
* [4] Differentiable Learning-to-Group Channels via Groupable Convolutional Neural Networks. ICCV2019
* [5] Efficient Structured Pruning and Architecture Searching for Group Convolution. ICCVw2019


**Summary Of The Paper:**

The authors present a new metric to determine the similarity between different grouped kernels and prune the unimportant $k\times k$ slices out of a 3D filter. They utilize the Lottery Ticket Hypothesis and propose a greedy search strategy to overcome the challenge of a huge search space. The experiment results show that the one-shot scheme can still be comparable to two-stage leading methods on the CIFAR-10 dataset, with a slightly lower training cost. The empirical success of this paper may serve as proof of the existence of the Lottery Ticket Hypothesis.

**Summary Of The Review:**

In general, I believe this work is somehow different from existing works but the current draft makes it challenging for these ideas to reach their full potential. The authors are encouraged to address the weaknesses above.

---

> ### Author Response · Authors · 2021-11-19
> **Addressing Cons 1.**
>
> We thanks this reviewer for the positive recognition, detailed reads, and suggestions. We found many of the proposed *weaknesses* by this reviewer to be reasonable and we have made adjustments/improvements to our manuscript accordingly.
>
> ## Cons 1.1. Lack of ImageNet Experiment
>
> We have now added result on ImageNet and we are **happy to report our method maintained the best accuracy retention among all comparable methods**. In addition, we have also solved the baseline issue of Tiny-ImageNet experiments and conducted two fair experiments on ResNet-56/101, and we are once again happy to report our method still maintains great accuracy retention across the board with an almost 1% lead on ResNet 101 + TinyImageNet in terms of $\Delta$ Acc.
>
> Please see [Section 1 of public comment To All Reviewers: Addressing Common Concerns](https://openreview.net/forum?id=LdEhiMG9WLO&noteId=hLoG26CMAuw) for details.
>
> (Thank you so much for being such a caring person on requesting ImageNet!)
>
> ## Cons 1.2. Comparing against post-train grouped convolution methods
>
> We'd agree that comparing to post-train group convolution methods can be interesting, but **it seems not all works suggested by the reviewer fit this definition.**
>
> * [1, 3] are proposing architecture modifications. Where [1] shall dynamically allocate input channels for a grouped convolution upon the saliency maps of input (see Figure 2 and its abstract "we propose dynamic group convolution that adaptively selects which part of input channels to be connected within each group for individual samples on the fly"). And [3] is adding unitary transforms in group convolution architecture to form "UGConvs" (see Figure 1), which is later trained from scratch (see Evaluation of Different Transforms, where the authors talk about the training setup). The results of [1, 3] are also incomparable to ours experiments as there is no information regarding params and FLOPs reduction.
> * [2] is learning a standard ConvNet architecture into grouped convolutions. This is showed in Figure 1(d) and its abstract "our proposed method automatically learns the group structure in the training stage in a fully end-to-end manner." So it is not post-train.
> * [4] is focusing on learning the optimal number of groups. It is also trained-from-scratch as showed in Figure 1(d) and its caption "the grouping strategy of DGConv is learned end-to-end together with the network parameters, so the group number and connection location are changing dynamically." The results are also incomparable to ours experiments as there is no information regarding params and FLOPs reduction.
>
> Thus, although it is insightful and interesting to compare to "pumped" grouped convolution methods, we can't do it for the above reasons. We do, however, wanted to include [2] in our A.4.4, where we compare our proposed method with the vanilla trained-from-scratch grouped convolution. But it seems [2] has no comparable experiment with us at all as it does not include any BasicBlock ResNet on CIFAR-10, nor does it include any ResNet on ImageNet.
>
> [5] is indeed a post-train pruning literature. But it seriously lacks comparable experiments to modern pruning arts. The only comparable experiment between ours and [5] is ResNet-50 on ImageNet. But the [5]'s ResNet-50 has 46% params pruned where our ResNet-50 on ImageNet (in Table) was pruned with 22% and 33% params drop to accommodate other pruning methods for comparison. So although [5] has a pruned accuracy of 74.1% and ours has 75.53% (+1.52% in favor of us), it is not a fair comparison.
>
> In fact, we do have experiments running on ResNet-50 + ImageNet with 44% params pruned — which is comparable to [5] — but we won't be able to include it by the 11/22 deadline. However, **at around 2/3 of our fine-tuning budget, our method already had a test accuracy of $75.0$% ($+0.9$% to [5]).** This demonstrate the clear superiority our method as we still have 1/3 of our fine-tuning budget to spend.

---

> ### Author Response · Authors · 2021-11-19
> **Addressing Cons 2 to 5**
>
>
> ## Cons 2: Lack of speed up comparison
>
> This is the part we have to unfortunately say no to a reasonable request. We did not, and will continue not to, provide a speedup comparison because of the following two reasons. 1) Such metric is hardly reliable for comparison without an identical environment setup; 2) **ML platforms like pytorch seriously lack optimization on grouped convolution, making the runtime analysis unreliable.**
>
>
> Please see [Section 5 of public comment To All Reviewers: Addressing Common Concerns](https://openreview.net/forum?id=LdEhiMG9WLO&noteId=xDU9Wlg29c4) for details. **Where we discuss in length of why torch is slowing us done, what's the reason behind it, and empirically show that it is indeed torch's fault. We also discussed why this won't be an issue for long, and how this likely won't affect the implementation of our method for its intended purposes.**
>
> We hope the reviewer may understand why we opt to omit speedup analysis for now.
>
> ## Cons 3: Time complexity of TMI-driven clustering
>
> **The procedure of TMI-GKP is in fact very lightweight**. The TMI-driven clustering procedure has the same big-O time complexity as a single one-shot filter clustering method.
>
> **Real-world runtime experiments also confirm our inference**: on ResNet-56, the whole TMI-GKP procedure costs less than 180 mins, and the greedy pruning procedure costs less than 5 mins.
>
> Please see [Section 4 of public comment To All Reviewers: Addressing Common Concerns](https://openreview.net/forum?id=LdEhiMG9WLO&noteId=9XqbfCJw5q5) for details. Where we explain the big-O analysis and provide real-word runtime experiments on different networks. We also reflected why multiple reviewers suspect our procedure to be computational-intensive while it is not, and taken actions to improve our presentation.
>
>
> **In particular, the reviewer's big O analysis does not reflect the procedure of our TMI-driven clustering result.** Because the reviewer's assume we would exhaustively iterate through all possible clustering results and evaluate all of them with the TMI system. But as defined in Algorithm 2, our method simply use the TMI scoring system to evaluate clustering results proposed by the three clustering schemes introduced in Section 3.2.1. Although one selling point of our method is one may add or remove clustering schemes to our framework, the number of clustering schemes available would always be a (relatively small) constant. We'd totally agree with the reviewer that if our algorithm has the time complexity proposed by the reviewer, it will be totally useless — luckily it is much more faster than that.
>
> ## Cons 4: Marginal lead to K-PCA + k-Means in A.4.3.
>
> **This is true, but it is more of a one-task luck**. We have then ran such K-PCA + k-Means combination on ResNet-20/56/100, and it is -0.33%, -0.21%, and -0.25% lower to the TMI-driven approach respectively.
>
> Although the improvement is not as significant as the one provide by the greedy approach (which we will discuss below), the improvement is consistent. Yet without it, our method may not exceed SOTA performance at all.
>
> We further include a discussion in A.4.3 on why empirical evidences suggest the TMI-driven clustering may not guarantee finding the best clustering in terms of accuracy retention, but rather a robust policy with more comprehensive considerations done to deliver a "better" and very usable solution.
>
> ## Cons 5: Toy experiment on the greedy pruning algorithm, and why it is effective.
>
> **What a coincidence! We actually included the exact toy experiment you suggested in A.1.2 of our initial submission**: where we conducted a toy experiment on the first ten convolutional layers (all with 16 in/out channels) on ResNet-32, all greedy approximated solutions are within 0.5% in terms of the cost difference to the optimal solution as defined in Eq (5). We discussed about it in A.1.2 but didn't mentioned it in the main text of our initial submission, now we have refer it and added a short description.
>
> In addition, added ablation studies in A.4.5 confirm the grouped kernels selected by this greedy approach are indeed important, as going against the approach would lead to at most 1.52% of accuracy drop. Also, added ablation studies in A.4.8 support our design of greedy algorithm and confirm our understand of how the scoring system should behave with regard to accuracy retention.

---

> ### Author Response · Authors · 2021-11-19
> **Addressing Cons 6 to 8 & Advocacy for (further) impact**
>
> ## Cons 6: Existence of ticket under structured pruning  (and why there are better ways to show the TMI system is valid)
>
>
> It is already commonly shown that LT can exist under a structured pruning setting (see [1]). With the numerous discovery of LT on different environments [2], it would be hard to believe that LT does not exist in our method.
>
> Although we might be able to conduct an iterative magnitude pruning (IMP) and find a winning ticket during the training procedure of our baseline network. **We found this (finding the ticket) to be very distanced to the actual method we proposed as**:
> 1. Our method is one-shot, but IMP is iterative;
> 2. Our method evaluates multiple potential ticket epochs, so the existence of one actual winning ticket doesn't mean much;
> 3. After pruning, our method directly fine-tune the network without requiring rewinding its weights to a matching/winning ticket epoch, so the existence of a winning ticket renders no use on retraining.
>
> However, out of curiosity, we tried weight rewinding ResNet-32 to epoch 53 (middle of all potential ticket epochs evaluated by the TMI system) and retraining on it instead of fine-tuning (but for the same number of epochs). We have an end result of 92.91%, which is very close to the optimal 92.96% we achieved with TMI-GKP. Note this 92.91% accuracy was achieved by one-shot pruning, so if IMP is implemented it would almost guarantee to be better, making this a ticket.
>
> Taking a step back and no longer focusing on finding a ticket. **We do agree the connection of TMI and LTH is loose in our initial submission.** We think this is because we expand Zhou et al. 2019's finding (the magnitude increase between the ticket epoch — which was believed to be at epoch 0 at the time — and final epoch of an individual weight is correlated to a pruned network's accuracy retention) to a filter level and replace the ticket epoch from epoch 0 to a certain epoch k. But we didn't properly show that TMI scores still bear any relationship to the accuracy retention of the pruned network after all these relaxations.
>
> We have included such study at [Section 2 of public comment To All Reviewers: Addressing Common Concern](https://openreview.net/forum?id=LdEhiMG9WLO&noteId=ZQ4P2rEcAAt). We are **happy to report that the correlation is significant** (2.55% lead by keeping the TMI preferred filters to keeping the TMI disliked filters). Yet a network with pruning done according to TMI score is much higher to a network with pruning done according to MI score (+1.22% lead), **confirming the TMI score with multiple tickets evaluation is indeed superior to the original single-shot MI score.**
>
> [1] Renda et al. Comparing Rewinding and Fine-tuning in Neural Network Pruning. ICLR 2020
>
> [2] Morcos et al. One ticket to win them all: generalizing lottery ticket initializations across datasets and optimizers. NeurIPS 2019.
>
> ## Cons 7: Misreading on GAL
>
> The reviewer probably read it backward. GAL removes 12% *of* params and reduces 38% *of* FLOPs, but not *to* 12%/38%. So our method outperforms GAL with far more parameters pruned and FLOPs reduced (around 43%).
>
> We have included an extra sentence on the performance evaluation section to clarify such two criteria.
>
> ## Cons 8: Ablation study on $\beta$
>
> The reviewer is correct that we didn't include an ablation on it. We have included it now in A.4.8. As mentioned above, the results support our design of greedy algorithm and confirm our understand of how the scoring system should behave with regard to accuracy retention.
>
> ## Advocacy for (further) impact
>
> Last, although the reviewer has already recognized the novelty and contribution of our work (which we are grateful of). We want to invite the reviewer to take a look at [Section 7 of public comment To All Reviewers: Addressing Common Concern](https://openreview.net/forum?id=LdEhiMG9WLO&noteId=OkONLTQG1Cx), where we discussed why our work can make potential impacts to the realm of lottery tickets and fields outside of neural network pruning. Thanks again.

---

### Official Review · Reviewer_RqjA · 2021-11-01

**Correctness:** 4
**Technical Novelty And Significance:** 3
**Empirical Novelty And Significance:** 2
**Recommendation:** 6
**Confidence:** 3

**Main Review:**

Strengths:
* The paper addresses an important research problem in the field of model compression as most kernel pruning methods suffer from the curse of sparsity and are therefore not computational friendly.
* The paper is well written. The intuition behind each proposed component of the framework (e.g., the proposed TMI score) is well discussed and most of them seem straightforward.
* The ablation study on different filter clustering schemes is insightful.

Weaknesses:
* The experiment setup is somewhat limited. The experiments are conducted on CIFAR10 and TinyImageNet only, so it is unclear whether it would work well on datasets like ImageNet. Moreover, only ResNet experiments are reported. It would be more convincing if the results on other popular CNN architectures are also reported (e.g., densely connected blocks and NAS-searched blocks like EfficientNet).
* As finding winning tickets is sensitive to the learning rate, is there any empirical justification for the learning rate decision?
* Although the proposed greedy approximation algorithm seems to be cost-efficient, it would be great to show the training time compared to baselines.



**Summary Of The Paper:**

The paper consults the empirical findings on the Lottery Ticket Hypothesis and proposes a kernel pruning framework with densely structured outputs. The proposed method seems to give noticeable improvement over SOTA methods with a rather elegant design.

**Summary Of The Review:**

In summary, I feel that although the proposed method seems to be novel, the evaluation part is not yet convincing.

---

> ### Author Response · Authors · 2021-11-19
> **Addressing Weakness 1**
>
> We thanks this reviewer for the positive recognition, detailed reading, and suggestions. We found all of the proposed *weaknesses* by this reviewer to be reasonable and we have made adjustments/improvements to our manuscript accordingly.
>
> ## Weakness 1.1. Lack of ImageNet Experiment
>
> We have now added results on ImageNet and we are once **again happy to report our method maintained the best accuracy retention among all comparable methods.**
>
> Please see Table 2 of revised manuscript and [Section 1 of public comment To All Reviewers: Addressing Common Concerns](https://openreview.net/forum?id=LdEhiMG9WLO&noteId=hLoG26CMAuw) for details.
>
> ## Weakness 1.2. Additional networks to ResNets
>
> In terms of applying our algorithm on additional networks, we certainly concur with the reviewer as the more experiments the better. But we are unable to do NAS tasks like EfficientNet as training the baseline along would cost like 10 days on a V100 GPU. Yet this network is rarely experimented among pruning literature. In fact, out of the 13 methods we compared against, only one literature conducted experiments on EfficientNet. Similar story to DenseNet, although slightly more popular, it is not among the top 20 evaluated pairs among pruning literature according to this survey paper [1]; and our GPUs were running on the ImageNet tasks.
>
> To show that our method may generalize outside of ResNet `BasicBlock` families, we propose the following two alternatives:
>
> As a semi-alternative, we have solved the baseline issue of Tiny-ImageNet experiments and conducted two fair experiments on ResNet-56/101 on Tiny-ImageNet. Where ResNet-101, although still being ResNet, has a different architecture (`BottleNeck`) to ResNet-20/32/56/110. And we are **once again happy to report our method still maintain great accuracy retention across the board with an almost 1% lead on ResNet-101 + Tiny-ImageNet in terms of $\Delta$ Acc.**
>
> **[Update]:** We also conducted a CIFAR-10 experiment on VGG-16 — where VGG Net is the most popular network experimented on CIFAR-10 after ResNet-56 according to [1] — and got the following results:
>
> *VGG-16 on CIFAR-10; FLOPS: 3.14E8; Params: 1.47E7. Methods with $^\odot$ are replicated by [Wang et al., 2020] (the PScratch paper), the rest are drawn from their original papers. Citations are done according to our submitted manuscript.*
>
> |              Method              | Baseline Acc. | Pruned Acc. |     $\Delta$ Acc    | $\downarrow$ FLOPs |
> |:--------------------------------:|:-------------:|:-----------:|:-------------------:|:------------------:|
> | Rethink [Liu et al., 2019]       |     93.63     |    93.62    |  $\downarrow$ 0.01  |          -         |
> | PScratch [Wang et al., 2020]     |     93.44     |    93.63    | $\uparrow$ **0.19** |          ~<50         |
> | PFEC [Li et al., 2019]           |     93.25     |    93.40    | $\uparrow$ **0.15** |        34.2        |
> | ThiNet$^\odot$ [Luo et al. 2017] |     93.99     |    93.85    |  $\downarrow$ 0.14  |          ~<50         |
> | CP$^\odot$ [He et al., 2017]     |     93.99     |    93.67    |  $\downarrow$ 0.32  |          ~<50         |
> | GAL [Lin et al., 2019]           |     93.96     |    92.03    |  $\downarrow$ 1.93  |        39.6        |
> | HRank [Lin et al., 2020]         |     93.96     |    93.43    |  $\downarrow$ 0.53  |        53.5        |
> | TMI-GKP (ours, $\text{PR} = 43.75$%)   |     93.94     |  **94.07**  |  $\uparrow$ 0.13  |        43.39       |
>
> Although two methods have a slightly better $\Delta$ Acc than our proposed TMI-GKP method, these two methods are based on baselines which are much lower than us, yet their end results (Pruned Acc.) are significantly lower than TMI-GKP. **In fact, our method delivers the best end result with a noticeable lead against all compared methods** — which clearly demonstrates the superiority of our method.
>
> We understand that VGG is not a replacement of the DenseNet experiment, but we hope the added experiments and ablation studies may  levitate reviewer's concern on the generalizability of our method.
>
> With one of the field most popular work [SFP by He et al., 2018] only experimented on ResNets (with less datasets evaluated than us) and many other impactful methods (such as [FPGM by He et al., 2019], [PScratch by Wang et al., 2020], [CP by He et al., 2017], and [ThiNet by Luo et al., 2017]) only evaluated CIFAR-10 on ResNet and VGG Net. We hope the reviewer may recognize the contribution of our work; granted our experiment coverage is already a lot wider than many of the aforementioned prior arts.
>
>
> [1] Blalock et al. What is the State of Neural Network Pruning? MLSYS 2020.

---

> ### Author Response · Authors · 2021-11-19
> **Addressing Weaknesses 2 and 3 & Advocacy for (further) impact**
>
> ## Weakness 2: Learning rate and our method
>
> The reviewer is correct that finding the winning/matching tickets is sensitive to the learning rate. But one of the key selling points of our method is we don't find any particular ticket, **so this is not to much a concern to us practice-wise**. Since we also opted not to use weight rewinding after pruning for retraining but just used standard fine-tuning procedure, our choice of learning rate is rather typical as 0.01 (and divide by 10 per a certain epochs).
>
> **Out of curiosity, we did a LT-style weight rewinding for retraining**, with initial lr set to 0.001 and 0.01 on ResNet-32 with CIFAR-10 (baseline: 92.82%, all other setting controlled). **The decrease in learning rate indeed sabotaged our accuracy**: as the former network has a pruned accuracy of 91.53 and the latter network has a pruned accuracy of 92.91 (+1.38% lead). We hope this answers the reviewer's question.
>
> ## Weakness 3: Overheads of greedy pruning procedure
>
> The greedy portion is more of an overhead after training and before fine-tuning, as it determines which grouped kernels to keep and which to prune. So this algorithm has no influence during training nor fine-tuning. However, we agree with the reviewer that time complexity of the pruning overhead should be examined.
>
> Please see [Section 4 of public comment To All Reviewers: Addressing Common Concerns](https://openreview.net/forum?id=LdEhiMG9WLO&noteId=9XqbfCJw5q5). Where we explain the big-O analysis and provide real-world runtime experiments on different networks. We also reflected why multiple reviewers suspect our pruning procedure to be computational-intensive while it is not, and taken actions to improve our presentation.
>
> For a quick overview: The procedure of TMI-GKP is in fact very lightweight. **The TMI-driven clustering procedure has the same big-O time complexity as a single one-shot filter clustering method. Real-world runtime experiment also confirms our inference**: on ResNet-56, the whole TMI-GKP procedure (clustering + pruning) costs less than 180 mins, and the greedy pruning procedure costs less than 5 mins.
>
> (If the reviewer is interested in the inference speedup analysis, please refer to [Section 5 of public comment To All Reviewers: Addressing Common Concerns](https://openreview.net/forum?id=LdEhiMG9WLO&noteId=xDU9Wlg29c4)).
>
> ##  Advocacy for (further) impact
>
> Last, although the reviewer has already recognized the novelty and contribution of our work (which we are grateful of). We want to invite the reviewer to take a look at [Section 7 of public comment To All Reviewers: Addressing Common Concerns](https://openreview.net/forum?id=LdEhiMG9WLO&noteId=OkONLTQG1Cx), where we discussed why our work can make potential impacts to the realm of lottery tickets and fields outside of neural network pruning.
>
> As an short overview: we do — of course, biasedly — believe by proposing **a framework that is superior to the popular filter pruning framework in almost every way, with significant performance lead and much more straightforward procedure**, our method deserves the proper attention of publication. Yet, we would like more scholars to explore the heavily overlooked field of structured kernel pruning due to the potential shown in ablation studies is absolutely huge.
>
> **In addition, our work may have a border implication on Lottery Tickets-related research** (proven that LT-induced heuristics can be used as a guidance of structured pruning, but not just retraining) **or even impact on fields outside of network pruning** (the "multiple tickets evaluation" trick might be universally applied to many tasks to avoid the cost of ticket searching). And we would love to say if these "tricks" can be generalized.
>
>
> Thanks!

---

### Official Review · Reviewer_rPne · 2021-11-02

**Correctness:** 3
**Technical Novelty And Significance:** 4
**Empirical Novelty And Significance:** 3
**Recommendation:** 5
**Confidence:** 4

**Main Review:**

Pros:
1. Dense Structured pruning: this paper has addressed a well-known and crucial problem of kernel-pruning in a structured way yielding a dense pruned network. They have empirically shown how this approach can help improve the representation power of a pruned network (delta increase in performance) and can be easily deployed (less FLOPs and hardware-friendly due to grouped-convolution).

2. Improved Accuracy without Iterative Fine-tuning: one important contribution of this paper is the exhaustive empirical study of their approach with related works (Table 2) with respect to i) need for iterative pruning or ii) special finetuning. Overall, their results imply that the proposed approach of grouped kernel pruning indeed improves the generalisation, but the FLOPs improvement is still either negligible or at par performance as compared to the baselines ( SFP [He et al., 2018] and 3D [Wang et al., 2021b].

3. Enhanced Clustering Scheme: Another important contribution is the better evaluation of filter-clusters by taking filter importance into account. Before this, it was done using a classical clustering Silhouette metric which doesn't take any network context (filter importance) into account.

Cons:
1. Filter Importance and Accuracy Retention? Authors claim that their method of filter-clustering takes accuracy retention into account by following two themes: i) filter importance and ii) "balance between the filter groups" (not all important filters should be aggregated in the same group). However, they have not proven/established any kind of direct relationship between accuracy retention and these aforementioned themes (especially for convolutional filters). For example, any experiment to prove the empirical relation between their defined metric TMI score (to evaluate cluster quality) and accuracy would be enough to prove the point.

2. Loopholes in notations (Algorithm 1): The algorithm 1 on page 7 maintains the matrix M of size (C_in-1 X C_in at Line 3) where C_in represents the number of kernel channels. As per the understanding, it should be of size C_in X C_in. Also, following the proposed algorithm, once a pruning ratio is reached, the algorithm should break the loop instead of continuing (Line 14).

3. Technical issue in algorithm 1: The greedy approach proposed by the algorithm always picks the next best kernel K given the current set of selected kernels (nodes) in a filter-group (graph).
Example: Suppose a filter-group has 6 kernels A,B,C,D,E,F with the distance-matrix (D) as defined below: D(A,B) = 100, D(C,D) = 100
D(A,C) = D(A,D) = D(B,C) = D(B,D) = -10
D(E,F) = 0
All other kernels have a distance value of -2. In such a scenario, if we try to select 4 kernels of maximum value set (subgraph), the answer should be (A,B,C,D) while the algorithm's greedy approach will never take these 4 kernels (nodes) together. It would be better if the authors can clarify this corner case.

Typos:
1. "Abstract" at Page 1: "structured pruning due to it" looks like an incomplete sentence.
2. "Introduction" at Page 1: "methods lie in between" looks grammatically incorrect.
3. "Related Work" at Page 3: Line "Our method is inspired by group convolution, a is well convolutional architecture" has issue in sentence formation


**Summary Of The Paper:**

This paper presents a novel approach in the well-established space of structured pruning methods (kernel-pruning) post-training, which primarily consists of three stages: (i) clusters the filters in a convolution layer into predefined number of groups, (ii) prune the unimportant kernels from each group, and (iii) permute remaining kernels to form a grouped convolution operation and then finetune. This paper is motivated by the idea of achieving grouped convolution which can be efficiently deployed on low-compute end devices, and definitely has huge practical implications.

**Summary Of The Review:**

Overall, I vote for borderline rejection (score 5). Specifically, I liked the idea of introducing a new metric of ticket magnitude increase score (TMI score) to cluster the filters in the first stage based on the "lottery ticket hypothesis". But there are some unproven claims and inconsistencies as pointed out in the cons section.

---

> ### Author Response · Authors · 2021-11-19
> **Addressing Cons 1 to 3 (and typos)**
>
> We thank this reviewer for the positive recognition, detailed reading, and suggestions. We found all of the proposed *weaknesses* by this reviewer to be reasonable and we have made adjustments/improvements to our manuscript accordingly.
>
> ## Con 1: TMI/MI Scores and accuracy retention at a filter level.
>
> We agree this is an important question to answer and we have now added a set of experiments analyzing the relationship between TMI/MI scores and accuracy retention. **The correlation is significant** (2.55% lead by keeping the TMI preferred filters to keeping the TMI disliked filters).
>
> In addition, we further showed a network with filter pruning done according to the TMI score has much higher accuracy to a network with filter pruning done according to the MI score (+1.23% lead), **confirming the TMI score with multiple tickets evaluation is indeed superior to the original single-shot MI score on a filter level.**
>
> Please see [Section 2 of public comment To All Reviewers: Addressing Common Concerns](https://openreview.net/forum?id=LdEhiMG9WLO&noteId=ZQ4P2rEcAAt) and A.4.1 & A.4.2 of our revised manuscript for details.
>
> ## Con 2: Time complexity loophole on Algorithm 1
>
> We said the matrix is $(C_{\text{in}} - 1) \times C_{\text{in}}$ because the initial grouped kernel to preserve is predetermined (as we utilized the multiple restarts trick), so a row can be saved. **However, the reviewer is correct that the algorithm should break the loop instead of continuing.** So the matrix is in fact roughly $\mathbb{R}^{s \times  C_{\text{in}}}$ for $s$ being the number of grouped kernels to preserve — **which makes our big-$\mathcal{O}$ even smaller** as $s \approx 0.5 C_{\text{in}}$ in most experiments. Thank you for pointing this out. We have now fixed the time complexity analysis and pseudocode of Algorithm 1.
>
>
> ## Con 3: Corner case on the greedy pruning approach (and why it is still effective)
>
> The reviewer is again correct that our algorithm may not handle this corner case. However, our algorithm is proposed as an approximation approach, so we are not too surprised to see it can't find the optimal solution in many scenarios, especially on an engineered case. **We'd argue our algorithm is able to find a "close-enough" solution to the optimal one, we know this because**:
> * We conducted a toy experiment on the first ten convolutional layers (80 filter groups pruned) on ResNet-32, all approximated solutions are within 0.5% in terms of the cost difference to the optimal solution (defined by Eq (5)). We discussed about it in A.1.2 but didn't mentioned in the main text, now we have referred it and added a short description.
> * Added ablation studies in A.4.5 confirm the grouped kernels selected by this greedy approach is indeed important, as going against the approach would lead to at most 1.52% of accuracy drop.
> * Added ablation studies in A.4.8 support our design of greedy algorithm and confirm our understanding of how the scoring system should behave with regard to accuracy retention.
>
>
> Please see [Section 4 of public comment To All Reviewers: Addressing Common Concerns](https://openreview.net/forum?id=LdEhiMG9WLO&noteId=9XqbfCJw5q5) for details.
>
>
> **All typos fixed. Thanks again!**

---

> ### Author Response · Authors · 2021-11-19
> **Addressing reviewer's claim made in Pros 2 & Advocacy for (further) impact**
>
>
> ## Pro 2: Why our FLOPs and Params reduction are NOT negligible.
>
> In addition, we noticed in Pro 2 where the reviewer suggests our FLOPs improvement is negligible. **This is in fact intentional because we choose the pruning ratio that may lead to a fair comparison with other methods regarding params or FLOPs reductions**. Granted our method often deliver a higher accuracy retention, we can always prune the network more aggressively to achieve a greater params & FLOPs reduction. In such case, we would end up with a pruned network where its pruned accuracy is on par with others, but with less params and FLOPs. The newly added ImageNet results may serve as an example, where we showed our method in two pruning rates to accommodate methods we want to compare against, it is clearly observable that we may trade pruned accuracy with params/FLOPs reduction.
>
>
> [Update] Per the request of reviewer Dc6d, we further compared our method in a more aggressive pruning rate with GAL [here](https://openreview.net/forum?id=LdEhiMG9WLO&noteId=G6AFUll4-Oz). Our method delivers a $1.17$% lead in terms of $\Delta \text{Acc}$ to GAL with $\downarrow \text{Params/Flops} \approx 60$%. Yet, when compared to Table 2 regarding ResNet-56 on CIFAR-10, **our network still outperforms many of them in terms of $\Delta \text{Acc}$ and majority of them in terms of $\text{Pruned Acc}$ with around $20$% more parameters pruned.**
>
> ## Advocacy for (further) Impact
>
> Last, although the reviewer has already recognized the novelty and contribution of our work (which we are grateful of). We want to invite the reviewer to take a look at [Section 7 of public comment To All Reviewers: Addressing Common Concerns](https://openreview.net/forum?id=LdEhiMG9WLO&noteId=OkONLTQG1Cx), where we discussed why our work can make potential impact to the realm of lottery tickets and fields outside of neural network pruning.
>
> As an short overview: we do — of course, biasedly — believe by proposing **a framework that is superior to the popular filter pruning framework in almost every way, with significant performance lead and much more straightforward procedure**, our method deserves the proper attention of publication. Yet, we would like more scholars to explore the heavily overlooked field of structured kernel pruning due to the potential shown in ablation studies is absolutely huge.
>
> **In addition, our work may have a border implication on Lottery Tickets-related research** (proven that LT-induced heuristics can be used as a guidance of structured pruning, but not just retraining) **or even impact on fields outside of network pruning** (the "multiple tickets evaluation" trick might be universally applied to many tasks to avoid the cost of ticket searching). And we would love to say if these "tricks" can be generalized.
>
>
> Thanks!

---

### Official Review · Reviewer_Dc6d · 2021-11-02

**Correctness:** 3
**Technical Novelty And Significance:** 3
**Empirical Novelty And Significance:** 3
**Recommendation:** 5
**Confidence:** 4

**Main Review:**

Strengths
- The core idea and contribution is stated and explained clearly. This helps the reader assimilate the paper quickly.

Weaknesses
- There are multiple instances of language issues throughout the manuscript e.g., Pg. 3, Section 2, Paragraph 2, first sentence. I encourage the authors to correct them to help the reader process the work seamlessly.
- Could the authors clarify the meaning of "Permuting the retained filters" ? Is it similar to replicating the connections of common input nodes across multiple groups? If so, doesn't it add more FLOPs to the forward pass?
- The nomenclature of dense, sparse and densely structured have a lot of overloaded meaning. It is a little difficult to parse the exact meaning of these terms. I encourage the authors to rethink the terminology and possibly clarify them.
- Pg. 3, last bullet point: The term "cleaner" and other abstract terms can be avoided to maintain focus on the argument at hand.
- A common theme across the Related works and other sections  is the highlighting of computational cost involved in complex pruning heuristics. Could the authors discuss the time-complexity involved in a) the TMI score generation at $W_t$ for the winning ticket and b) TMI score generation across the entire DNN (l layers) and c) TMI scores for multiple groupings, which itself is computationally expensive?
- If we factor in the performance of unsupervised clustering approaches from the Appendices, the Table maintains the same numbers for performance, params and FLOPs across all tested methods. Could the authors clarify these results?
- Could the authors clarify the magnitude increase formula , from Pg. 5 and subsequently used everywhere?
- When discussing the choice of windows for lottery tickets, the authors mention they lookup values from Renda et al.(2020). However, this leaves the choice of windows for new dataset/DNN architectures open. Could the authors discuss more on how they would approach such a target?
- Mathematical notations for k  have been overloaded (Alg. 1 and previously). Also, the variable names have a mix of mathematical notation and pseudo-code. I encourage the authors to reassess their choice of variable names.
- Could the authors discuss their choice of layer pruning ratios ?
- Could the authors clarify the special fine-tuning setup discussed in Section 3.2.2 and the deviation of other baselines from it in more detail? This could add more strength to the argument of loss computational overhead.
- If the pruning and retaining settings mirror the training setup, the terminology of fine-tuning might be incorrectly used. Could the authors clarify if their training and pruning setups are the same?
- The results posted in Table 1 are lower than R. Yu, A. Li, C.-F. Chen, J.-H. Lai, V. I. Morariu, X. Han, M. Gao, C.-Y.Lin, and L. S. Davis, “Nisp: Pruning networks using neuron importance score propagation,” in IEEE Conference on Computer Vision and PatternRecognition, 2018, N. Gkalelis and V. Mezaris, “Fractional step discriminant pruning: A filter pruning framework for deep convolutional neural networks,” in IEEE International Conference on Multimedia & Expo Workshops, 2020 and M. R. Ganesh, J. J. Corso, and S. Y. Sekeh, “Mint: Deep network compression via mutual information-based neuron trimming,” in IEEEInternational Conference on Pattern Recognition, 2020. Could the authors update their results and compare against methods with high performance.
- Overall, the amount of results and discussion presented in the main body of the paper seem insufficient. I encourage the authors to re-organize their manuscript to further support their claims and analyze their results.

After Rebuttal
- I appreciate the author's clarifications on a number of the comments posted.
- After considering the revised version of the paper, I have updated my scores across multiple metrics.
- The update of Table 1 to include a number of results helps emphasize the applicability of the proposed work to a number of datasets. The inclusion of "Amc: Automl for model compression and acceleration on mobile devices" , a top performing method on ResNet50-ILSVRC2012 would help strengthen the comparisons drawn from Table 1.
- At a higher level, a pruning methodology's performance is based on pushing the limits of performance at extreme sparsity levels.
In this context, comparisons to the most sparse models mentioned in works, like GAL, DHP, etc. would help highlight the importance of the current work further.

**Summary Of The Paper:**

The proposed work focuses on kernel pruning. The core idea revolves around grouping filters using a similarity criterion and removing common convolutional kernels with the group. The proposed work explores optimal grouping schemes for filters and after pruning unwanted filters, the retained filters are restructured and the network can be fine-tuned to recoup prediction accuracy.

**Summary Of The Review:**

The method proposed in the work provides an alternative pruning approach that should help decrease the computational complexity. However, overheads involved in repeated formulation of groups and TMI evaluation need to be discussed and compared. The results presented in the manuscript do not possess sufficient depth and are missing analyses.

---

> ### Author Response · Authors · 2021-11-19
> **Addressing Weakness 1 to 5**
>
> We thanks this reviewer for the long and detailed review.
>
> ## Weakness 1: Typos
> Fixed. We apologize for the typos. The overall presentation of the paper has also been polished.
>
> ## Weakness 2: Questions on channel permutation
>
> We does replicate connections as what we did is known as channel shuffling. e.g., as shown in Figure 1 (d), Input Channel 3 is used for both Group 1 and Group 3. But the overall number of connections is not increased compares to group convolution without channel shuffle (3 input channels per each group), **so there is no addition in FLOPs**. And compares to standard convolution, the number of connections is greatly reduced. Please see the sparsity of Figure 1(b) or A.4.4 — where we happened to show how a group convolution may induce params and FLOPs reduction.
>
> ## Weakness 3: Clarification on terms "sparse, "dense," and "densely structured."
>
> "Dense" and "sparse" are probably the two most intuitive terminologies when coming to describe the sparsity of a network, they are commonly found in many survey-style prior arts such as [1, 2]. We agree that "densely structured" — a terminology we made — partially overlaps with "dense," but we found this to be necessary because different literature have different definitions on what is considered a "structured pruning" method. Some of them considered only the dense one are structured [2]; some other believe as long as a certain geometry constraint is followed then it is structured, even if you are removing individual weights from kernels [3]. Thus, **we need a term to distinguish the two, where we found "densely structured" to be a good choice with almost no learning curve.**
>
> **We did polished the wording in Introduction to make it more meticulous.**
>
> [1] Mao et al. Exploring the Granularity of Sparsity in Convolutional Neural Networks. CVPR 2017 (Workshops)
>
> [2] Ma et al. PCONV: The Missing but Desirable Sparsity in DNN Weight Pruning for Real-Time Execution on Mobile Devices. AAAI 2020.
>
> [3] Anwar et al. Structured Pruning of Deep Convolutional Neural Networks. ACM 2017
>
>
> ## Weakness 4: Change of expression on "cleaner" and other abstract terms.
>
> We removed the term "cleaner" and some other adjectives. We hope the reviewer may find it less abstract now.
>
> ## Weakness 5: Time complexity/Runtime analysis of the pruning procedure.
>
> Please see A.2.2 of our revised manuscript and [Section 4 of public comment To All Reviewers: Addressing Common Concerns](https://openreview.net/forum?id=LdEhiMG9WLO&noteId=9XqbfCJw5q5) for details. Where we explain the big-O analysis of the TMI-driven clustering procedure for one layer, on one potential ticket epoch $W_k$, with one clustering scheme, then expand it to multiple potential ticket epochs with multiple clustering schemes. It should answer the (a) (c) questions raised by the reviewer.
>
> We opt to not provide a specific analysis on (b) — the TMI-driven clustering procedure on different conv layers — as the big-O analysis reveals the procedure's time complexity is related to the total number of weights in a conv layer. So the analysis of one layer is directly applicable to another layer.
>
> For a quick overview: The procedure of TMI-GKP is in fact very lightweight. The TMI-driven clustering procedure has the same big-O time complexity as a single one-shot filter clustering method. Real-world runtime experiment also confirms our inference: on ResNet-56, the whole TMI-GKP procedure (clustering + pruning) costs less than 180 mins, and the greedy pruning procedure costs less than 5 mins.

---

> ### Author Response · Authors · 2021-11-19
> **Addressing Weaknesses 6 to 12**
>
> ## Weakness 6: Question on identical Params and FLOPs in table A.4.3.
>
> If the reviewer is referring to table in A.4.3. This is because all listed methods here are clustering the original network into the same number of groups, then all filter groups were pruned with the same pruning rate. The only difference between methods are their filters are clustered differently. For this reason, all pruned networks here have the exact same dimension and architecture (as one can `model_A.load_state_dict(model_B['model_state_dict'])`). Thus, the FLOPs and params reduction are identical across the table.
>
> This is also purposefully done because we intent to control unrelated variables to conduct ablation study on different clustering schemes. If networks with different clustering schemes came in with different Params/FLOPs, we are unable to determine if the observed improvement is actually due to the clustering scheme, or just because some networks having more params/FLOPs.
>
> ## Weakness 7: Expression clarifications on magnitude increase formula.
>
> Sure, we added better notations and descriptions regarding the magnitude increase formula.
>
> ## Weakness 8: How to apply our method to new network/dataset (outside of Renda et al. 2020).
>
> We think the solutions are two-fold:
> 1. Modern literature like [1] have found that lottery tickets are likely to be robust across different datasets, models, and optimizers. Thus, making the discovery transferable. Also granted our method evaluate a series of epoch snapshots as potential tickets, as long as we may have an educated guess, the result will often be acceptable;
> 2. Efficiently finding the winning tickets is a popular topic to study, and scholars have achieved reasonable success on it [2. 3].
>
> We have included a short discussion about it at the end of Section 3.2.2.
>
> [1] Morcos et al. One ticket to win them all: generalizing lottery ticket initializations across datasets and optimizers. NeurIPS 2019.
>
> [2] You et al. Drawing early-bird tickets: Towards more efficient training of deep networks. ICLR 2020.
>
> [3] Tanaka et al. Pruning neural networks without any data by iteratively conserving synaptic flow. NeurIPS 2020.
>
> ## Weakness 9: Overloaded variable $k$ in Algorithm 1.
>
> Fixed, we appreciate the close reading. This is indeed a dangerous overloading which may lead to serious confusions.
>
> ## Weakness 10: Questions on pruning ratio.
>
> We choose the pruning ratio that may lead to a fair comparison with other methods regarding params and FLOPs reductions. Granted other pruning methods often have a params and FLOPs reductions of 40-50% on CIFAR-10, we picked our pruning rate to be 43.75% as it leads to similar params and FLOPs reduction. Similar stories on Tiny-ImageNet and ImageNet experiments. We have added an extra description in the captions of Table 1&2.
>
> ## Weakness 11: Clarifications on special fine-tuning.
>
> Sure, we have further clarified the definition of standard fine-tuning by citing the procedure graph from another paper. We also included a short description on the relationship between iterative pruning and special fine-tuning.
>
> In terms of the deviations, it would be hard to exhaustively walk through all methods with special fine-tuning and explain how they are different from the standard fine-tuning setup defined in Section 3.2.2. But we added a couple more examples in Section 4.2.
>
> ## Weakness 12: Questions on train/fine-tune schedule & Potential incorrect usage of the term "fine-tune."
>
> In Section 4.1 of our initial submission. we stated *"Our pruning settings are largely identical to our training settings except for the learning rate, which is set to 0.01 at the start. For experiments on Tiny-ImageNet, an extra 50 epochs of fine-tuning budget is granted (for a total of 350 epochs spent on fine-tuning.)"*
>
> So no, our training and pruning (recovering) setup are not the same in our initial submission due to having a different learning rate schedule and a different training budget on Tiny-ImageNet experiments. In our revised manuscript, we found 300 epochs of fine-tuning to be sufficient for Tiny-ImageNet experiments, so we now indeed have identical number of epochs for training and fine-tuning (with different learning rate schedule).
>
> We don't know if the reviewer will find this updated setting to be an incorrect usage of the term "fine-tuning." **But we are very confident that this is the correct terminology.** As some heavily cited papers [1, 2] in our field have used the term "fine-tuning" to describe the experiment design with same number of epochs assigned before and after the pruning operation.
>
> - *"we use the same training epochs to train/fine-tune the network"* [1]
> - *"Here we experiment with fine-tuning for more epochs (e.g., for the same number of epochs as Scratch-E"* [2]
>
> [1] He et al. Filter Pruning via Geometric Median for Deep Convolutional Neural Networks Acceleration. CVPR 2019.
>
> [2] Liu et al. Rethinking the Value of Network Pruning. ICLR 2019.

---

> ### Author Response · Authors · 2021-11-19
> **Addressing Weaknesses 13 and 14**
>
> ## Weakness 13: Comparing to experiments with "higher results"
>
> **With all due respects, the reviewer's claim is largely untrue as the results posted in the listed papers are (mostly) not higher.** For the ease of expression, we will refer the performance of a method as $x + y$, where $x$ is the baseline accuracy and $y$ is accuracy change after pruning.
>
> 1. The only two comparable results between our manuscript and [1] are ResNet-56 and ResNet-110 on CIFAR-10. Note [1] shows no baseline and its best performance on:
>     * ResNet 56 on CIFAR-10 is baseline-0.03%. Where our TMI-GKP delivers a baseline+0.16%, and we are already compared to methods with higher performance like baseline+0.07% (3D by Wang et al. 2021) and baseline +0.12% (GAL by Lin et al. 2019).
>     * ResNet-110 on CIFAR-10 is baseline-0.18%. Where our TMI-GKP delivers a baseline+0.64%, and we are already compared to methods with higher performance like baseline+0.05% (FPGM by He et al. 2019) and baseline-0.06% (DHP by Li et al. 2020).
> 2. The two common results between [2] and Table 1 of our manuscript are ResNet-56 and ResNet-110 on CIFAR-10.
>     * For ResNet-56 on CIFAR-10. [2] best shot is 93.59-0.46%. Where our TMI-GKP delivers 93.78+0.16%. We are also already compared to methods with higher results such as 3D (93.69+0.07%), FPGM (93.59-0.33%)... in fact, all methods compared on ResNet-56 on CIFAR-10 in our Table 1 have better results compare to [2] with the exception of SFP (in terms of $\Delta$ acc).
>     * For ResNet-110 on CIFAR-10. [2] indeed has a good result of 93.68+0.31% — beating all other compared methods in our Table 1. But it is still lower than the our TMI-GKP (94.26+0.64%).
> 3. The only comparable result between our manuscript and [3] is ResNet-56 on CIFAR-10. The reviewer is right on this one as [3] delivers 92.55+0.92=93.47%, where TMI-GKP only does 93.78+0.16=93.95%. But [3]'s baseline is significantly lower to other compared methods (which are mostly 93.2%+) and its overall accuracy is noticeably lower than TMI-GKP and 3D (93.69+0.07= 93.76%) listed in our Table 1.
>
> For the listed reasons above, we think ResNet-110 on CIFAR-10 from [2] is probably the only experiment that worth an inclusion. But due to the authors of such work have not disclose its FLOPs and params reduction, we cannot include it in our table as we don't know if it would be a fair comparison. **We do, however, agree with the reviewer that [1] is an important paper, and we have already cited such paper in our initial submission.**
>
> We hope the review may agree with our analysis.
>
> [1] Lin et al. NISP: Pruning networks using neuron importance score propagation. CVPR 2018.
>
> [2] Gkalelis et al. Fractional step discriminant pruning: A filter pruning framework for deep convolutional neural networks.  IEEE International Conference on Multimedia & Expo Workshops, 2020.
>
> [3] Ganesh et al. Mint: Deep network compression via mutual information-based neuron trimming. ICPR 2020.
>
>
> ## Weakness 14: Restructure main text to include more results.
>
> We now added more experiment results and discussion in Section 4.2 in the main text. Unfortunately, due to the page limitation we are unable to include more ablation studies in the main text, but we did managed to slightly expand the introduction of ablation studies in main text with page-jumping anchor placed. Please refer to Section 4.3 at page 9.

---

> ### Author Response · Authors · 2021-11-19
> **Advocacy for a Higher Score**
>
> ## Why we think 3 is too low of a score  (agreed by all other reviewers)
>
> We hope the revision may please the reviewer. We considered 3 to be on the lower end. Especially given out of the 14 "weaknesses," 5 of them are cosmetic suggestions (1, 3, 4, 9, 14), 6 of them are questions or asking for clarifications (2, 6, 7, 10, 11, 12), and 1 of them (13) is for experiment comparison addition which the reviewer probably got it wrong.
>
> So the "core weaknesses" are (5, 8), where the reviewer questions the overheads introduced by our TMI-GKP pruning procedure and the transferability of LT. We believe these are valid concerns and we have carefully addressed the both of them with [Section 4 of public comment To All Reviewers: Addressing Common Concerns](https://openreview.net/forum?id=LdEhiMG9WLO&noteId=9XqbfCJw5q5), A.4.7 of revised manuscript, and the added discussion/citations at Section 3.2.2 - Increase robustness with multiple ticket evaluations.
>
> We hope the reviewer may consider significantly raising the score, as **it seems way too harsh to vote for a 3 for lack of big O time complexity analysis (which is not involved in most pruning literature) and a lack of discussion for a question regarding general LTH.**
>
> While we agree it is hard to identify the "true score" of a paper, but we may take other reviewers' feedback for intuition. There are six "common concerns" simultaneously raised by at least two reviewers of our paper. **All other reviewers raised at least two of such concerns, yet all of them voted for a 5 or 6.** Specifically, reviewer hEhG and 4Wb3 raised all six of such concerns, but both of them still voted for a 5 and seems open to adjustments ([update]: hEhG has upgraded to 6). **Reviewer Dc6d only raised one of the such concerns, which we recognize and appreciate, but we don't think it implies a fair 3.**
>
>
> ## Advocacy on technical significance and novelty (agreed by all other reviewers)
>
> In addition, the reviewer voted for a "2: The contributions are only marginally significant or novel" on our technical significant and novelty. **While all 4 other reviewers voted for 3 or 4, yet almost all of them recognize the technical significance and novelty of our method in their comments.**
>
> Content-wise, at the Introduction and Related Work sections, we have clearly identified that kernel pruning under the context of structured pruning is a heavily overlooked area: with the absolute majority of structured pruning methods focusing on filter-pruning, and prior arts on structured kernel pruning being "poorly" designed and executed (expensive and complex procedure with a lot of tricks used, yet lacks comparable result to modern pruning literature). Multiple reviewers (hEhG, rPne,  RqjA) specifically recognized the novelty of our work in this aspect.
>
> ## Advocacy for further Impact
> We want to invite the reviewer to take a look at [Section 7 of our public comment To All Reviewers: Addressing Common Concerns](https://openreview.net/forum?id=LdEhiMG9WLO&noteId=OkONLTQG1Cx), where we discussed why our work can make potential impacts to the realm of lottery tickets and fields outside of neural network pruning. **We truly believe this is a presentation problems and we are willing to revise our work to make it applicable to a boarder audience.**
>
> As an short overview: we do — of course, biasedly — believe by proposing **a framework that is superior to the popular filter pruning framework in almost every way, with significant performance lead and much more straightforward procedure**, our method deserves the proper attention of publication. Yet, we would like more scholars to explore the heavily overlooked field of structured kernel pruning due to the potential shown in ablation studies is absolutely huge.
>
> **In addition, our work may have a border implication on Lottery Tickets-related research** (proven that LT-induced heuristics can be used as a guidance of structured pruning, but not just retraining) **or even impact on fields outside of network pruning** (the "multiple tickets evaluation" trick might be universally applied to many tasks to save the cost of ticket searching). And we would love to say if these "tricks" can be generalized.
>
> Thank you.

---

> ### Author Response · Authors · 2021-11-30
> **Thanks for the score upgrade, and regarding mentioned methods by Reviewer Dc6d.**
>
> We thank the reviewer for the score upgrade across multiple metrics, and we are grateful to learn that the messages we tried to deliver during the rebuttal phase have reached the other end. We hereby address the two more comments regarding experiment comparison.
>
> ### Comment 1: Comparing with AMC — We can't, as this is a fine-grained (unstructured) method where ours is structured.
>
> **The proposed procedure in AMC is a fine-grained (unstructured) approach;** we know this because:
> > Sec 4. *"Experiment Results: For fine-grained pruning [19], we prune the weights with least magnitude."*
> > Sec 3.2. Action Space: *"Most of the existing works use discrete space as coarse-grained action space... However, we observed that model compression is very sensitive to sparsity ratio and requires fine-grained action space."*
> > Sec 4.2. *"Push the Limit of Fine-grained Pruning."*
>
> **On the other hand, our proposed TMI-GKP method is structured.** We believe we don't need to explain in detail why are the two types of methods are not comparable, as that is the absolute common knowledge of the field, but for the sake of completeness, we will leave a paragraph below — please excuse us if this is something you already known by heart.
>
> **The two types of methods are typically not compared against each other** as unstructured pruning methods are known to be capable of yielding much better accuracy retention (due to having a higher degree of freedom on how and where to introduce sparsity). However, the high accuracy retention of unstructured pruning methods often come at the cost of low deployability: as their pruned networks are filled with unregulated sparsity, where actual compression and acceleration benefits can hardly be gained without relaying on custom-indexing, sparse convolution libraries, or even dedicated hardware devices. [Liu et al., 2019; Mao et al., 2017; Renda et al., 2020.]
>
> To put it simply, unstructured methods replace certain individual weights inside the network with zeros with the network dimensions largely unchanged, yet structured methods remove "groups of weights," sometimes making the pruning network smaller in dimensions. **Our method is *densely structured*,** as it not only removes "groups of weights;" the pruned network of our method is small and entirely dense with no intentional zero mask left — **which is the most hardware/library-friendly type of network in theory.**
>
> One key selling point of our proposed method is we achieve this densely structured result with kernel pruning, which is a method that is considered "unstructured" by nature. **So we gain the advantage of having higher pruning freedom without sacrificing the benefit of having a dense pruned network.**
>
>
> ### Comment 2: Structured pruning methods are typically NOT evaluated with extreme pruning ratio. We happened to already compared to many "most sparse models" mentioned in GAL and DHP.
>
> We agree with the reviewer that at a **very high-level**, a pruning method's performance is based on *"pushing the limits of performance at extreme sparsity levels."* This is very true in most unstructured pruning literature, as pushing extreme pruning ratio will reveal certain mechanisms or behaviors of the network-in-question.
>
> But **structured pruning methods — which are developed with practical concerns in mind — are often NOT evaluated with extreme pruning ratio.** This is because when most (if not all) structured pruning methods are pushed to their extreme "sparsity," the accuracy drop will be huge, and the pruned networks therefore render no practical use. Figure 1 in [1] gives a good example: when a ResNet-56 is structurally pruned to only 1.70x, the accuracy drop is already more than -2%. In contrast, an unstructurally pruned ResNet-56 is pruned to 9.31x, yet its accuracy drop is not even -2%. This is why most structured pruning literature have a $\downarrow \text{Params} \approx 40$%.
>
> **In terms of comparing the most sparse models mentioned in works like GAL and DHP, we happened to already done that for a couple of times:**
> * In GAL, the most sparse model on ResNet-110 + CIFAR-10 is $\downarrow \text{Params} = 44.8$% with $\Delta \text{Acc} = \downarrow 0.76$%. We pruned our model to $\downarrow \text{Params} = 43.52$%, yet our $\Delta \text{Acc} = \uparrow 0.64$%. Which is $\uparrow 1.4$% to GAL, a significant lead.
> * In DHP, the pruning ratios are not so extreme as the most aggressive ones reported on CIFAR-10 are only 50 to 62% with $\downarrow \text{Params} \approx 45$%. Thus, we have already compared with the most extreme DHP models for 4 times.
>
>
> [1] Renda et al., Comparing Rewinding and Fine-tuning in Neural Network Pruning. ICLR 2020
>
> ---
>  We hope the above two responses may answer the reviewer's concerns and push the score to the other side of the aisle. We believe our method deserves proper attention for the reasons listed below, and we think the reviewer would agree that we have carried out a constructive rebuttal.

---

> ### Author Response · Authors · 2021-11-30
> **Added ResNet-50 on CIFAR-10 experiment with $\downarrow$ Params/FLOPs $\approx 60$%, result exceeds GAL by a significant margin (+1.17%).**
>
> ### Comment 2 cont.
>
> Since reviewer Dc6d is interested in our method's performance against the *"most sparse models"* mentioned in works like GAL and DHP. The only CIFAR-10 experiment that is not yet fairly compared is GAL's ResNet-56 on CIFAR-10 with $\downarrow$ Params/FLOPs $\approx 60$%. Thus, we ran the experiment overnight, and here is the result:
>
>
> | Method | Baseline Acc (%) | Pruned Acc (%) | $\Delta$ Acc (%) | $\downarrow$ Params (%) | $\downarrow$ FLOPs (%) |
> |:-:|:-:|:-:|:-:|:-:|:-:|
> | GAL | 93.26 | 91.58 | $\downarrow$1.68 | 65.9 | 60.2 |
> | TMI-GKP ($\text{PR} = 62.5$%) | 93.78 | 93.27 | $\downarrow$ 0.51 | 62.12 | 61.75 |
>
> With a commensurate amount of parameters pruned and FLOPs reduction, **our method has a $\uparrow 1.17$% advantage in terms of $\Delta$ Acc to GAL**. This kind of significant margin — along with the other margins [mentioned below](https://openreview.net/forum?id=LdEhiMG9WLO&noteId=jMdNxNOReyr) — should persuade the reviewer that our method is superior (on accuracy retention) to works like GAL and DHP when set to a more aggressive pruning ratio.
>
> We hope the added experiment may please the reviewer to upgrade the score further as we have solidly addressed all mentioned concerns. Since the 11/29 deadline seems to be extended, please let us know if there is any further question. Thanks.

---

### Official Review · Reviewer_hEhG · 2021-11-03

**Correctness:** 3
**Technical Novelty And Significance:** 3
**Empirical Novelty And Significance:** 2
**Recommendation:** 6
**Confidence:** 4

**Main Review:**


Strengths:

1. The proposed approach is technically sound in general. Each part of the methodology (i.e., clustering, finding LTH, and pruning) is well motivated. The idea to leverage group convolution for kernel pruning is practical.

2. The paper is well written and easy to follow.

Weakness:

1. The empirical evaluations are not self-contained, somewhat weak, and incomplete:
  - Table 1 is not referred to. More experimental results and ablation studies should be included in the main text.
  - The performance gain on CIFAR-10 can be minor compared with existing baselines, e.g., ResNet-32 has 92.82% acc, which is even worse than other baselines.
  - The accuracies of models before pruning on TinyImageNet are apparently lower than other baselines, which is unfair.
  - There are no results on ImageNet.

2. More ablations are in need to fully verify the proposed approach.
  - It is not verified if the findings in Zhou et.al.(2019) still hold in filter pruning. I wonder if the magnitudes of grouped filters still correlate with the final accuracies.
  - The effect of hyper-parameters $\alpha$ and $\beta$ is seldom discussed in the experiment.

3. The proposed pruning pipeline involves multiple steps, which is kind of complex to me. One needs to group the filters and find the optimal group division, identify the lottery tickets and formulate pruning as a maximum edge-weight connected subgraph problem. What is the time consumption for each step in such a pipeline, aside from network fine-tuning after pruning?


Other comments:

1. "we expect such relationship will expand to a filter-level where filters with large TMI scores may help on accuracy retention and therefore be deemed “more important.”: It would be more convincing if such assumptions are empirically verified first before moving on to the next steps.

2. "we must first hypothesize a subset of grouped kernels that are most “distinctive” from each other, which helps to preserve the representation power of the original filter group." : Is this empirically verified? In other words, what if you switch the strategy to keep the least "distinctive" grouped kernels?

3. Have you tried the practical speed-up of pruned networks via grouped convolutions?

**Summary Of The Paper:**

This paper proposes a new approach for kernel pruning, by leveraging group-wise sparsity to obtain a compact architecture that can be parallelly computed. The approach first clusters the kernels based on the defined measurement via recent findings in the lottery ticket hypothesis (LTH), and then develop a simple and efficient greedy approximation algorithm for filter pruning.

**Summary Of The Review:**

Overall this is a technically sound paper. However, my major concerns are two-fold: 1. The proposed methodology is over complicated in my view, yet there is little gain in accuracy. 2. The experiment still needs to be enriched to verify the proposed solution from multiple aspects.

---

> ### Author Response · Authors · 2021-11-19
> **Addressing Weakness 1.1 to 1.4**
>
> We thanks this reviewer for the positive recognition and suggestions. We found many of the proposed *weaknesses* by this reviewer to be reasonable and we have made adjustments/improvements to our manuscript accordingly.
>
> ## Weakness 1.1. Lack of table reference and need more experiments/ablation studies in main text
>
> We now referred Table 1 and added more experiment results in Section 4.2 of the main text. Unfortunately, due to the page limitation we are unable to include more ablation studies in the main text, but we did managed to slightly expand the introduction of ablation studies in main text with page-jumping anchor placed. Please refer to Section 4.3 on page 9.
>
> ## Weakness 1.2. Performance on ResNet-30 on CIFAR-10
>
> We don't quite understand why this is a concern. It is rather common for a pruned network to have lower performance than a unpruned (baseline) network, due to the latter network has more parameters. Table 2 may confirm this understanding.
>
> We suspect the reviewer is trying to say something along the line of "The TMI-GKP method shows a +0.14% improvement from its 92.82% baseline on ResNet-32 with CIFAR-10. But the overall end result (92.82 + 0.14 = 92.96%) is lower than the 93.18% baseline of some other methods, yet one method (3D by Wang et al. 2021) shows +0.09% improvement on the 93.18%. Thus, 3D and other methods with the 93.18% baseline are probably better than TMI-GKP."
>
> If this is the case, we understand the reviewer's concern. However, we'd argue our TMI-GKP method still has the best performance, because:
>
> * 92.82 is a comparable baseline to 92.63 and 93.18 (baselines of other compared methods). Thus, we should only focus on the $\Delta$ Acc, where TMI-GKP is leading.
> * All 93.18 baselines came from the same paper (Wang et al. 2021). And even with a slightly lower baseline to the four 93.18 methods, we still got significantly better end result (baseline + $\Delta$ Acc) than three of them.
> * TMI-GKP is superior to 3D because:
>     * a) we use a much smaller fine-tuning budget (300 v. 560 epochs);
>     * b) our method is one-shot yet 3D is iterative;
>     * c) the same "lower than baseline" problem occurs reversely on 3D to TMI-GKP on ResNet-56 with CIFAR-10 and ResNet-101 with Tiny-ImageNet. In both cases, we have higher baselines, and our pruned performances are much higher (we have a +1.62% lead on ResNet-101 + Tiny-ImageNet to 3D).
>
> We hope the reviewer may agree with our analysis.
>
> ## Weakness 1.3. Baseline issue on Tiny-ImageNet.
>
> We agree with the reviewer's analysis. We have now obtained two baselines with similar accuracy to the compared methods and we are **happy to report our method still maintain great accuracy retention across the board with an almost 1% lead on ResNet 101 + TinyImageNet in terms of $\Delta$ Acc.**
>
> Please see [Section 1.1 of public comment To All Reviewers: Addressing Common Concerns](https://openreview.net/forum?id=LdEhiMG9WLO&noteId=hLoG26CMAuw) and Table 2 of our revised manuscript for details.
>
> ## Weakness 1.4. Lack of ImageNet Experiment
>
> We have now added result on ImageNet and we are once again **happy to report our result maintained the best accuracy retention among all comparable methods.**
>
> Please see [Section 1 of public comment To All Reviewers: Addressing Common Concerns](https://openreview.net/forum?id=LdEhiMG9WLO&noteId=hLoG26CMAuw) and Table 2 of our revised manuscript for details.

---

> ### Author Response · Authors · 2021-11-19
> **Addressing Weakness 2 and 3**
>
> ## Weakness 2.1. Magnitudes increase and filter pruning
>
> We have added a set of experiments analyzing the relationship between TMI/MI scores and accuracy retention. In short, **the correlation is significant** (1.19% lead by keeping the MI preferred filters to keeping the MI disliked filters).
>
> Please see [Section 2 of public comment To All Reviewers: Addressing Common Concerns](https://openreview.net/forum?id=LdEhiMG9WLO&noteId=ZQ4P2rEcAAt) and A.4.1 of our revised manuscript for details.
>
> ## Weakness 2.2. Ablation study on hyperparameters
>
> We agree such ablation study is important and we have now added them. **The results support our design of greedy algorithm at Eq (5) and confirm our understand of how the scoring system should behave.**
>
> Please see [Section 6 of public comment To All Reviewers: Addressing Common Concerns](https://openreview.net/forum?id=LdEhiMG9WLO&noteId=Wf4CefitQGV) and A.4.8 of our revised manuscript for details.
>
> ##  Weakness 3. Time complexity/runtime analysis of the pruning procedure
>
> The procedure of TMI-GKP is in fact very lightweight. **The TMI-driven clustering procedure has the same big-O time complexity as a single one-shot filter clustering method**, and we have already showed our greedy pruning procedure is efficient in theory at the end of Section 3.3 of our manuscript.
>
> **Real-world runtime experiments also confirm our inference**: on ResNet-56, the whole TMI-GKP pruning procedure costs less than 180 mins, and the greedy pruning procedure costs less than 5 mins.
>
> Please see [Section 4 of public comment To All Reviewers: Addressing Common Concerns]((https://openreview.net/forum?) and A.4.7 of our revised manuscript for details. Where we explain the big-O analysis and provide real-word runtime experiments on different networks. We also reflected on why multiple reviewers suspect our procedure to be compute-intensive while it is not, and taken actions to improve our presentation.

---

> ### Author Response · Authors · 2021-11-19
> **Addressing Other Comments.**
>
>
> ### Comment 1: Ablation study on TMI score and filter pruning.
>
> We have now included such study, please see [Section 2 of public comment To All Reviewers: Addressing Common Concerns](https://openreview.net/forum?id=LdEhiMG9WLO&noteId=ZQ4P2rEcAAt) and A.4.1 of our revised manuscript for details.
>
> **The correlation is significant** (+2.55% lead by keeping the TMI preferred filters to keeping the TMI disliked filters). Yet a network with filter pruning done according to the TMI score has much higher accuracy to a network with filter pruning done according to the MI score (+1.23% lead), **confirming the TMI score with multiple tickets evaluation is indeed superior to the original single-shot MI score on a filter level.**
>
> ### Comment 2: Ablation study on the greedy pruning strategy.
>
> We have now included such study (and another variation upon it), please see [Section 3.2 of public comment To All Reviewers: Addressing Common Concerns](https://openreview.net/forum?id=LdEhiMG9WLO&noteId=3OH0KZepcSh). **The studies confirm the grouped kernels selected by our greedy approach are indeed important**, as going against the greedy approach (in two different ways) would lead to at most 1.52% of accuracy drop. Also, added ablation studies in A.4.8 support our design of the greedy algorithm.
>
> ### Comment 3: Speed up analysis.
>
> This is the part we have to unfortunately say no to a reasonable request. We did not, and will continue not to, provide a speedup comparison because of the following two reasons. 1) Such metric is hardly reliable for comparison without an identical environment setup; 2) **ML platforms like pytorch seriously lack optimization on grouped convolution, making the runtime analysis unreliable.**
>
>
> Please see [Section 5 of public comment To All Reviewers: Addressing Common Concerns](https://openreview.net/forum?id=LdEhiMG9WLO&noteId=xDU9Wlg29c4) for details. Where we discuss in length of why torch is slowing us done, what's the reason behind it, and empirically show that it is indeed torch's fault. We also discussed why this won't be an issue for long, and how this likely won't affect the implementation of our method for its intended purposes.
>
> We hope the reviewer may understand why we opt to omit speedup analysis for now.

---

> ### Author Response · Authors · 2021-11-19
> **In terms of the Summery Review and Advocacy for (further) Impact**
>
> ### Method Complexity
>
> We agree that our methodology is not the simplest on description. But compared to all other methods with iterative pruning [1], feature map analysis [2], or even custom training setup [3], we consider our one-shot method to be much more straightforward. Theoretical time complexity analysis and real-world runtime ablation study also confirm the actual lightweight-ness of our method. Our method might only seem complicated because we have to go through the background on lottery ticket and graph search.
>
> Also, since our method is basically filter clustering + grouped kernel pruning, it is hard to scale down any portion of the method. We can't scale down the TMI clustering part to use a single clustering method, as it won't be robust enough (showed in A.4.3). We can't randomize the clustering procedure, as it would lead to worse accuracy retention (showed in A.4.2). And we certainly can't forgo the greedy algorithm with a simple alternative like train-from-scratch grouped convolution or L2 pruning, because they will lead to significant decrease in accuracy (showed in A.4.4 and A.4.6). Yet A.4.8 is also showing us an effective grouped kernel pruning algorithm should consider the two aspects that Eq (5) is considering. So even if there's an alternative approach, it will likely not be much simpler than the greedy one.
>
> [1] Wang et al. Accelerate CNNs from Three Dimensions. ICML 2021.
>
> [2] Yu et al. Accelerating Convolutional Neural Networks by Group-wise 2D-filter Pruning. IJCNN 2017.
>
> [3] Wu et al. Deep k-Means: Re-Training and Parameter Sharing with Harder Assignments for Compressing Deep Convolutions. ICML 2018.
>
>
> ### Lack of Experiments
>
> In terms of the lack of experiments, we hope the added ImageNet results, refined Tiny-ImageNet experiments, and the new ablation studies may better demonstrate the effectiveness of our method and justify our design. Thank you.
>
> ### Advocacy for (further) Impact
>
> Last, although the reviewer has already recognized the novelty and contribution of our work (which we are grateful of). We want to invite the reviewer to take a look at [Section 7 of public comment To All Reviewers: Addressing Common Concern](https://openreview.net/forum?id=LdEhiMG9WLO&noteId=OkONLTQG1Cx), where we discussed why our work can make potential impact to the realm of lottery tickets and fields outside of neural network pruning. Thanks again.

---

### Author Response · Authors · 2021-11-19
**To All Reviewers: Addressing Common Concerns.**

We thanks all reviewers for their time and valuable feedback. As we read through the feedback of all reviewers, we found some common concerns (often with slight variations) raised by multiple reviewers. For the sake of simplicity and better layout, also to provide a more comprehensive response to the raised concerns, we like to respond to these concerns here in a collective manner.

Due to word limit we will have to break up our response into multiple comments. Here's is the Table of Contents for guidance.

# 1. [ImageNet Experiment (and more on Tiny-ImageNet)](https://openreview.net/forum?id=LdEhiMG9WLO&noteId=hLoG26CMAuw)
## 1.1. Troubleshoots on Tiny-ImageNet baseline issue.
# 2. [Ablation Studies related to TMI Score](https://openreview.net/forum?id=LdEhiMG9WLO&noteId=ZQ4P2rEcAAt)
## 2.1. TMI/MI Scores and Filter Pruning.
## 2.2. TMI Scores and Filter Clustering.
# 3. [Ablation Studies on the Greedy Algorithm](https://openreview.net/forum?id=LdEhiMG9WLO&noteId=3OH0KZepcSh)
## 3.1. Ability to approximate the optimal solution.
## 3.2. Contribution to accuracy retention.
# 4. [Runtime and Time Complexity Analysis on the Pruning Procedure](https://openreview.net/forum?id=LdEhiMG9WLO&noteId=9XqbfCJw5q5)
## 4.1. Big O analysis of the clustering procedure.
## 4.2. Runtime experiments on the pruning procedure.
## 4.3. Adjust the pruning procedure to meet a time/computation budget.
# 5. [Speedup Analysis](https://openreview.net/forum?id=LdEhiMG9WLO&noteId=xDU9Wlg29c4)
# 6. [Ablation Studies on Hyperparameters](https://openreview.net/forum?id=LdEhiMG9WLO&noteId=Wf4CefitQGV)
# 7. [Advocacy on the Impact of Our Framework](https://openreview.net/forum?id=LdEhiMG9WLO&noteId=OkONLTQG1Cx)

Thanks in advance for your time and patience.

---

> ### Author Response · Authors · 2021-11-19
> **1. ImageNet Experiment (and more on Tiny-ImageNet)**
>
> 3/5 reviewers (hEhG, RqjA, 4Wb3) have expressed concerns regarding the lack of ImageNet experiment results. We certainly agree that ImageNet results are almost indispensable and we have now added such results — we apologize for not including them in the initial submission, they were still running by the 10/5 deadline.
>
> We are happy to report that **our method demonstrated superior performance against other methods with comparable params/flops reduction**. Please refer to Table 2 of revised manuscript (on the last page).
>
> ## 1.1. Troubleshoots on Tiny-ImageNet baseline issue.
>
> As a side note, we have also addressed the baseline issue regarding our Tiny-ImageNet experiments. The reason we didn't have a comparable baseline is mostly because we scaled Tiny-ImageNet inputs to 32x32 to be directly used in CIFAR network like ResNet-56, while Wang et al. 2021 likely didn't. Although we are still not able to replicate Wang 2021's baseline with the exact training setup listed in their paper, by training with more epochs, we are able to obtain these comparable baselines: 55.59 v. 56.55 on ResNet-56 and 65.51 v. 64.83 on ResNet-101 (in terms of Ours v. Wang's).
>
> We are once again happy to report that **our method has demonstrated comparable performance on ResNet-56 + Tiny-ImageNet** (being the second best, -0.12% to the first with 120 less epochs spent on fine-tuning) and **achieved significantly better accuracy retention on ResNet-101 + Tiny-ImageNet (+0.94% lead to the second best).**

---

> ### Author Response · Authors · 2021-11-19
> **2. Ablation Studies related to TMI Score**
>
> 3/5 reviewers (hEhG, rPne, 4Wb3) have expressed concerns regarding the lack of ablation study on the relationship between the TMI score (or the original magnitude increase score from Zhou et al. 2019) and accuracy retention. While different reviewers asked slightly different questions, we hereby provide a comprehensive set of ablation studies that hopefully may address all potential concerns.
>
> ## 2.1. TMI/MI Scores and Filter Pruning.
>
> We first address the question of whether the TMI/MI score has a relationship with accuracy retention. Although the question is straightforward, it is up to different interpretations as the TMI score was proposed as a tool to determine which clustering result is optimal — which requires multiple tickets evaluation. Thus, it is hard to determine the TMI score of a filter, as it is sensitive to the choice of $k$ in Eq (1).
>
> What we did is we run Eq (1) on the same set of $k$s as our TMI-GKP algorithm, then we rank all filters according to their TMI score under each $k$. Namely, when $k=35$, we may have filter $A$ to be rank $1$, filter $B$ to be rank $2$... and for $k=36$ we may have filter $B$ to be rank $1$ and filter $C$ to be rank $2$... We then sum all ranks of such filters across different $k$, where the filter with smallest sum is considered the one that is most preferred by the TMI system. We denote this sum the *TMI Filter Rank Score*.
>
> We opt to use rank-per-each-$k$, but not simply adding TMI scores of the same filter on different $k$s together and rank all filters, because the former approach is a) less sensitive to potential extreme value introduced by a certain $k$s and b) similar to the proposed scoring mechanism defined in Eq (4), which is used in TMI-GKP.
>
> We zero-mask different filters according to the following five criteria. All experiments were conducted on ResNet-32 (baseline accuracy: 92.82%) with pruning rate set to 50% (half of the filters per each layer are zero masked)
>
> 1. **TMI preferred**: We kept filters that have the lowest *TMI Filter Rank Scores*, and zero-masked the rest.
> 2. **TMI complement**: We kept filters that have the highest *TMI Filter Rank Scores*, and zero-masked the rest. Since the pruning rate is set to 50%, this is essentially taking the complement of the above scheme.
> 3. **MI preferred**: We kept filters that have the highest magnitude increase since initialization, and zero-masked the rest.
> 4. **MI complement**: The complement of above scheme.
> 5. **Random**: Half of the filters per each layer were randomly zero-masked.
>
> The results are:
>
> | TMI preferred | TMI complement | MI preferred | MI complement | Random |
> |:-:|:-:|:-:|:-:|:-:|
> | **64.48**% | 61.93% | 63.25% | 62.06% | 63.92% |
>
> Which clearly demonstrate that both better TMI filter rank scores and MI scores are correlated with better accuracy retention. In addition, **the 1.23% lead of TMI preferred to MI preferred also implies that our TMI system — with multiple tickets evaluation — is superior to the original magnitude increase system.**
>
> ## 2.2. TMI Scores and Filter Clustering.
>
> Specifically, reviewer rPne questioned the relationship between accuracy retention and our clustering determination strategy ("balance between the filter groups" etc) and we thought this might also interest other reviwers. Although reviewer rPne will be satisfied with "any experiment to prove the empirical relation between their defined metric TMI score (to evaluate cluster quality) and accuracy," we additional provide the following experiments to better anatomize our method.
>
> - **TMI Shuffled**: Take a clustering result determined by the TMI scoring system (Eq (5)) and shuffle its filters across different groups. By "shuffle," we mean that if a set of filters are originally in the same group, after the shuffle, no two of the above-mentioned filters will be in the same group anymore.
>
>
> | Model | Baseline (%) | TMI-GKP $\Delta$ Acc (%) | TMI shuffled $\Delta$ Acc (%) |
> |:-:|:-:|:-:|:-:|
> | ResNet-20  | 92.35 | $\downarrow$ **0.39** | $\downarrow$ 0.67 |
> | ResNet-32  | 92.82 | $\uparrow$ **0.14** | $\downarrow$ 0.09 |
> | ResNet-56  | 93.78 | $\uparrow$ **0.16** | $\downarrow$ 0.05 |
> | ResNet-110 | 94.26 | $\uparrow$ **0.64** | $\uparrow$ 0.25 |
>
>
> **The experiment results imply that the TMI scoring system utilized in our method positively contribute to the accuracy retention of pruning**. Although the improvement is not as significant as the one provide by the greedy approach (which we will discuss below), the improvement is consistent. Yet without it, our method may not exceed SOTA performance at all.
>
> We further include a discussion in A.4.3 on why empirical evidences suggest the TMI-driven clustering may not guarantee on finding the best clustering in terms of accuracy retention, but rather a robust policy with more comprehensive considerations done to deliver a "better" and very usable solution.

---

> ### Author Response · Authors · 2021-11-19
> **3. Ablation Studies on the Greedy Algorithm**
>
> 3/5 (hEhG, rPne, 4Wb3) reviewers raised concerns regarding the effectiveness of our greedy approach (Algorithm 1 in the paper). The concerns are mainly two-fold: 1) whether the proposed greedy approach is able to approximately find the optimal solution; and 2) whether the proposed greedy approach may lead to better accuracy retention (namely, if the assumption of "kernels that are most distinctive from each other are better," is valid).
>
> ## 3.1. Ability to approximate the optimal solution.
> For the first question, we actually included a toy experiment (which happened to be exactly what reviewer 4Wb3 suggests) at Section A.1.2. in our initial submission.
>
> We conducted a trial of TMI-GKP with $\beta = 0$ (namely, we looked for a subset of grouped kernels with the greatest inner heterogeneity) on ResNet-32. **For the first ten convolutional layers (80 filter groups pruned), all approximated solutions are within 0.5% in terms of the score difference (defined by Eq (4)) to the optimal solution** — which implies the effectiveness of our approximation algorithm.
>
> We apologize for not mentioning this result in the main text of our initial submission. We now properly referred this result at the end of Section 3.3 with a short description.
>
> ## 3.2. Contribution to accuracy retention.
>
> Reviewer hEhG specifically asked "what if you switch the strategy to keep the least 'distinctive' grouped kernels?" And we found this to be an interesting question. Like the ablation study of TMI score and accuracy retention, this question is also up to the following ~~two~~ three variations:
>
> 1. **Greedy complement**: where we kept the group kernels that were originally pruned in TMI-GKP.
> 2. **Greedy reverse**: where we flip the `argmax` on line 11 of Algorithm 1 to `argmin` with $\beta = 0$ in Eq (5). This means the algorithm is now searching for the next grouped kernel that is "most similar" to the selected grouped kernels.
> 3. **Greedy reverse with $\beta = 1$** (request by reviewer 4Wb3): this is same to the above policy but with $\beta = 1$. This means the algorithm is still searching for the next grouped kernel that is "most similar" to the selected grouped kernels. But among the $C^{\ell}_{\text{in}}$ candidate pruning strategies, it will pick the one with best outer homogeneity (pruned and kept filters are most similar).
>
>
> | Model | Baseline (%) | TMI-GKP  $\Delta$ Acc (%) | Greedy complement  $\Delta$ Acc (%)  | Greedy reverse  $\Delta$ Acc (%) | Greedy reverse with $\beta = 1$ $\Delta$ Acc (%)
> |:-:|:-:|:-:|:-:|:-:|:-:|
> | ResNet-20 | 92.35 | $\downarrow$ **0.39** |  $\downarrow$ 1.5|  $\downarrow$ 1.38 | $\downarrow$ 1.52 |
> | ResNet-32 | 92.82 | $\uparrow$ **0.14** |  $\downarrow$ 1.38 |  $\downarrow$ 0.56 | $\downarrow$ 0.42 |
> | ResNet-56 | 93.78 | $\uparrow$ **0.14** |  $\downarrow$ 0.88 |  $\downarrow$ 0.54 | $\downarrow$ 0.45 |
> | ResNet-110 | 94.26 | $\uparrow$ **0.64** |  $\downarrow$ 0.76 | $\downarrow$ 0.67 | $\downarrow$ 0.61 |
>
> *Previously, we got an outlier result on ResNet-56 + *Greedy preserve* as $\Delta \uparrow 0.01$%. We have updated the result after repeating the same experiment for three times; the mean is now reported.*
>
> (pruning rate set as 43.75%).
>
> **The experiment results once again demonstrate the significant superiority of our greedy approach.** In addition, we observe *Greedy reverse* to have better accuracy retention to *Greedy complement*. We believe this is because the pruning strategy produced by *Greedy reverse* may have overlaps with the one produced by TMI-GKP; yet the pruning strategy produced *Greedy complement* is mutually exclusive with the one produced by TMI-GKP. **This indirectly suggests the grouped kernels selected by our greedy approach is certainly "the better" one,** as only a partial overlap with the TMI-GKP may lead to noticeably better accuracy retention.

---

> > ### Comment · Reviewer_4Wb3 · 2021-11-20
> > **Questions to the Ablation Studies on the Greedy Algorithm**
> >
> > - Q. to 3.1:
> >   - What if $\beta=1$, as suggested in A.4.8.
> > - Q. to 3.2:
> >   - "Random grouped kernels" (i.e., all filters are randomly divided into K groups) can be a strong baseline in the Table listed above.
> >   - How to explain the result of +0.01% Acc. occurred on ResNet-56 with Greedy reverse, which seems to indicate that TMI can be ineffective in some cases.
> >   - Did the authors report the best result on the validation set or the accuracy of the last epoch?

---

> > > ### Author Response · Authors · 2021-11-20
> > > **Quick response to "Q. to 3.2"**
> > >
> > > As we are preparing more experiments to answer your other concerns, we think you might appreciate a quick response on this comment.
> > >
> > > ### 1. "Random grouped kernels" as baseline
> > >
> > > Are you suggesting we should use a randomly generated clustering result as the "base" for different kernel pruning policies proposed here? **We argue using the TMI-driven clustering result is a better choice because:**
> > >
> > > 1. It is part of the proposed method, where it is a good practice to keep other components of a method-in-question unchanged when conducting ablation studies.
> > > 2. This ablation study is focusing on different grouped kernel pruning policies. To grant all policies a fair evaluation, they should have an equivalent search space to begin with — which is something a randomly generated clustering result can't provide.
> > >
> > > May the reviewer elaborate a bit more on why a "random grouped kernels" can be a strong baseline? If the reviewer is interested in the effect of a randomized clustering result, [Section 2. Ablation Studies related to TMI Score](https://openreview.net/forum?id=LdEhiMG9WLO&noteId=ZQ4P2rEcAAt) might worth the reviewer's attention.
> > >
> > >
> > > ### 2. Greedy reverse with +0.01% Acc indicates ineffectiveness of TMI?
> > >
> > > We don't quite get why this indicates the ineffectiveness of TMI, **as TMI determines the clustering result and has no influence on which grouped kernel to prune/preserve inside each filter group**. May the reviewer elaborate a bit more here?
> > >
> > > [Update]: After repeating the experiment for three times, we may confirm the initial +0.01% pruned acc on ResNet-56 with *greedy reverse* was an outlier, please see details [here](https://openreview.net/forum?id=LdEhiMG9WLO&noteId=tx1C6iXQsti).
> > >
> > > ### 3. Details on reported accuracy.
> > >
> > > We report the best test/validation accuracy achieved during the fine-tuning process, same as [1].
> > >
> > > [1] Li et al. Pruning Filters for Efficient ConvNets. ICLR 2017

---

> > > > ### Comment · Reviewer_4Wb3 · 2021-11-20
> > > > **Re: Quick response to "Q. to 3.2"**
> > > >
> > > > Apologize for the ambiguity and thanks for the quick reply.
> > > > - I have noticed the "randomized clustering result" listed in Section 2.
> > > > - According to the definition of "Greedy reverse", the somewhat **counter-intuitive** setting "searching for the next grouped kernel that is "most similar" to the selected grouped kernels." even contributes to an improvement (e.g., +0.01%) over baseline result, which makes me wonder whether the hand-crafted metric can be generally effective (in another word, entities should not be multiplied beyond necessity).
> > > >
> > > > Just kindly remind, the authors may pay attention to the length of the rebuttal (it has been a long rebuttal).

---

> > > > > ### Author Response · Authors · 2021-11-21
> > > > > **Sorry, but we still don't quite understand how it is related to the effectiveness of TMI.**
> > > > >
> > > > > We agree with the reviewer on Occam's razor principle, but we still don't get why does Greedy reverse have a $\Delta +0.01$ accuracy increase has anything to do with the effectiveness of TMI.
> > > > >
> > > > > ### 1. TMI has no influence on grouped kernel pruning policy
> > > > >
> > > > > The TMI scoring system determines the clustering result. Once the clusters are formed (filters assigned into different filter groups), TMI does not influence which grouped kernel to prune/preserve inside each filter group. The set of experiments we conducted here are purely about how different grouped kernel pruning policies perform within the same search space — which are the filter groups preferred by the TMI system — **so we don't quite understand how is a bad grouped kernel pruning policy's $\Delta +0.01$% may have any indication on the effectiveness of TMI, which is about filter clustering.**
> > > > >
> > > > > The effectiveness of TMI is studied at [2. Ablation Studies related to TMI Score ](https://openreview.net/forum?id=LdEhiMG9WLO&noteId=ZQ4P2rEcAAt), which the reviewer already noticed.
> > > > >
> > > > > ### 2. Maybe the reviewer meant the ineffectiveness of the greedy approach?
> > > > >
> > > > > We suspect the reviewer is trying to say the $\Delta +0.01$% accuracy increase implies the ineffectiveness of our greedy pruning policy. As we are doing all the "inner heterogeneity outer homogeneity" analysis, yet our lead is only $+0.13$% to the *greedy reverse* policy on ResNet-32 + CIFAR-10.
> > > > >
> > > > > If this is the case, we understand the reviewer's train of thought. **But this is clearly an outlier, as all other experiments on ResNet20/56/110 + CIFAR-10 suggest significant performance lead in favor of the greedy approach we used in TMI-GKP** ($+0.99$%, $+0.70$%, and $+1.31$%).
> > > > >
> > > > > If our interpretation is correct, we want to highlight that, **practically, there isn't really any "multiplied entries" between the (full-on) greedy approach and the *greedy reverse*** as the former one is already extremely fast. e.g., The full-on greedy approach costs 16 seconds to prune on ResNet-32 (see the second table at [Section 4.2. Runtime experiments on the pruning procedure](https://openreview.net/forum?id=LdEhiMG9WLO&noteId=9XqbfCJw5q5)). Yet, by checking the log, the *greedy reverse* with $\beta = 0$ also costed 16 seconds when pruning the same ResNet-32.
> > > > >
> > > > > ---
> > > > >
> > > > > **Last, to reviewer 4Wb3: we are sincerely grateful for having this conversation with you — not a lot of reviewers are willing to go through this level of deep-dive with the authors. We understand a reviewer has multiple papers assigned, and we will try our best to keep our rebuttal brief and reader-friendly.**
> > > > >
> > > > > ---
> > > > > **[Update]: The  $\Delta +0.01$% result of ResNet-56 with *Greedy Reverse*  was an outlier.** We repeated this experiment for three times and got an average result of $93.24$% ($\Delta -0.54$% to baseline and $\Delta -0.68$% to TMI-GKP, which is significantly lower). Please see details [here](https://openreview.net/forum?id=LdEhiMG9WLO&noteId=tx1C6iXQsti).
> > > > >
> > > > > While we are still willing to discuss with reviewer 4Wb3 regarding how would this experiment relate to the effectiveness of TMI from an algorithm design persecutive. Numerical-wise, the previous $\Delta +0.01$% shouldn't be a concern anymore. We apologize for reporting an outlier at the first place.

---

> > > ### Author Response · Authors · 2021-11-21
> > > **Greedy reverse with $\beta = 1$ experiments added, results in favor of our proposed TMI-GKP method. The $\Delta +0.01$% on ResNet-56 + Greedy reverse was an outlier.**
> > >
> > > ### 1. Added *greedy reverse* with $\beta = 1$ results in favor of TMI-GKP's design.
> > >
> > > Per reviewer 4Wb3's request, we have added three *greedy reverse* with $\beta = 1$ experiments on ResNet-20/32/56 + CIFAR-10. **The results support the design of our greedy algorithm as TMI-GKP demonstrates significant performance lead  ($+1.13$%, $+0.56$%, and $+0.59$%).**
> > >
> > > Note this is a "weird" policy to have, as the set of grouped kernels preserved by this policy will be similar to each other, yet "more similar" to the pruned grouped kernels than greedy reverse with $\beta = 0$. However, the "more similar" to pruned grouped kernels part bears limited meaning, as the preserved grouped kernels are similar to each other after all. So the implication of this policy is limited, other than it is significantly worse than TMI-GKP with the standard greedy approach — **which implies the "inner heterogeneity" of preserved grouped kernels are important.**
> > >
> > > ~~We do noticed that *greedy reverse* with  $\beta = 0$ performs similar to  *greedy reverse* with $\beta = 1$ on ResNet-20/30/110, but not on ResNet-56. We will inspect whether this is because we got a very luck run on  *greedy reverse* $\beta = 0$ on ResNet-32 + CIFAR-10 (the one with $\Delta +0.01$% and causing all these questions), or this is just a trend as the network goes deeper.~~
> > >
> > > ### 2.  The $\Delta +0.01$% on ResNet-56 + Greedy reverse was an outlier.
> > > **[Update]: The  $\Delta +0.01$% result of ResNet-56 with *Greedy Reverse*  was an outlier.** We repeated this experiment for three times and got an average result of $93.24$% ($\Delta -0.54$% to baseline and $\Delta -0.68$% to TMI-GKP). This is also close to *greedy reverse* $\beta = 1$'s result on ResNet-56 + CIFAR-32.
> > >
> > > This is in line with our anticipation. As per the design of Algorithm 1 and Eq (5), the set of grouped kernels preserved by greedy reverse with $\beta = 1$ should be very similar to the set group kernels preserved by greedy reverse with $\beta = 0$. This is because both of them will preserve a set of grouped kernels that are very similar to each other; and as 56.25% of grouped kernels per layer are preserved, **the two policies might very likely end up on the same (or similar) set of grouped kernels**. Empirical evidences support this hypothesis, as the performance differences between the two policies are marginal.
> > >
> > >
> > > We apologize for reporting an outlier at the first place, and we hope this answers the reviewer's questions.

---

> ### Author Response · Authors · 2021-11-19
> **4. Runtime and Time Complexity Analysis on the Pruning Procedure**
>
> 4/5 reviewers (hEhG, Dc6d, RqjA, 4Wb3) raised concerns regarding the complexity of different stages of our pruning procedure; yet the last reviewer (rPne) also showed interests in the big-O analysis of our greedy approach. To address the concerns, we hereby provide a big-O analysis of our pruning procedure, some experiments regarding its runtime, and a discussion on how to adjust the pruning procedure to meet a time/computation budget.
>
> ## 4.1. Big O analysis of the clustering procedure.
>
> Our pruning procedure is basically two-stage: filter clustering and grouped kernel pruning. Since we have already analyzed the big O time complexity of the second stage at the end of Section 3.3, we hereby focus on the time complexity of our filter clustering procedure.
>
> The "rough" time of our TMI-driven filter clustering method is:
>
>
> $CS_{\text{num}} \cdot \mathcal{O}(CS(W^{\ell})) + CS_{\text{num}} \cdot k_{\text{num}} \cdot \mathcal{O}(W^{\ell}),$
>
> where $W^{\ell}$ represents the number of weights in layer $W^{\ell}$, $CS_{\text{num}}$ represents the number of clustering schemes, $k_{\text{num}}$ represents the number of potential ticket epochs evaluated, $ \mathcal{O}(CS(W^{\ell}))$ represents the time complexity of a clustering scheme, and $\mathcal{O}(W^{\ell})$ represent the time complexity of calculating the TMI scores and conduct evaluation upon such scores as defined in Eq (4).
>
> We provide a detailed walkthrough on how we got this big O notation in A.2.2 of the revised manuscript, please refer to that for more information.
>
> In TMI-GKP, such $k_{\text{num}}$ is set to $\leq 35$ and $CS_{\text{num}}$ is $3$ as we have three different clustering schemes available (Section 3.2.1). By the "absorption" law of big O analysis, the theoretical time complexity is only $\mathcal{O}(CS(W^{\ell}))$ — which is identical to a standard single-shot filter clustering procedure.
>
> **Clearly, the TMI-driven filter clustering procedure is not computational-intensive at all.** We think such procedure seems compute-intensive to some reviewers because a) we probably didn't do too good of a job on describing it. To address that, we now polished our wording and added one example run at the footnote of Page 6; b) it is clouded by the background and development introduction of Lottery Ticket Hypothesis and formation of the graph searching problem. These are something we can't bypass, as multiple ablation studies have demonstrated the effectiveness of TMI-driven filter clustering and greedy pruning approach; c) we didn't provide any real-world runtime experiment, which we now have (see below).
>
>
>
> ## 4.2. Runtime experiments on the pruning procedure.
>
> The following experiments are conducted on a 2.00GHz 4 core Intel Xeon CPU and Tesla V100. Evaluated on 35 potential ticket epochs.
>
> In our code implementation, we cluster a layer, then prune it, then move on to the next layer, then prune it again. So the runtime is a product of both the TMI filter clustering procedure and the greedy grouped kernel pruning procedure.
>
>
> | Model | ResNet-20 | ResNet-32 | ResNet-56 | ResNet-101 |
> |:-:|:-:|:-:|:-:|:-:|
> | Total Procedure Time | 2,977 sec (49.62 min) | 4,741 sec (78.18 min) | 10,376 sec (172.93 min) | 19,818 sec (330.3 min) |
>
> We also separately analyzed the runtime of our greedy grouped kernel pruning procedure. This is done by assigning a pre-determined permutation matrix to the network (same effect as clustering), so the actual greedy-only approach will be slightly faster. Also note the runtime of this algorithm is theoretically related to the choice of $\gamma$ in Eq (5), but the greedy procedure itself is so fast to the point it doesn't matter anymore.
>
>
> | Model | ResNet-20 | ResNet-32 | ResNet-56 | ResNet-110 |
> |:-:|:-:|:-:|:-:|:-:|
> | Greedy Pruning Time | 47 sec | 16 sec | 250 sec (4.17 min) | 2,205 sec (36.75 min) |
>
> **We consider this sort of runtime is totally tolerable as an overhead.** When compared to methods involve iterative prune-train cycles, custom loss function, or feature maps analysis, our method is significantly more efficient and applicable to a broader set of pre-trained networks.

---

> > ### Author Response · Authors · 2021-11-19
> > **4.3. Adjust the pruning procedure to meet a time/computation budget.**
> >
> > We have added a section at A.2.3 to discuss the adjustability of our method. Although we have demonstrated in the above section that our method's procedure is in fact reasonably fast, we admit that it can be slow if given a wide network to prune or were ask to evaluate many potential ticket epochs — as our algorithm will have to repetitively evaluate the network with different TMI scores over and over again. We hereby provide several points for adjustability of our method.
> >
> > 1. **Reduce range of ticket window**: If a ticket window is defined to be $[k_1, k_2]$, consider using $[k_1', k_2']$ where $k_1 < k_1'$ and $k_2' < k_2$. So less potential ticket epochs were evaluated.
> > 2. **Add a ticket step**: For a ticket window $[k_1, k_2]$, consider adding a $k_{\text{step}}$ so instead of evaluating $k_1, k_{1 + 1}, k_{1 + 1 + 1}, ...$ we now evaluate $k_1, k_{1 + k_{\text{step}}}, k_{1 + k_{\text{step}} + k_{\text{step}}}, ...$. By doing this, less potential ticket epochs were evaluated, while a wide range of potential ticket epochs from different stage of the network training are still considered.
> > 3. **Relax the granularity of clustering evaluation**: The proposed TMI-GKP determines the optimal clustering scheme at a per layer manner. For CNN model with block-like structures (like ResNet), one may opt to determine the optimal clustering scheme for one layer of the block, then proceed to use such clustering scheme on the whole block.
> > 4. **Adjust clustering schemes**: One may opt to reduce the number of clustering schemes available for the TMI score evaluation. Or one may opt to use clustering schemes which are less computational demanding. Granted the TMI system is likely to capture the "better" clustering scheme among the options, the method would still function, but likely not at its full potential.

---

> ### Author Response · Authors · 2021-11-19
> **5. Speedup Analysis**
>
> 2/5 reviewers (hEhG and 4Wb3) asked for inference speed up provided by our method. This is the part we have to unfortunately say no to a reasonable request. We did not, and will continue not to, provide a speedup analysis because of the following two reasons.
>
>
> Speedup analysis is sensitive to the hardware and implementation of the network, so a proper speedup comparison would require us to replicate the code of other methods, which we simply don't have such manpower to do. In fact, among all compared methods at Table 2, the only one who replicates others' code (Wang et al. 2021) also opts to not include a speedup comparison. Out of 12 methods we compared against, only 5 of them have done a speed analysis and all of them are pre-2019.
>
> Despite the limitation of speed up comparison, we'd agree that an "inner" speedup analysis between our pruned network and the original network is a very reasonable request. **Unfortunately, we can't deliver a reasonable result because `pytorch` is slowing us down as the optimization on grouped convolution in torch is much slower than standard convolution.** This is likely because their current implementation of grouped convolution is simply performing the standard convolution channel-by-channel [2]. Please see these two GitHub issues ([issue_1](https://github.com/pytorch/pytorch/issues/10229), [issue_2](https://github.com/pytorch/pytorch/issues/18631)) and paper [1] and [2]. To provide an empirical prove, running [this test code](https://github.com/pytorch/pytorch/issues/18631#issuecomment-478155467) with `groups = 8`, the average speed of the grouped convolution is 20.29 times slower than the standard convolution despite the former one has less params and flops. See below output for details:
>
> ```
> 1000 loops, best of 5: 780 µs per loop
> 100 loops, best of 5: 16.4 ms per loop
> ```
>
> The two potential solutions to this problem are a) implement our own cuda optimization, which is frankly beyond our ability and scope of the type of paper we submitted; or b) convert the torch model to `ONNX` and hopefully ONNX has a better support of grouped convolution. But unfortunately it doesn't according to this [issue](https://github.com/microsoft/onnxruntime/issues/9192).
>
> **The good news is, both torch and ONNX are working on better grouped convolution optimization** as stated in the responses of their GitHub issues. As scholars already successfully accelerated grouped convolution across different machine learning platform [1, 2], it is reasonable to expect ML platforms to achieve the same goal, with similar or even better efficiency.
>
> Practically, since a grouped convolution with `groups = n` can be implemented as $n$ standard convolution and deploy on $n$ edge devices in a parallel fashion, **our algorithm is still applicable in a production setting and will enjoy the benefits we mentioned in our manuscript.**
>
> We hope this response may help the reviewers better aware the current status of speedup analysis on grouped convolutions, and understand why we are not able to provide such experiment.
>
> [1] P. Gibson et al. Optimizing Grouped Convolutions on Edge Devices. ASAP 2020
>
> [2] Z. Qin et al. Diagonalwise Refactorization: An Efficient Training Method for Depthwise Convolutions. IJCNN 2018

---

> > ### Comment · Reviewer_4Wb3 · 2021-11-20
> > **Partly agree**
> >
> > It is reasonable to attribute the poor real-world speedup to the framework, however, the authors may over-claim the contribution "which enables parallel computing capability and greatly increases the practical deployability of our methods". Considering the real cost is even larger than the standard convolution, I doubt that the "deploy on  edge devices in a parallel fashion" with communication cost is indeed impractical.
> >
> > If the proposed scheme heavily relies on the customized hardware/library, the authors may include more structured pruning methods [1,2,3] as strong baselines instead of channel pruning (which is generally effective to all platforms w or w/o optimized implementation). It seems unfair to directly compare with channel pruning methods on ILSVRC-12 (note that the improvement over FPGM is still marginal).
> >
> > * [1] PatDNN: Achieving Real-Time Dnn Execution on Mobile Devices with Pattern-Based Weight Pruning. ASPLOS2020
> > * [2] PCONV: The Missing but Desirable Sparsity in Dnn Weight Pruning for Real-Time Execution on Mobile Device. AAAI2020
> > * [3] SparseTrain: Leveraging Dynamic Sparsity in Software for Training DNNs on General-Purpose SIMD Processors. PaCT2020

---

> > > ### Author Response · Authors · 2021-11-20
> > > **Re: Partly agree**
> > >
> > > We certainly agree with the reviewer that poor real-world speedup is reasonable to attribute. We already attributed it here, we will highlight it in our code repository, and we are willing to add a discussion in our manuscript. We just don't think it is a fair evaluation of our method.
> > >
> > >
> > > ### 1. Being ahead of torch, but not too far.
> > >
> > >  Assume we all agree the `Conv2d()` standard convolution is the most fundamental operation. The questions are, is vanilla grouped convolution more complicated? If so, by how much?
> > >
> > > We would argue "yes, but not much," as design-wise, the only difference between the two is vanilla grouped convolution requires channel-routing. **Considered `groups` is a supported parameter of the torch's `Conv2d()` method**, the slow speed seems more of a torch's optimization problem than our design problem. As multiple ML platforms claim they will work / are currently working on this optimization, we are rather optimistic about our method performance on "off-the-shelf" ML platforms in the near future.
> > >
> > > **We hope the reviewer won't discredit our work too much because we are slightly ahead of torch, which we believe is something rather common in many research fields.**
> > >
> > >
> > > ### 2. Why we don't want to (and can't) compare with methods exploring intra-kernel sparsity
> > >
> > > This is something worth a long reply as it involves some discussions about the "level of customization" of a method, which is related to the the idea of controlled/regulated sparsity. But as requested, we will try to keep it brief here.
> > >
> > > We don't think it is necessary for us to compare to the suggested methods by reviewer 4Wb3, because **we are not advocating people to use our method in combination with the libraries proposed by [1, 2 of [Section 5 above](https://openreview.net/forum?id=LdEhiMG9WLO&noteId=Y1NGBxAzgql)].** We are mentioning the existence of such optimizations to show that it is not impossible to optimize grouped convolution. And as TF/torch/ONNX are all working on such optimization, it is reasonable to expect good results. However, we agree that our expression can be misleading, and we have improved it now.
> > >
> > > Also, **the suggested methods by reviewer 4Wb3 lack comparable experiments to ours and most pruning literature**, with [1] [2] only disclosing top-5 accuracy on comparable experiments and [3] is not a pruning paper. (ref numbers are subject to [reviewer 4Wb3's comment](https://openreview.net/forum?id=LdEhiMG9WLO&noteId=Y1NGBxAzgql).)
> > >
> > > ---
> > >
> > > Last, we want to briefly mention the concept of the *level of customization* required for a method to work. **This bears no influence on the above conclusions, so please feel free to skip over:**
> > >
> > > * The suggested methods by reviewer 4Wb3 are exploring intra-kernel sparsity, which requires heavy customization done to enjoy the benefit of structured pruning. Where our resultant grouped convolution network is naturally dense and structured. So the deployability of our method is still a lot higher.
> > > * The libraries we mentioned in [1, 2 of Section 5 above] are applicable to all grouped convolutions, where the libraries suggested by reviewer 4Wb3 are mostly customized to their proposed pruning methods.
> > >
> > > So, it is fair to say that, even if we are advocating use of custom libraries — which we are not — the level of customization required of our method is vastly lower than the methods suggested by reviewer 4Wb3. **In fact, structured pruning methods exploring intra-kernel sparsity require the highest level of customization. Yet densely structured methods requires little to no customization at all.**
> > >
> > > ---
> > >
> > >
> > >
> > > ### 3. Parallel capability of grouped convolution is proven
> > >
> > > > Reviewer 4Wb3: Considering the real cost is even larger than the standard convolution, I doubt that the "deploy on edge devices in a parallel fashion" with communication cost is indeed impractical.
> > >
> > >
> > > We don't see how are the two related. The real cost/runtime is longer because torch's lack of optimization, but this has no influence on communication costs. Prior arts — such as the one reviewer 4Wb3 suggested earlier [1] — have also maturely demonstrated the capability of parallel deployment of group convolution models:
> > >
> > > *"Group convolution, which was first introduced in AlexNet [26] for accelerating the training process across two GPUs, has been comprehensively applied in computation-efficient network architecture designs [58,40,55,4,48]."* [1]
> > >
> > > **We don't think we were over claiming, as these claims are rather standard for networks utilizing grouped convolution architectures.** However, we do agree with the reviewer that poor inference speed of grouped convolution on current ML platform worth highlighting. Judging the evidence involve GitHub issue and code running, we think a navigation to our repo would be the proper format — would this alleviate the reviewer's concern? [Update]: we added such information in A.3.2.
> > >
> > > [1] Zhou et al. Dynamic Group Convolution for Accelerating Convolutional Neural Networks. ECCV 2020

---

> ### Author Response · Authors · 2021-11-19
> **6. Ablation Studies on hyperparameters**
>
> 2/5 reviewers (hEhG and 4Wb3) are concerned about the lack of ablation study of hyperparameters used in Eq (4) and Eq (5). We have now added respective ablation studies for all tuned hyperparameters to our manuscript.
>
> In short, we are not tuning $\alpha$ in Eq (4) as it is hard-set to be $0.5$ across all experiments. But we did tried different choices of $\beta$ and $\gamma_{\text{ratio}}$ (a term used to determine $\gamma$ in Eq (5), and we have included a set of experiments on the two hyperparameters. Please refer to A.3.1 for discussion of hyperparameters and A.5.6 for experiment results.

---

> > ### Comment · Reviewer_4Wb3 · 2021-11-20
> > **Is that a statistically significant difference?**
> >
> > I have carefully read the newly added A.4.8. However, I found the best setting only leads to about +0.2% Top-1 Acc. on CIFAR-10. Is that a statistically significant difference?
> >
> > According to "The experiment results suggest $\gamma_\text{ratio}$ should be a relatively small value as increasing its value may
> > lead to worse accuracy retention of the pruned network", it seems that $0.1$ can be a better choice than $0.2$.
> >
> > Besides, it seems that A.3.1 is inconsistent with the best setting reported in A.4.8. How to determine $\beta$ in all experiments? The ablation study on $\beta$ shows that the result is rather robust to the selection of $\beta$, which further weakens the assumption of "outer homogeneity". I wonder the result of Res32-CIFAR10 with $\beta=0$.

---

> > > ### Author Response · Authors · 2021-11-21
> > > **Best setting actually leads to +0.43% acc. Requested experiments support our design.**
> > >
> > > We thank this reviewer for carefully checking out our revision. We have conducted the requested experiments, and we are happy to report the new results support our design.
> > >
> > > ### 1. Best setting actually leads to +0.43% acc.
> > >
> > > First, we like to point out the reviewer probably read the two tables in A.4.8 separately, but in reality the two parameters collectively constitute the "best setting" of an algorithm. In such case, the best setting is 92.96% ($\beta = 1, \gamma_{\text{ratio}} = 0.2$) and the worst is 92.53%  ($\beta = \text{auto}, \gamma_{\text{ratio}} = 1$). **A $+0.43\%$ difference, which we believe is statistically significant.**
> > >
> > > **We agree that the presentation can be better**, and we will emphasize the two tables should be read together by our next revision.
> > >
> > > However, we like to emphasize that $\gamma_{\text{ratio}} = 1$ is a bad parameter choice by principle for reasons listed in A.4.8. If we don't consider $\gamma_{\text{ratio}} = 1$ and remove both the highest and lowest settings of A.4.8, the range of pruned acc of TMI-GKP is $92.8 - 92.65 = 0.15$%. **This suggests our algorithm is pretty robust to the choice of parameters** (A.3.1 and Section 4 of this comment also suggest the same), **which is usually considered a virtue** — as it may reduce the amount of work needed on tuning the algorithm.
> > >
> > >
> > > ### 2. New experiments support the importance of evaluating "outer homogeneity."
> > >
> > >
> > > We added the following two experiments per request:
> > >
> > > | ResNet-32 Baseline  | TMI-GKP ($\beta = 1, \gamma_{\text{ratio}} = 0.2$) $\Delta$ Acc| ($\beta = 0$ $\Delta$) Acc | ($\beta = \text{auto},\gamma_{\text{ratio}} = 0.1$) $\Delta$ Acc |
> > > |:-:|:-:|:-:|:-:|
> > > | 92.82% | $\uparrow$ 0.14% | $\downarrow$ 0.11%| $\uparrow$ 0.15% |). A $+0.43%$ difference, which we believe is statistically significant.
> > >
> > >
> > > * The $\beta = 0$ pruned accuracy is $-0.26$% to our current best. **While we don't think it is an extremely significant difference, it is definitely worthwhile to do the "outer homogeneity" evaluation.** Especially since such evaluation costs no extra time on pruning (both of them cost 16 seconds to prune a ResNet-32 [as shown here](https://openreview.net/forum?id=LdEhiMG9WLO&noteId=IhFfJrcRaa)).
> > >
> > > * **We thank the reviewer for suggesting $\gamma_{\text{ratio}} = 0.1$**, which indeed leads to slightly better result ($+0.01\%$ to the previous best with $\gamma_{\text{ratio}} = 0.2$). We will try this setting on ResNet-110 + CIFAR 10 to see if it will generalize on a deeper network.
> > >     * **[Update]**: We tried the all four ResNets on CIFAR, $\gamma_{\text{ratio}} = 0.1$'s performance does not generalize, please see chart below:
> > >
> > > | | ResNet-20 | ResNet-32 | ResNet-56 | ResNet-110 |
> > > |:-:|:-:|:-:|:-:|:-:|
> > > | Baseline | 92.35 | 92.82 | 93.78 | 94.26 |
> > > |  $\gamma_{\text{ratio}} = 0.1$ $\Delta$ Acc (%) | 91.82 | **92.97** | 93.85 | 94.79  |
> > > | TMI-GKP ($\gamma_{\text{ratio}} = 0.2$) $\Delta$ Acc  (%) | **91.96** | 92.96 | **93.94** | **94.90** |
> > >
> > > **The above results are in line with our conclusion in A.4.8**, where we suggest $\gamma_{\text{ratio}}$ should be a relatively small value. This is supported by the fact that both $\gamma_{\text{ratio}}$ — being relatively small values — performed very well across the tasks. Also, the above results suggest it is beneficial to find the proper $\gamma_{\text{ratio}}$ value, **this is yet another evidence on how a proper "outer homogeneity" evaluation can provide positive contribution** to the accuracy retention of a pruned network.
> > >
> > > ### 3. A.3.1 is consistent with A.4.8.
> > >
> > > The best parameters combination proposed in A.4.8 is $\beta = 1, \gamma_{\text{ratio}} = 0.2$, which is for ResNet-32 on CIFAR-10. In A.3.1, we said:
> > >
> > > > For experiments showed in Table 2, we fix $\alpha$ to $0.5$ and $\gamma_{\text{ratio}}$ to $0.2$ for convenience... For the experiment of ResNet-32 on CIFAR-10, we set $\beta$ to $1$.
> > >
> > > Which is consistent with the results in A.4.8.
> > >
> > > ### 4. How to determine $\beta$? We only tried two settings for most experiments.
> > >
> > > We honestly didn't spend too much of an effort on tuning our algorithm. Especially for $\beta$, since it has the `auto` setting, which often works pretty well out-of-the-box. As $\beta$'s underlying meaning is to balance the "inner heterogeneity outer homogeneity" of preserved grouped kernels. We often just try it out with a relatively balanced value like $2$ (since the first term in Eq (5) usually evaluates way more grouped kernel pairs than the second term) or `auto` (which is designed to be perfectly balanced in terms of number of grouped kernel pairs evaluated). On the occasions we tried both, the differences are not significant at all:
> > >
> > > * ResNet-56 on CIFAR-10: $93.92$% and $93.94$%
> > > * ResNet-50 on ImageNet ($\text{pruning rate} = 75$%): $75.51$% and $75.53$%
> > >
> > > We hope the above material answers the reviewer's questions.

---

> ### Author Response · Authors · 2021-11-19
> **7. Advocacy on the impact of our framework**
>
> Last, and outside of responding to reviewers' concerns, we like to use this set of ablation studies to advocate the impact of our work. As clearly demoed by above ablation studies, different grouped kernel pruning strategy may lead to vastly different accuracy retention. Thus, a better search space (determined by the clustering result, which is related to inductive bias like the lottery tickets and unsupervised clustering schemes) and an even better searching strategy than the proposed greedy one may further advance the potential of our proposed framework. **We believe the power of kernel pruning may go well beyond on our implementation, and we invite our fellow scholars to explore this heavily overlooked realm.**
>
> Although we are certainly biased, we do think **our kernel pruning framework can be as vastly adopted as the filter pruning framework.** Since:
> 1. Both of them are *densely structured*, yet our framework offers a higher degree of freedom on where to prune and naturally comes with many benefits of grouped convolution such as parallel deployability.
> 2. Our method demonstrates significant performance lead to filter/channel pruning methods. We achieve such results with a much more straightforward procedure (no iterative pruning, no intermediate maps analysis, no special training, no special fine-tuning, and no rewinding of any kind required) and often with much less fine-tuning budget (300 v. 560 epochs). **This implies the "raw power" of our framework is likely a lot greater to the maturely developed filter pruning framework**, confirming #1.
> 3. Grouped convolution is yet another maturely studied field. With our method — just utilizing vanilla grouped convolution and hard-set parameters — already being able to achieve performance with such a significant lead, we'd imagine grouped convolutions with more sophisticated design (dynamic channel routing, learnable group structures, etc.) can further boost the performance of our proposed framework.
>
> In addition, we argue our work provides two important contributions to the realm of general lottery ticket research:
>
> 1. **Our work also serves as a proof of lottery tickets induced heuristics can be used to guide a structured pruning strategy, but not just retraining.** This is an often overlooked — if not entirely novel — adaptation despite the popularity of lottery-related research. As most scholars are focusing on using the winning ticket weight rewinding + retraining technique as an alternative to the standard fine-tuning procedure, or to more efficiently find a ticket, or to "train-from-scratch." We have seen Zhou et al. 2019 did lottery-guided unstructured pruning, but a) it is unstructured and b) it is more of a proof of concept, where we proposed an articulated framework on how to do it, what to look out, and how future works would potentially do it better.
> 2. We believe one reason that the above adaptation is lacking is because that the cost of finding a winning ticket is still high. In such case,  the "multiple potential tickets evaluation" trick we introduced in our method may help in levitating such concerns. **This might have an impact beyond the field of neural network pruning**, as scholars of other tasks might find winning ticket-induced heuristics to be useful but are deterred by the cost of winning ticket finding procedures such as the *Iterative Magnitude Pruning* . **But with our trick, now researchers may utilize an approximated version of WT-induced heuristics by just inspecting the weight shifting log saved during training.**
>
> We appreciate most reviewers for recognizing the novelty and significance of our work. We hope the added experiments, ablation studies, and emphasis on further impact may interest you. **We will be on this forum and respond to further questions and concerns in an almost real-time manner, so please don't be hesitate to post a comment. Good weekend :)**

---

### Author Response · Authors · 2021-12-04
**To Reviewer Dc6d, rPne, and RqjA: All Concerns Resolved?**

As the (extended) posting deadline is probably also coming to an end, and given that we have carried out a very comprehensive and up-front rebuttal which is appreciated by all reviewers who have replied yet,

### **we'd like to consider that all raised concerns are adequately resolved by now. Would you please let us know if there are more questions/concerns left?**

We don't want to "DDoS" your mailbox — although we already did plenty, which we apologize — so we will collectively post our message to all reviewers who haven't replied to our latest rebuttal here.

---

## To Reviewer Dc6d:

We appreciate that you find our initial rebuttal have clarified/answered your questions. **We have now further addressed why your two post-rebuttal suggestions** (to compare structured pruning methods with an unstructured pruning method, and to evaluate structured pruning methods with extreme pruning rates) **are generally not applicable in the field of network pruning.** We have listed evidence from the AMC paper you referred to, related prior arts, and empirical results to confirm our claims are true.

We also pointed out why our presented experiments are already compared to other structured pruning methods at more aggressive pruning rates as you would like. And, per your request, **the added experiment also demonstrates the significant superiority of our proposed method** under a more aggressive setting ($\uparrow 1.17$% to GAL).

We hope you may consider further raising the score **or identify any concern that hinders you from raising it to a 6+.** Thank you.


---

## To Reviewer rPne:

We found all weaknesses proposed by you to be concrete and important. You questioned:

* **The effectiveness of TMI/MI scores at filter level:** As we now showed via ablation studies [here](https://openreview.net/forum?id=LdEhiMG9WLO&noteId=9XqbfCJw5q5), the correlation is significant ($\uparrow2.55$% for TMI, $\uparrow 1.19$% for MI, yet our TMI system has a $\uparrow 1.23$% to the original MI system)
* **Loophole in time-complexity analysis:** Fixed, and our big-$\mathcal{O}$  is therefore even smaller.
* **Effectiveness of greedy algorithm:** We now proved our proposed procedure is very effective both by toy experiment (greedy solutions within $0.5$% to their optimal counterparts) and ablation study (at most $\uparrow 1.52$% lead by following the greedy scheme to its counters).

**We consider your questions to be very "clear-cut" and our up-front rebuttal should resolve your concerns in their entirety.** We also addressed one more of your comments (a potential misunderstanding) and emphasized our novelty/contribution [here](https://openreview.net/forum?id=LdEhiMG9WLO&noteId=wIt8Et3vowA). Granted we are very much a borderline case right now, we hope the added material may persuade you for a higher score. Thanks.

---

## To Reviewer RqjA:

We appreciate your recognition and we found the weaknesses you raised are valid and constructive. You questioned:

* **The lack of ImageNet results and experiments outside of the ResNet family.** We have now added the ImageNet results and VGG experiments; our method still performs well in these experiments.
* **The choice of learning rate.** We explained why the ticket-finding learning rate is not a concern to our procedure: as we are doing multiple tickets evaluations, and our method does not require weight rewinding for retraining. So, the standard fine-tuning `lr = 0.01` should be a suitable initialization. We also confirm this with an extra experiment [here](https://openreview.net/forum?id=LdEhiMG9WLO&noteId=hTMPnj-u5vP) ($\uparrow 1.38$% lead to `lr = 0.001`).
* **Overhead of pruning procedure.** We have provided a careful big-$\mathcal{O}$ analysis which suggests the TMI clustering procedure is very lightweight (same as any single-shot filter clustering method). Added real-word run-time experiments [here](https://openreview.net/forum?id=LdEhiMG9WLO&noteId=9XqbfCJw5q5) also confirm our understanding.

Like above, **we also believe your feedback is constructive and "clear-cut," which allows us to do an up-front rebuttal that should resolve all your concerns.** We hope you may consider raising the score especially granted we are at borderline right now. Please see if the added emphasis on novelty and contribution [here](https://openreview.net/forum?id=LdEhiMG9WLO&noteId=wIt8Et3vowA) may persuade you further. Thanks.

---

> ### Author Response · Authors · 2021-12-04
> **To Reviewer 4Wb3: All Concerns Resolved?**
>
> ## To Reviewer 4Wb3:
>
> We thank you for the detailed read and constructive feedback. You have raised 9 concerns in your initial review. For an overview, 5 of them are related to adding more experiments (#1.1, 3, 4, 5, 6); we did all of them and believe **the good performance demonstrated throughout the added experiments and ablation studies should levitate these concerns**. 2 of them (#1.2 and 8) — with all due respects — are due to misreading of your side, **which we have pointed out why and improved our presentation where applicable.**
>
> We then had some healthy exchange two weeks ago, where questions regarding #2 speed up analysis, #8 ablation studies on hyperparameters, and the benefit of evaluating both "inner heterogeneity and outer homogeneity" were raised. We believe **with the plethora amount of added experiments [here](https://openreview.net/forum?id=LdEhiMG9WLO&noteId=Wf4CefitQGV) and [here](https://openreview.net/forum?id=LdEhiMG9WLO&noteId=3OH0KZepcSh), the latter two concerns are well addressed** as we confirm the algorithm works as we perceived with different parameter settings. Yet, experiments from multiple angles support the design of doing both evaluations (particularly, the "outer homogeneity" evaluation adds only negligible runtime to the algorithm).
>
> We may have some miscommunications or even a disagreement on #2. We have now clarified the potential miscommunication and addressed this problem in length [here](https://openreview.net/forum?id=LdEhiMG9WLO&noteId=CLhYr1n_MZi). In short, we agree that the slow speed of current grouped convolution implementation on ML platforms is vital to attribute (which we now did), but this is something that will be optimized according to the platforms' roadmap; prior arts also confirmed such optimization is totally doable.
> **We hope the reviewer won't fault us too much just because we are a little ahead of `torch`, as this is a common scenario in many research fields.** We argue this is especially the case in terms of grouped convolution. As "over-faulting" the slow speed of grouped convolution to the algorithm itself — instead of the ill-implementation of grouped convolution in torch — will probably kill off most grouped convolution-related research, which is certainly against the norm of our field.
>
> **We have added 10+ experiments per your requests during the rebuttal, and we are thankful that you are willing to do such a deep dive with us the authors.** We hope our hard work — together with your help — will pay off. Please see if our comprehensive rebuttal and added emphasis on novelty/contribution [here](https://openreview.net/forum?id=LdEhiMG9WLO&noteId=wIt8Et3vowA) may persuade you to raise the score. And please let us know if there is any concern you considered not resolved and we are certainly willing to response. Thanks again and thank you for putting the OpenReview format in good use.

---

### Decision · Program_Chairs · 2022-01-20

**Decision:**

Accept (Poster)

**Comment:**

This paper studies structured pruning methods, called kernel-pruning in the paper which is also known as channel pruning for convolutional kernels.  A simple method is proposed that primarily consists of three stages: (i) clusters the filters in a convolution layer into predefined number of groups, (ii) prune the unimportant kernels from each group, and (iii) permute remaining kernels to form a grouped convolution operation and then fine-tune the network. Although the novelty of the method is not high, it is simple and effective in experiments after the supplementary sota results in the long rebuttal. Majority of reviewers increase their ratings after the rebuttal (though one reviewer promised this but forgot to act), while some reviewers have concerns on the fairness to other authors by adding lots of new results in unlimited rebuttal and refuse to check more. In terms of the top end of performance, a reviewer thinks that "the authors haven't quite exceeded the results from existing works ("Discrimination-aware channel pruning for deep neural network" and "Learning-compression” algorithms for neural net pruning" for CIFAR-10 and many others on ImageNet)". In all, this work indeed lies on the boundary. After a discussion with other committee members, we recommend the acceptation of this work, if the authors could incorporate all the new results in rebuttal and get the reproducible codes released in the final version.

---

> ### Public Comment · ~Shaochen_Zhong2 · 2022-01-29
> **Thank you, and a few more clarifications (2/2).**
>
> ## 3. One reviewer's claim on CIFAR-10 results is not true.
>
> The meta-review suggests
> > "a reviewer thinks that the author haven't quite exceeded the results from existing works ([1] and [2] for CIFAR-10 and many others on ImageNet)."
>
> **This reviewer's claim is simply not true regarding CIFAR-10.**
>
> [1]'s only comparable result on CIFAR is ResNet-56 in its Table 1, which is 93.80% baseline + 0.01% $\Delta$ = 93.81% pruned acc. Our proposed TMI-GKP has a 93.78% baseline + 0.16% $\Delta$ = 93.94% pruned acc. Further, the abovementioned [1]'s result also relies on dynamic pruning rate, a trick we opted to not use (highlighted on Page 9). Without this trick, [1] has 93.80% baseline - 0.31% $\Delta$ = 93.49% pruned acc, which is much lower than us.
> [2] is an unstructured pruning method, which is incomparable to the structured pruning method we proposed for reasons listed in Introduction. We also explained this in-length to Reviewer Dc6d [here](https://openreview.net/forum?id=LdEhiMG9WLO&noteId=jMdNxNOReyr).
>
> **We'd admit our result on ImageNet (75.53%) is indeed not the best**. e,g., [3] shows 75.90% with more params pruned. **However**, we'd like to note a) [3] uses 420 epochs of iterative prune-train cycles (where ours is one-shot with 90 epochs), and b) [3] is an impressive "pastiche-style" method build upon tons of prior filter/channel/layer/resolution pruning literature, but ours is a "rough and simple" (but effective) adoption of the grouped kernel pruning framework.
> Given the overlooked nature of kernel pruning, our goal is to bring attention to it. We therefore can't introduce a sophiciated adoption of this framework due to page limitation, nor it is as helpful (to our goal) as the simple one-shot-no-trick one we presented.
>
> ## 4. Our rebuttal was posted before the deadline.
>
> The meta-review suggests
> > "some reviewers have concerns on the fairness to other authors by adding lots of new results in unlimited rebuttal and refuse to check more."
>
> **We like to note that all our rebuttals were posted by Nov 19, where the reviewer deadline is Nov 29** (and such deadline got further extended for an unclear duration). Our latests experiment addition is [this](https://openreview.net/forum?id=LdEhiMG9WLO&noteId=G6AFUll4-Oz), posted on Nov 30; but this is due to we were responding to reviewer Dc6d's [post-rebuttal edit](https://openreview.net/forum?id=LdEhiMG9WLO&noteId=06pMCRzDCOs) modified by Nov 29. And we imagine if an "extension to Nov 29" means anything, that'd be Nov 30.
>
> After Nov 30, we have only edited our posts for formatting purposes (highlights, hyperlinks, language, etc.) and digested our rebuttal to either urge a response or provide our AC a summary with proper context. No more new results were added.
>
>
> ---
>
> [1] Zhuangwei Zhuang et al. Discrimination-aware Channel Pruning for Deep Neural Networks. NeurIPS 2018.
> [2] Miguel Á. Carreira-Perpiñán et al. "Learning-Compression" Algorithms for Neural Net Pruning. CVPR 2018.
> [3] Wenxiao Wang et al. Accelerate cnns from three dimensions: A comprehensive pruning framework. ICML 2021.
>
> Thank you for you time and patience, if you have stayed here for this long. We once again like to emphasize that **our added comments are purely for public record purposes, and they are by no mean a rebuttal nor complaint**. We have already asked for more attention form the committee and we are grateful for the turnout.
>
> Sincerely,
> *Paper 2257* Authors
>
> ---
>
> (We made this account to add a public comment, it seems OpenReview doesn't allow adding public comment to your own submission after the final decision.)

---

> ### Public Comment · ~Shaochen_Zhong2 · 2022-01-29
> **Thank you, and a few more clarifications (1/2).**
>
> We authors are grateful for the acceptance under a rating of `76655`; especially while the `655` reviewers were not responding to our latest rebuttal. **We appreciate the members of reviewing committee were paying extra attention to our long rebuttal**, and we are glad our effort has been paid off.
>
> For us and many members of the community, the OpenReview archive — especially the meta-review — provides scholars a valuable resource to quickly digest a paper. And for this reason, **we like to "clarify" a few points mentioned in the meta-review, for public record purposes.** This is by no mean a rebuttal nor complaint.
>
> ## 1. The proposed TMI-GKP is not a channel pruning method.
>
> The meta-review reads
> > "kernel-pruning in the paper which is also known as channel pruning for convolutional kernels."
>
> This statement is a bit ambiguous and against the "default" interpretation of *channel-pruning*. Channel-pruning refers to pruning all $i$-th kernels out of all filters within a layer, but kernel pruning is more fine-grained than that: as you can prune the $i$-th kernel out of one filter, but $j$-th kernel out of another filter, and so on...
>
> The meta-reviewers are technically not wrong, because kernel pruning are indeed "channel pruning for convolutional kernels" if we consider *only one single filter* to be the whole input source. But that is probably not the default mindset when we think about CNNs or channel-pruning, so we like to add a note here.
>
> **Here is [an illustration](https://gist.github.com/choH/02fa516d5dce3805b884385f9225fbfe) we made to distinguish filter, channel, kernel, and grouped kernel pruning (our proposed method). We hope it helps.**
>
> ## 2. Novelty is recognized by multiple reviewers.
>
> The meta-review claims
> > "the novelty of the method is not high."
>
> This is again technically not wrong, since we are not the first to discover the kernel pruning + grouped convolutions combination (see Related Work). However, we'd like to note this combination is heavily overlooked and poorly implemented by the few available prior arts for various reasons (see Introduction and Related Work); yet it has HUGE potentials.
>
> **This framework — in our opinion — is superior to the popular filter pruning framework in almost every way**, as both of them offer *densely structured* pruned network which are most hardware/library-friendly (something that can't be obtained via unrestrained kernel pruning operations). But ours allows a higher degree of freedom on where to prune and naturally comes with many benefits of grouped convolution, such as parallel deployability.
> Performance-wise, our one-shot-no-trick implementation beats many modern filter pruning methods with much more sophiciated procedure/tricks, larger fine-tuning budget, etc. (full analysis [here](https://openreview.net/forum?id=LdEhiMG9WLO&noteId=OkONLTQG1Cx)).
>
> **3/5 reviewers ([rPne](https://openreview.net/forum?id=LdEhiMG9WLO&noteId=MDPJ0GzGHVR), [RqjA](https://openreview.net/forum?id=LdEhiMG9WLO&noteId=G2vWQKgY-hd), and [4Wb3](https://openreview.net/forum?id=LdEhiMG9WLO&noteId=5Hj0pWsf_w_)) also agree our work to be novel.** And we'd love to see more methods within this framework to be developed.